# Keratin-mediated hair growth and its underlying biological mechanism

Seong Yeong An[1], Hyo-Sung Kim[2], So Yeon Kim[1,7], Se Young Van[1], Han Jun Kim[2,8], Jae-Hyung Lee [3], Song Wook Han[4], Il Keun Kwon[5], Chul-Kyu Lee[6], Sun Hee Do [2✉] & Yu-Shik Hwang [1✉]

Here we show that intradermal injection of keratin promotes hair growth in mice, which results from extracellular interaction of keratin with hair forming cells. Extracellular application of keratin induces condensation of dermal papilla cells and the generation of a P-cadherin-expressing cell population (hair germ) from outer root sheath cells via keratin-mediated microenvironmental changes. Exogenous keratin-mediated hair growth is reflected by the finding that keratin exposure from transforming growth factor beta 2 (TGFβ2)-induced apoptotic outer root sheath cells appears to be critical for dermal papilla cell condensation and P-cadherin-expressing hair germ formation. Immunodepletion or downregulation of keratin released from or expressed in TGFβ2-induced apoptotic outer root sheath cells negatively influences dermal papilla cell condensation and hair germ formation. Our pilot study provides an evidence on initiating hair regeneration and insight into the biological function of keratin exposed from apoptotic epithelial cells in tissue regeneration and development.

[1] Department of Maxillofacial Biomedical Engineering, College of Dentistry, Kyung Hee University, Seoul 02447, Republic of Korea. [2] Department of Veterinary Clinical Pathology, College of Veterinary Medicine, Konkuk University, 120 Neungdong-ro, Gwangjin-gu, Seoul 05029, Republic of Korea. [3] Department of Oral Microbiology, College of Dentistry, Kyung Hee University, Seoul 02447, Republic of Korea. [4] KeraMedix Inc, # 204, Open Innovation Bld, Hongryeung Bio-Cluster, 117-3 Hoegi-ro, Dongdaemun-gu, Seoul 02455, Republic of Korea. [5] Department of Dental Materials, College of Dentistry, Kyung Hee University, Seoul 02447, Republic of Korea. [6] Headquarters of New Drug Development Support, Chemon Inc. 15 F, Gyeonggi Bio Center, Cheongju, Gyeonggi-do 16229, Republic of Korea. [7] Present address: Department of Dental Hygiene, College of Health Science, Cheongju University, Cheongju 360-764, Republic of Korea. [8] Present address: Terasaki Institute for Biomedical Innovation, Los Angeles, CA 90064, USA. ✉email: shdo@konkuk.ac.kr; yshwang@khu.ac.kr

Keratin is a cytoskeletal protein that forms intermediate filaments within epithelial cells and participates in maintaining the strength of the cells[1]. It is a major protein found within the hair that contributes to its mechanical strength[2]. Human hair consists of three main layers: the medulla in the center of the hair, the cortex surrounding the medulla, which contains a fiber mass mainly consisting of keratin protein, and the cuticle, the outer layer of the hair shaft[3]. During hair growth, dermal papilla (DP) cells secrete various paracrine factors to induce migration of stem cells from the bulge region of the outer root sheath (ORS) to the upper region of the follicle, and the migrated cells become transit amplifying cells, which then undergo differentiation into matrix cells. Hair growth is initiated by cortical cells differentiated from matrix cells located in the follicle bulb region, and a large amount of keratin is synthesized mainly in the cortex[4–6]. Deposition and rearrangement of keratin filaments are followed by the assembly of keratin-associated proteins and intracellular deposited keratin in spindle-shaped epithelial cells of the cortex, and the assembly is stabilized by the formation of inter- and intra-molecular disulfide bonds[7]. At the stages of the anagen-catagen transition of the hair cycle, apoptosis of cells begins to appear in the epithelial strand, and then the apoptotic cells are phagocytosed by macrophages and neighboring epithelial cells. Ultimately, keratin remains the main protein in the hair[8–11].

In our previous study, mouse models with full-thickness dorsal excisional wounds were used to assess the effect of keratin-based hydrogels on wound healing[12]. Interestingly, hair growth was observed only in areas treated with keratin hydrogel, along with accelerated wound healing, which led us to study the biological function of hair-derived keratin in hair regeneration. In this study, intradermal injection of human hair-derived keratin promoted hair follicle formation and subsequent hair growth in a mouse model. Extensive research has been conducted to elucidate the biological function of keratin as an intermediate filament participating in intracellular signaling pathways and the mechanical role of keratin, and as an intracellular scaffold modulating cell stiffness and morphology in response to microenvironmental changes[13,14]. However, the control of cellular behavior via the extracellular interaction of keratin with cells has not been well studied, especially in hair growth. Hence, the underlying mechanism through which keratin stimulates hair growth was examined by studying the interaction of keratin with DP cells and ORS cells, which are known as the main types of cells that regulate hair growth and regeneration, and the whole experimental procedure is illustrated in supplementary fig. 1. DP condensation and the generation of a P-cadherin expressing cell population (hair germ, HG) were induced via the extracellular interaction of DP and ORS cells with keratin. Furthermore, DP condensation and P-cadherin-expressing HG formation were mediated by spatial exposure from TGFβ2-induced apoptotic ORS cells and deposition of keratin, which represents the role of injected exogenous keratin in a similar manner as the exposed keratin from apoptotic ORS cells during hair cycle. Our pilot study provides a possible explanation that keratin is not only a structural protein of hair but also a factor that induces hair regeneration.

## Results

**Intradermal injection of hair-derived keratin promotes hair growth**. First, we performed in vivo experiments in mice to evaluate keratin-mediated hair growth by injecting hair-derived keratin into the hair-removed dorsal skin area. The dorsal area of C57BL/6 mice was shaved with an animal clipper, and 0.5% (w/v) and 1.0% (w/v) keratin were injected. Twenty-eight days after keratin injection, hair growth and hair follicle formation in keratin-injected C57BL/6 mice were analyzed for the number and stage of hair follicles. We found that hair growth and the formation of anagen follicles were promoted, compared to nontreated mice (Fig. 1a–d). Only a single injection of keratin resulted in much higher hair growth compared to the control, and almost equivalent hair growth compared to minoxidil, applied every day for 28 days. In addition, in situ RNA hybridization showed an increase in the Lgr5-positive cell populations in the lower bulge region and hair follicles after keratin injection (Supplementary Fig. 1). Such keratin-mediated hair growth was confirmed in a separate mouse study, and the promoting effect of keratin injection on hair growth was verified. There were no differences in the number of hair follicles formed between mice injected with keratin and those treated with minoxidil (Supplementary Fig. 2a–c). However, the number of anagen follicles and the size of hair follicles were found to be increased in keratin-injected mice compared to minoxidil-treated mice (Supplementary Fig. 2d–f). Keratin injection-mediated hair growth was observed throughout the surface of the back skin of mice and mixed stages of hair follicles were also found, which might be due to the dispersion of keratin solution after injection, and the injected keratin remained up to 2 weeks after injection (Supplementary Fig. 3).

**Keratin induces condensation of DP cells and germ formation from ORS cells in vitro**. To understand how injected keratin induces hair follicle formation and growth, we studied the interaction of keratin with DP and ORS cells, which are known to be the main cells participating in hair follicle formation[4–6,15]. The most distinct characteristic of DP cells exposed to keratin for 3 days was their condensation to form spherical aggregates (Fig. 1e, f) with high expression levels of β-catenin, SOX2, CD133, and alkaline phosphatase (ALPase) (Fig. 1f), which is a molecular identity signature reflecting the hair-inductive property of DP cells[16,17], and high expression levels of FGF7, FGF10, and BMP6 (Fig. 1f), reflecting paracrine factors controlling hair growth[18] (keratin-mediated condensation of DP cells on day 1 and time-lapse images are presented in Supplementary Fig. 4). However, the growth of DP cells was suppressed upon exposure to keratin (Supplementary Fig. 5a), and relatively lower Ki67 expression and BrdU incorporation were observed in keratin-treated DP cells (Supplementary Fig. 5b–d). Condensed DP cell aggregates contained cells expressing Ki67, and β-catenin was expressed in the contact region between cells within DP cell aggregates (Supplementary Fig. 6), in which DP cells remained viable during culture (Supplementary 7). Such keratin-mediated condensation of DP cells was observed when cells were seeded at different cell densities on Matrigel (Supplementary Figs. 8 and 9), and no difference in DP cell condensation was found when they were treated with different keratin concentrations (Supplementary Fig. 10). RNA sequencing analysis showed downregulation of the expression levels of genes associated with cell division and upregulation of mRNAs encoding proteins related to integrins, growth factors, migration, and extracellular matrix organization (Supplementary Fig. 11a, b). Real-time qPCR analysis showed upregulation of various genes, such as *CD133, SOX2, corin, SHH, versican*, β-*catenin, BMP6, FGF7*, and *FGF10*, known as a molecular identity signature reflecting the hair-inductive property of DP cells[16–19] (Supplementary Fig. 12). In addition, the effect of keratin on maintenance of the condensed DP cell aggregates was analyzed. The DP cell spheroids showed higher levels of gene expression, indicating the hair-inductive property of DP cells as compared to a DP cell monolayer (Fig. 2a). The spherical shape of DP cell aggregates was consistently maintained, showing high levels of

expression of β-catenin, SOX2, CD133, ALPase, FGF7, FGF10, and BMP6 (Fig. 2b–d, Supplementary Fig. 13) in the presence of keratin.

ORS cells formed colonies within a few hours of exposure to keratin, and subsequently formed strand-like extended structures by day 3 (Fig. 3a, b). The proliferation of ORS cells was suppressed upon exposure to keratin, as indicated by decreased BrdU incorporation (Supplementary Fig. 14). High β-catenin expression, known to occur during migration, further differentiation of stem cells in the ORS region[6], and a local cell population expressing P-cadherin, known to be a marker of secondary HG formation[20,21], were observed along with extended structures in keratin-treated ORS cells (Fig. 3c, Supplementary Fig. 15). In addition, keratin-treated ORS cells showed lower expression levels of CD34 than untreated ORS cells but maintained high levels of SOX9 expression (Fig. 3c). With reduced expression of CD34, the Lgr5+ cell population, which is known to participate in hair germ formation[22–24], emerged in keratin-treated ORS cells (Supplementary Fig. 16). In addition, real-time qPCR analysis showed upregulation of the expression of various genes, such as

*SOX9, EDAR, FOXN1*[25], *MSX2*[26], *SHH*[27], *EFNB1, ITGA6*[28] and *β-catenin*, indicating matrix and shaft differentiation of stem cells from the ORS bulge region (Supplementary Fig. 17). Furthermore, RNA sequencing analysis of keratin-treated ORS cells revealed upregulation of mRNA expression levels of acidic hair keratins, mainly *KRT31, KRT33B, KRT34*, and *KRT37* (Supplementary Fig. 18a). We also observed increased expression of KRT34 and β-catenin proteins (Supplementary Fig. 18b). These findings imply that hair keratin-mediated alterations in the protein and gene expression profiles indicate germ formation and further differentiation of ORS cells.

**TGFβ2 induces apoptosis of ORS cells, keratin release and deposition, mediating condensation of DP cells**. The findings described above demonstrated that the exposure to keratin induced the condensation of DP cells and the formation of P-cadherin-expressing germs of ORS cells, which led us to ask whether the observed interaction of DP and ORS cells with keratin might be related to a biological event that occurs during hair cycling. During the anagen-catagen transition stage, TGFβ2 is

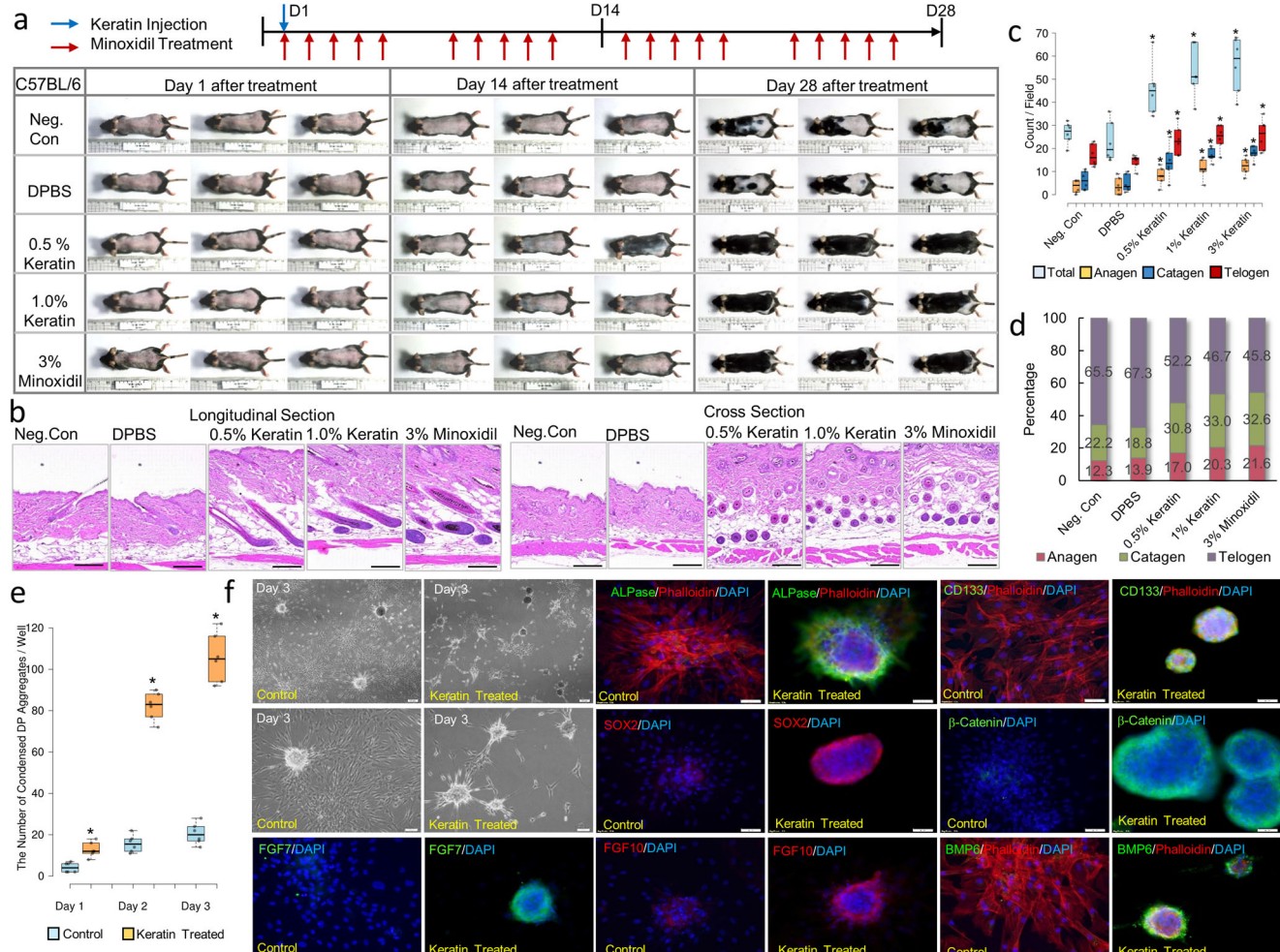

**Fig. 1 Intradermal injection of human hair-derived keratin with different concentrations induced hair follicle formation and improved hair growth. a** Images of hair growth on the back skin of mice at day 1, day 14, and day 28 after injection of keratin. **b** Histological images of the back skin of mice at 4 weeks after injection of keratin. **c, d** Graphical representation and quantification of hair follicles with different hair cycle stages in skin sections of mice (*n* = 12 sections, in six mice; mean ± standard deviation (s.d.)). *P,0.05, indicates a difference between control and keratin-injected groups. Scale bars, 200 μm. **e** Graphical representation of DP cell condensation in the presence of keratin. *P,0.01, indicates a difference between control and keratin-treated. (*n* = 6; mean ± standard deviation (s.d.)); Control, non-treated DP cell; keratin-treated, DP cells in the presence of keratin. **f** Condensation of DP cell by immunofluorescent staining in the presence of keratin; 4′,6-diamidino-2-phenylindole (DAPI), blue; SOX2, alkaline phosphatase (ALPase), FGF10, BMP6, red; β-catenin, CD133, and FGF7, green. Scale bars, 100 μm.

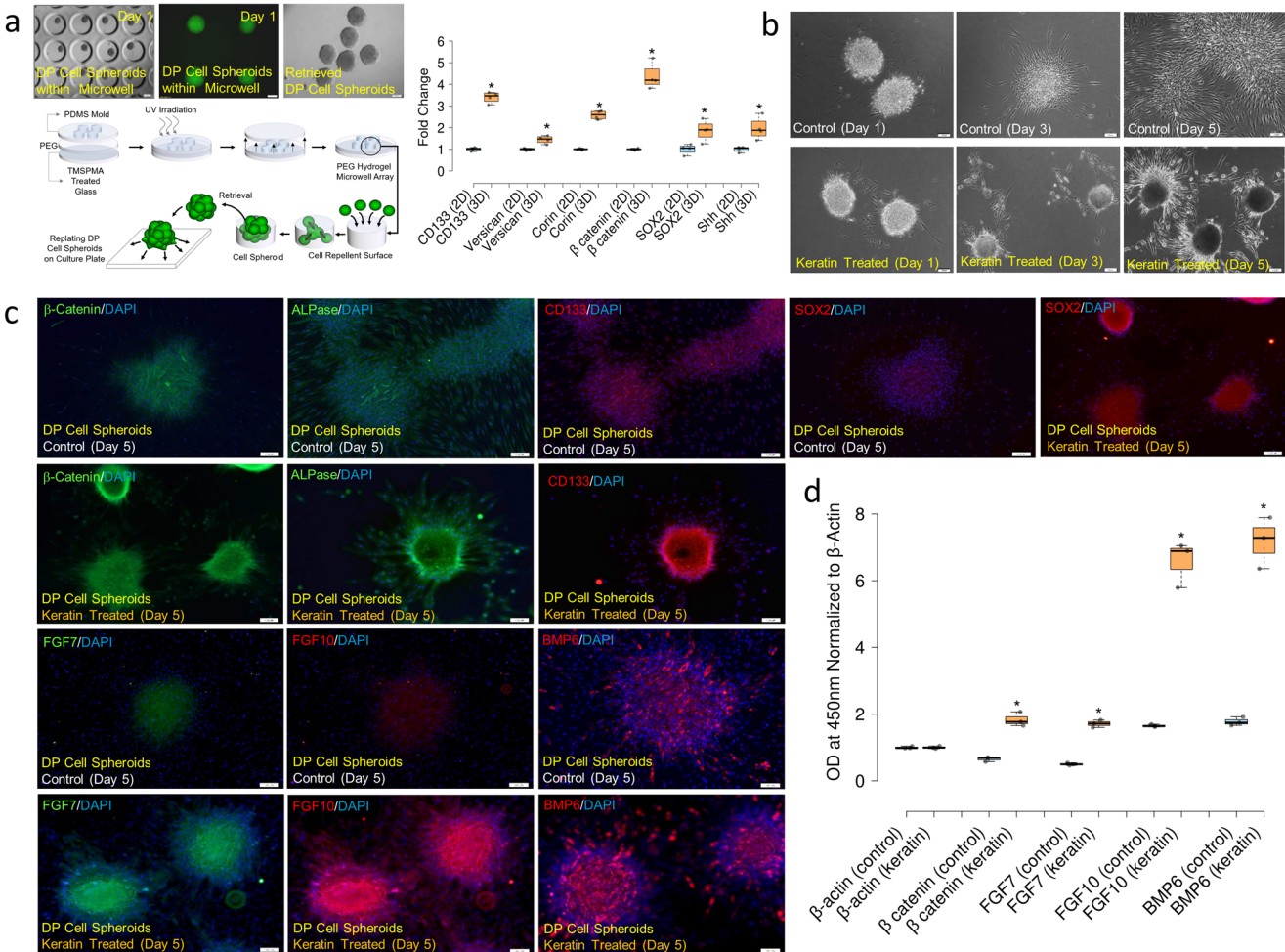

**Fig. 2 Hair keratin treatment retained DP cell condensation with upregulated expressions of DP cell property-related molecular markers. a** Schematic illustration of microwell-mediated DP cell spheroid formation and images of DP cell spheroids within microwells and retrieved DP cell spheroids from microwells. Graphical representation of DP cell property-related gene expressions of DP cells and DP cell spheroids. *$P$,0.01, indicates a difference between DP cells and DP cell spheroids; 2D, DP cells; 3D, DP cell spheroids. Scale bars, 100 μm. ($n = 4$; mean ± standard deviation (s.d.)). **b** Images of replated DP cell spheroids in the presence of keratin by observation using a light microscope. Scale bars, 100 μm. **c** Images of replated DP cell spheroids in the presence of keratin by immunofluorescent staining; DAPI, blue; SOX2, CD133, FGF10, BMP6, red; β-catenin, ALPase, FGF7, green. Scale bars, 100 μm. **d** Graphical representation of DP cell property-related molecular expressions of replated DP cell spheroid culture in the presence of keratin; ELISA. *$P$,0.01, indicates a difference between control and keratin-treated. ($n = 3$; mean ± standard deviation (s.d.)).

synthesized by DP cells stimulated by dihydrotestosterone and is spatiotemporally localized in the lower part of the hair bulb at the catagen stage, thus suppressing the proliferation of epithelial cells but inducing caspase-mediated apoptosis[11]. Therefore, we hypothesized that exposure to keratins derived from apoptotic ORS cells during hair cycling might drive DP cell condensation and secondary HG formation through interaction with DP and ORS cells. To address this question, we induced apoptosis of ORS cells by treatment with TGFβ2 and characterized microenvironmental changes, such as the release or deposition of keratin. Apoptosis array analysis showed the upregulation of apoptosis-related markers, including Bax, caspase-3, cytochrome C, and SMAC, in ORS cells treated with TGFβ2 (Fig. 4a, Supplementary Fig. 19). Extended structures composed of spindle-shaped ORS cells developed only in the presence of TGFβ2; annexin V-and TUNEL-positive apoptotic cells were mainly found in the extended structures of TGFβ2-treated ORS cells, and elevated expression levels of caspase-3 and massive deposition of keratin were observed along the extended structures (Fig. 4b), which was confirmed by western blot analysis (Supplementary Fig. 20). To determine whether the released or deposited keratin derived from

TGFβ2-induced apoptotic ORS cells could influence DP cell condensation, the condensation activity of DP cells was tested by direct contact co-culture and culture in conditioned medium. Local condensation of DP cells with the formation of spherical cell colonies was observed in the concentric region of the extended structures in the TGFβ2-treated ORS cell layers (Fig. 4c, Supplementary Fig. 21a). The conditioned medium collected from TGFβ2-treated ORS cell cultures contained relatively higher levels of keratin, and DP cell condensation into spherical cell aggregates was distinctly increased following the culture of DP cells in the conditioned media (Fig. 4d, Supplementary Fig. 21b). These results indicate that the deposition or release of keratin from TGFβ2-induced apoptotic ORS cells could regulate the induction of DP cell condensation.

**Keratin release and deposition through caspase-6-mediated keratin degradation stimulates condensation of DP cells.** Keratin fragmentation occurs during apoptosis of epithelial cells[29]; intracellular insoluble keratin is fragmented by caspases during apoptosis and released as soluble fragments[30]. In our study,

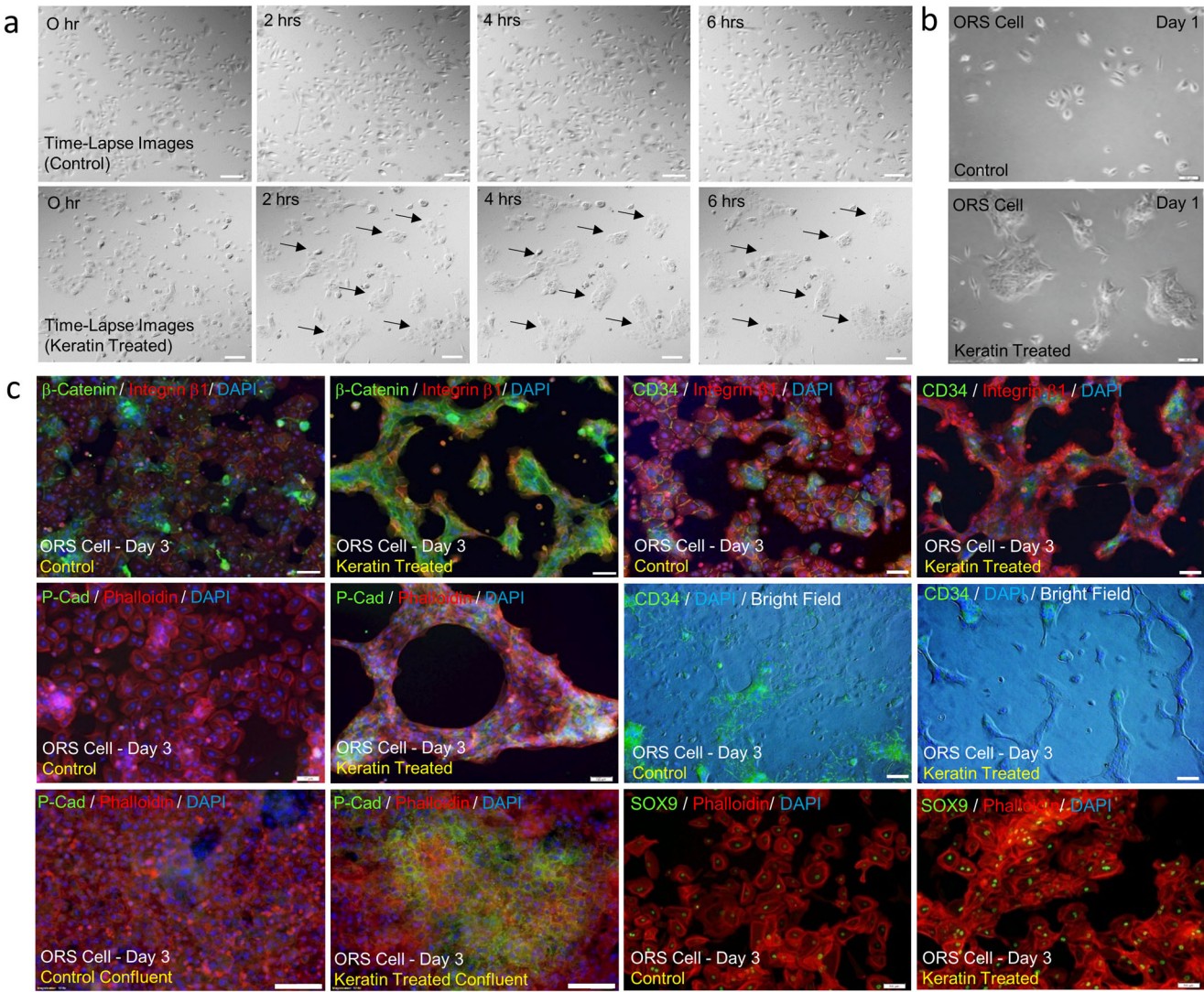

**Fig. 3 Hair-derived keratin-induced colony formation and P-cadherin expressing germ formation of ORS cells. a** Time-lapse images of ORS cells in the presence of keratin; black arrows indicate colony formation. Scale bars, 100 μm. **b** Image of ORS cells and quantification of ORS cell growth in the presence of keratin. *$P$,0.01, indicates a difference between control and keratin-treated. Scale bars, 100 μm. ($n = 6$; mean ± standard deviation (s.d.)); ORS outer root sheath. **c** In vitro germ formation of ORS cell by immunofluorescent staining in the presence of keratin; DAPI, blue; phalloidin, integrin β1, red; P-cadherin, CD34, β-catenin, SOX9, green. Control confluent, ORS cell culture at confluent cell density under ORS culture medium; keratin-treated confluent, ORS cell culture at confluent cell density in the presence of keratin. Scale bars, 100 μm.

apoptosis array analysis showed a two-fold increase in caspase-3 expression levels in TGFβ2-treated ORS cells, and another study reported that type I keratin, including hair keratin, contains a cleavage site, VEVD, for caspase-6[31]. When hair keratin was digested with caspase-3 and caspase-6, keratin fragments were generated only in hair keratin digested by caspase-6 (Fig. 5a). Furthermore, higher gene and protein levels of caspase-6 and its cleaved form (active caspase-6) were observed in TGFβ2-treated ORS cells (Fig. 5b, c). These findings imply that the release or deposition of keratin derived from TGFβ2-treated ORS cells through caspase-6-mediated proteolysis can influence DP cell condensation. To test this, we silenced the expression of caspase-6 in ORS cells by siRNA transfection and examined the levels of released keratin (Fig. 5d). We observed lower levels of released keratin in caspase-6-knockdown ORS cells in the presence of TGFβ2 (Fig. 5e), and lower condensation activity of DP cells cultured in conditioned media collected from caspase-6-knockdown ORS cell cultures or co-cultured on the caspase-6-knockdown ORS cell layer in the presence of TGFβ2 (Fig. 5f–h).

Caspase-6-knockdown ORS cells developed relatively spread structures even in the presence of TGFβ2, which were different from the extended structures of TGFβ2-treated non-knockdown ORS cells. TUNEL-positive apoptotic cells were found in both the spread and extended structures, and keratin deposition was influenced by caspase-6 expression levels (Fig. 5h, Supplementary Fig. 22). We then immunodepleted keratin from the conditioned medium of TGFβ2-treated ORS cell cultures using a column containing anti-human type I + II hair keratin antibody-conjugated beads assay and examined its effect on the condensation of DP cells. The elimination of keratins in the conditioned media was confirmed (Fig. 6a), and there was no substantial difference in the levels of growth factors contained in the conditioned media before and after immunodepletion (Fig. 6b, Supplementary Fig. 23). The increase in TGFβ2 in the conditioned medium was derived from the exogenous TGFβ2 added to induce apoptosis of ORS cells, and TGFβ2 did not influence DP cell condensation by itself (Supplementary Fig. 24). The immunodepletion assay showed that the removal of keratins

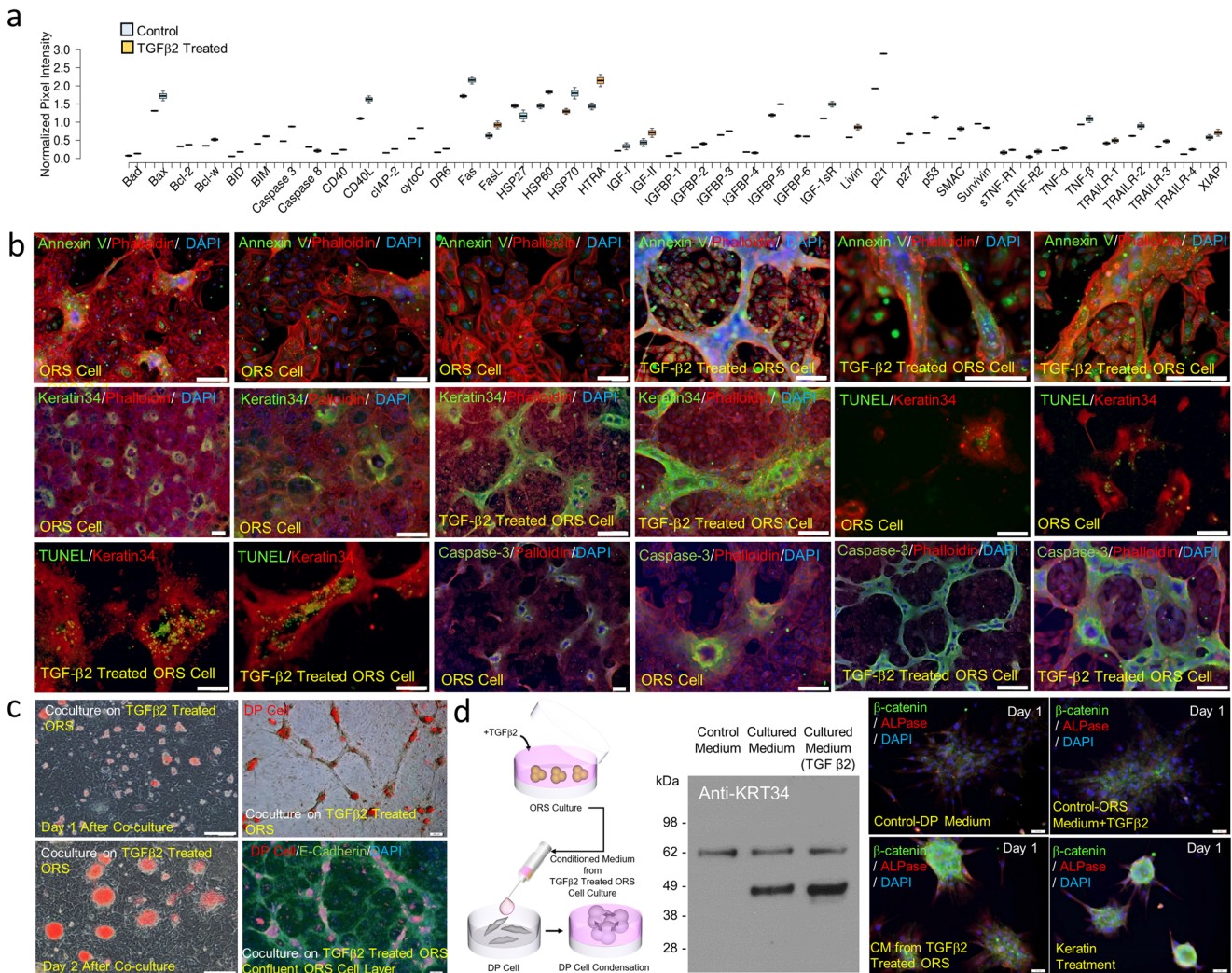

**Fig. 4 TGF-β2-induced apoptosis of ORS cells, keratin exposure was generated from TGFβ2-induced apoptotic ORS cells, and keratin release from TGFβ2-mediated apoptotic ORS cells induced DP cell condensation. a** Graphical quantification of apoptosis array of ORS cells and TGFβ2-treated ORS cells. **b** TGFβ2-induced apoptosis and its following keratin exposure of ORS cells by immunofluorescent staining; DAPI, blue; phalloidin, keratin 34, red; annexin V, keratin 34, TUNEL, caspase 3, green. Scale bars, 200 μm. **c** Images of DP cell condensation on TGFβ2-treated ORS cell layers. Co-culture of cell tracker-treated DP cells (red) on TGFβ2-treated ORS cell layers. Immunofluorescent image; E-cadherin, green; DAPI, blue. Scale bars, 200 μm. **d** DP cell condensation under conditioned medium collected from TGFβ2-treated ORS cell culture. Western blot image of released keratin 34 from ORS cell culture and TGFβ2-treated ORS cell culture. Immunofluorescent image; ALPase, red; β-catenin, green; DAPI, blue; Control-DP medium, DP culture medium; Control-ORS Medium-TGFβ2, ORS medium including TGFβ2; CM from TGFβ2-treated ORS, conditioned medium collected from TGFβ2-treated ORS cell culture; Keratin treatment, DP medium containing 1(w/v)% keratin. Scale bars, 50 μm.

suppressed DP cell condensation (Fig. 6c), which reflects the functional role of keratins in DP condensation.

**Spatial deposition of keratin derived from TGFβ2-induced apoptotic ORS cells induces germ formation**. In addition to keratin-mediated condensation of DP cells, the ability of keratin released from TGFβ2-induced ORS cells to induce germ formation was tested Contrary to DP cell condensation, the keratin released in conditioned media was not effective in generating a P-cadherin-expressing cell population, which was confirmed by an immunodepletion assay (Supplementary Fig. 25). This result prompted us to ask whether secondary HG formation by cells expressing P-cadherin is influenced by spatially deposited keratin, which is caused by the spatiotemporal apoptosis of ORS cells. The expression of TGFβ2, which is restricted to the outermost ORS cell layer in the anagen phase, has been reported to be upregulated spatiotemporally in the boundary region between germinal

matrix cells and DP cells in the lower bulb region during late anagen and catagen[8]. Therefore, to study the spatial deposition of keratin derived from TGFβ2-induced apoptotic ORS cells and its effect on germ formation by ORS cells, the time-course effect of TGFβ2 treatment on expression levels of caspase-6 protein, keratin deposition, and germ formation was characterized. The extended structures were progressively developed in the TGFβ2-treated ORS cell layers over the cultivation time, and the P-cadherin-expressing germ was spatially developed in the TGFβ2-treated ORS cell layers (Supplementary Fig. 26). In addition, immunocytochemical staining of the TGFβ2-treated ORS cell layers showed that a population of RUNX1-and P-cadherin-positive cells, representative markers of germ formation[10,32–34], emerged in the concentric region of the extended structures, and the caspase-6-expressing apoptotic cell population and the keratin-deposited area also increased over time in the extended structures (Supplementary Fig. 26a), and such highly intensified staining was confirmed by flow cytometric

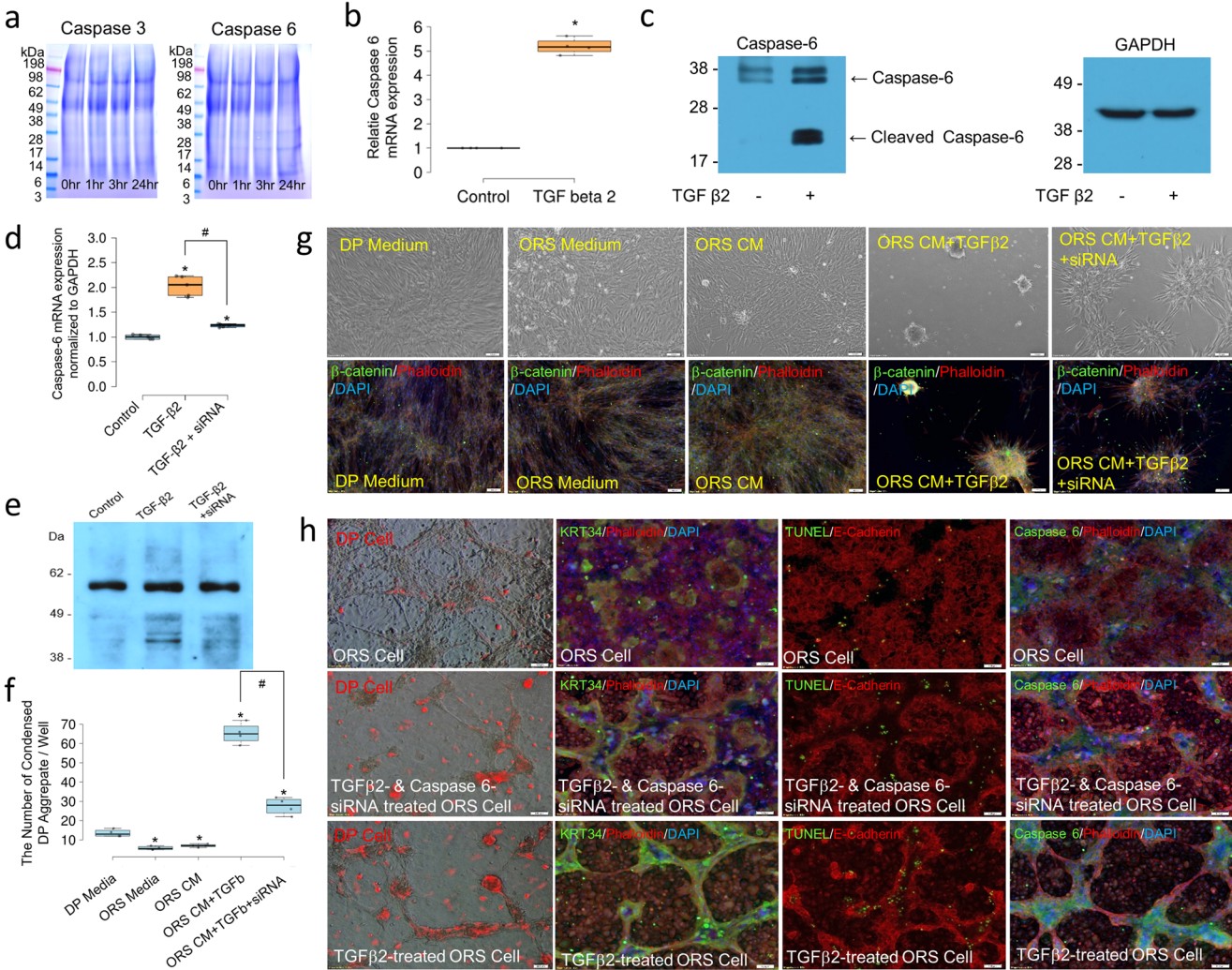

**Fig. 5 Caspase 6-mediated keratin degradation and DP cell condensation activity in conditioned medium collected from caspase 6-knockdown ORS cell culture in the presence of TGFβ2. a** SDS-PAGE images of caspase 3- and caspase 6-mediated keratin degradation. **b** Caspase 6 mRNA expression in TGFβ2-treated ORS cells by real-time qPCR. P,0.01, indicates a difference between ORS cells (control) and TGFβ2-treated ORS cells (TGFβ2). $n = 4$; mean ± standard deviation (s.d.)). **c** Western blot images of caspase 6 expression in TGFβ2-treated ORS cells. **d** Caspase 6 mRNA expression in ORS cell culture in the presence of TGFβ2 and caspase 6-knockdown ORS cell culture in the presence of TGFβ2 by real-time-qPCR; Control, ORS cells; TGFβ2, ORS cell culture in the presence of TGFβ2; TGFβ2+siRNA, caspase 6-knockdown ORS cell culture in the presence of TGFβ2. *P,0.01, indicates a difference between control and experimental group. #P,0.01, indicates a difference between ORS cell culture in the presence of TGFβ2 and caspase 6-knockdown ORS cell culture in the presence of TGFβ2. $n = 5$; mean ± standard deviation (s.d.)). **e** Western blot image of keratin in conditioned medium collected ORS cell culture in the presence of TGFβ2 and caspase 6-knockdown ORS cell culture in the presence of TGFβ2. **f** DP cell condensation activity in conditioned medium collected from caspase 6-knockdown ORS cell culture in the presence of TGFβ2; DP medium, basic DP medium; ORS Medium; basic ORS medium; ORS CM, conditioned medium collected from ORS cell culture; ORS CM + TGFβ2, conditioned medium collected from ORS cell culture in the presence of TGFβ2, ORS CM + TGFβ2+siRNA, conditioned medium collected from caspase 6-knockdown ORS cell culture in the presence of TGFβ2. *P,0.01, indicates a difference between DP cell culture in DP medium and DP cell culture in other culture media and conditioned media. #P,0.01, indicates a difference between DP cell culture in conditioned medium collected from ORS cell culture in the presence of TGFβ2 and DP cell culture in conditioned medium collected from caspase 6-knockdown ORS cell culture in the presence of TGFβ2. $n = 4$; mean ± standard deviation (s.d.)). **g** Images of DP cell condensation in conditioned medium collected from caspase 6-knockdown ORS cell culture in the presence of TGFβ2, and DP cell condensation by immunofluorescent staining; phalloidin, red; β-catenin, green; DAPI, blue; Scale bars, 100 μm. **h** Images of DP cell condensation on caspase 6-knockdown ORS cell layer in the presence of TGFβ2. Co-culture of cell tracker-treated DP cells (red) with TGFβ2-treated ORS cell layers. Scale bars, 200 μm. Immunofluorescent image of caspase 6-knockdown ORS cells in the presence of TGFβ2: Phalloidin, E-cadherin, red; KRT34, TUNEL, caspase 6, green; DAPI, blue; Scale bars, 100 μm.

analysis (Supplementary Fig. 26b). Next, to consider the effect of spatial deposition of keratin on the formation of P-cadherin-expressing germs in vitro, the expression of KRT31/KRT34 in ORS cells was downregulated by siRNA transfection. Downregulation of the expression of keratin in both conditioned media and ORS cells after KRT31/KRT34 downregulation was confirmed (Fig. 6d). In contrast to the well-developed stranded structures in the ORS cell layers, the KRT31/KRT34- knockdown

ORS cells did not form extended structures even in the presence of TGFβ2, and keratin deposition and the emergence of RUNX1- and P-cadherin-expressing ORS cell populations were markedly suppressed in the KRT31/KRT34-knockdown ORS cell culture in the presence of TGFβ2 (Fig. 6d). KRT31/KRT34 downregulation did not influence cell growth or the generation of Lgr5+ and P-cadherin+ cell populations in ORS cells (Supplementary Figs. 27 and 28).

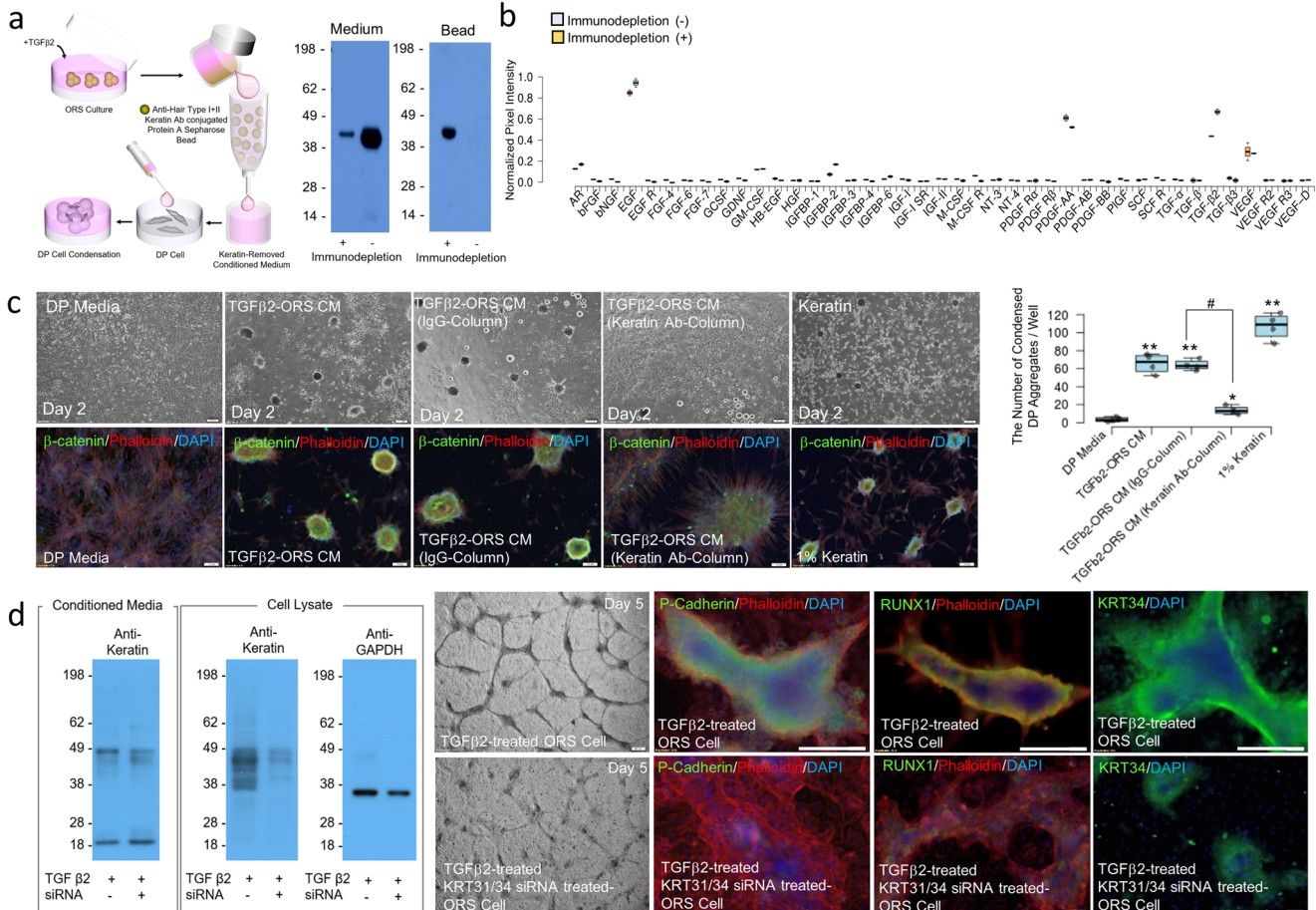

**Fig. 6 DP cell condensation activity in keratin-eliminated conditioned medium of TGFβ2-treated ORS cell culture using a column containing anti-human type I + II hair keratin antibody-conjugated beads, and germ formation in ORS cell culture and KRT31/KRT34 knockdown ORS cell culture in the presence of TGF-β2. a** Western blot image of the keratin-eliminated conditioned medium and the keratin-bound beads. **b** Graphical quantification of growth factor content of the conditioned medium and the keratin-eliminated conditioned medium collected from TGFβ2-treated ORS cell culture using antibody array for growth factors. **c** DP cell condensation activity in keratin-eliminated conditioned medium of TGFβ2-treated ORS cell culture using a column containing anti-human type I + II hair keratin antibody-conjugated beads. Immunofluorescent image; phalloidin, red; β-catenin, green; DAPI, blue; DP media, DP culture medium; TGFβ2-ORS CM, conditioned medium collected from TGFβ2-treated ORS cell culture; TGFβ2-ORS CM (IgG-column), conditioned medium collected from TGFβ2-treated ORS cell culture and then treated with normal IgG-conjugated beads; 1% Keratin, DP medium containing 1(w/v)% keratin; TGFβ2-ORS CM (keratin Ab-column), keratin-eliminated conditioned medium collected from TGFβ2-treated ORS cell culture. Graphical representation of DP cell condensation activity. *P,0.05 and **P,0.01, indicate a difference between DP media and experimental groups. #P,0.01, indicates a difference between TGFβ2-ORS CM (IgG-column) and TGFβ2-ORS CM (keratin Ab-column). Scale bars, 100 μm. (n = 4; mean ± standard deviation (s.d.)). **d** Germ formation of ORS cells and KRT31/KRT34 knockdown ORS cells. Western blot image of keratin content of conditioned media and cell lysates from negative control siRNA-transfected ORS cell culture in the presence of TGFβ2 and KRT31/KRT34 knockdown ORS cells culture in the presence of TGFβ2. Image of ORS cells and KRT31/KRT34 knockdown ORS cells in the presence of TGFβ2. Germ formation of ORS cells and KRT31/KRT34 knockdown ORS cells in the presence of TGFβ2 by immunofluorescent staining; phalloidin, red; P-cadherin, RUNX1, KRT34, green; DAPI, blue. Scale bars, 200 μm.

**In vivo knockdown of keratin expression suppresses anagen hair follicle formation and hair growth**. The in vitro data from studies on the interaction of keratin with DP and ORS cells and the release and deposition of keratin from TGFβ2-induced apoptotic ORS cells have shown a pivotal role of keratin in controlling DP condensation and HG formation. Finally, to determine whether downregulation of KRT31/KRT34 expression can suppress hair follicle formation and hair growth in vivo, keratin expression in mice was temporarily downregulated by intravascular lipofectamine-mediated delivery of KRT31/KRT34 siRNAs. RT-PCR analysis showed effective KRT31/KRT34 downregulation of KRT31/KRT34 mRNA expression on day 7 (Fig. 7a). Furthermore, it was found that KRT31/KRT34 downregulation inhibited hair growth compared to the control (Fig. 7b). Notably, dysregulation of hair follicle cycling was

observed following downregulation of KRT31/KRT34 in mice; and histological analysis of hair follicle sections showed a strong suppression of the formation of anagen follicles, with no appearance of anagen follicles in 56% of skin tissue sections following KRT31/KRT34 downregulation in mice on day 7 (Fig. 7c). In addition, the formation of catagen follicles was relatively reduced on day 7 upon downregulation of KRT31/KRT34 in mice with or without exogenous keratin injection (Fig. 7c). An anagen bulb containing a population of cells expressing P-cadherin was hardly seen in immunohistological sections of KRT31/KRT34 knockdown mice (Fig. 7d, Supplementary Fig. 29). Emergence of Lgr5-positive cell population and molecular expression of KRT34 were also distinctly decreased in KRT31/KRT34 knockdown mice on day 7 (Supplementary Figs. 30 and 31). In contrast, relatively higher *Lgr5*-positive staining was observed in exogenous keratin-

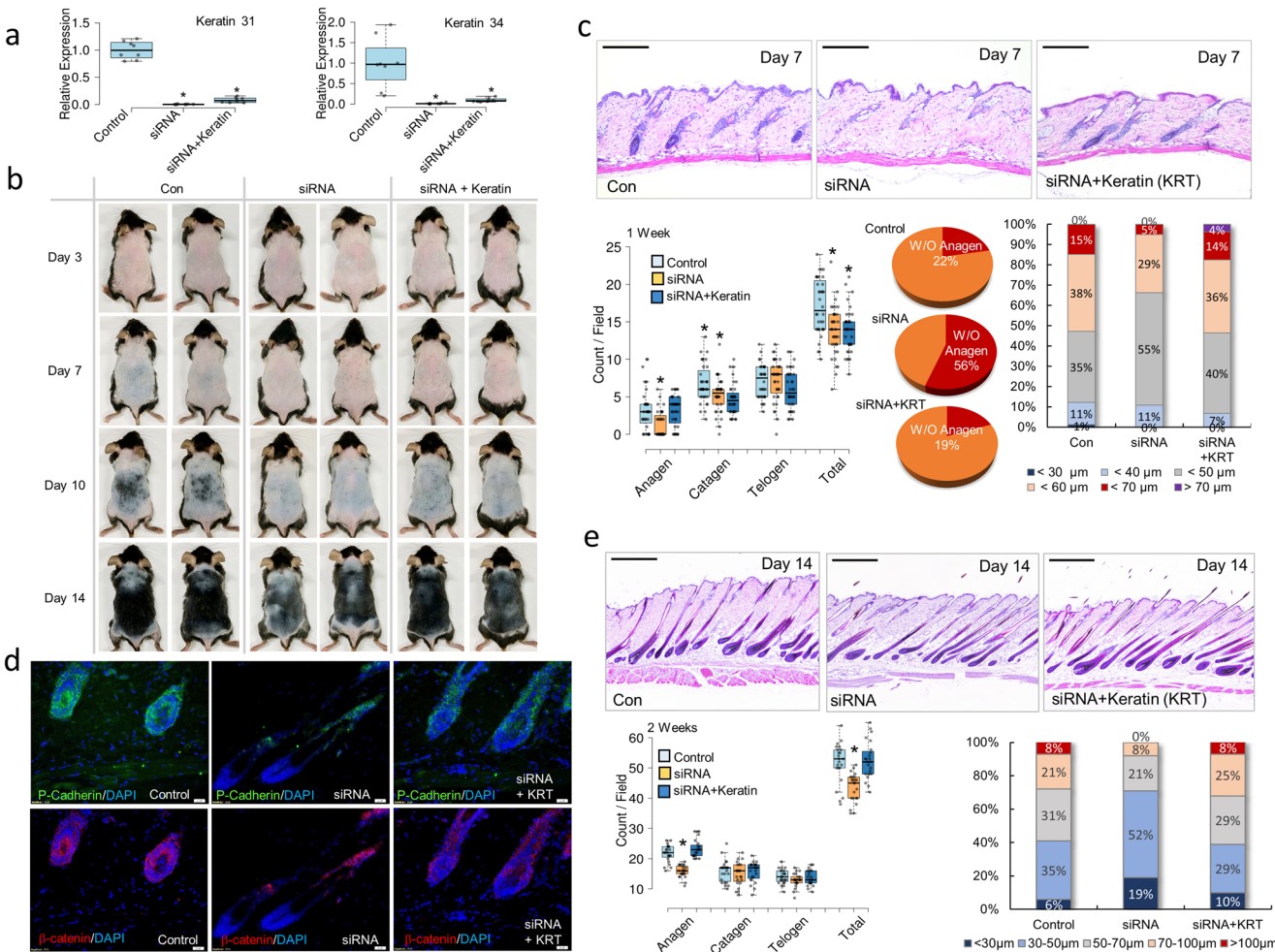

**Fig. 7 Temporal knockdown of KRT31/KRT34 expressions in vivo suppressed hair growth. a** Graphical representation of KRT31 and KRT34 mRNA expressions in mice on day 7 after KRT31/KRT34 knockdown. Control, mice injected with negative control siRNA-loaded lipofectamine; siRNA, KRT31/KRT34 knockdown mice; siRNA+KRT, KRT31/KRT34 knockdown, and hair keratin-injected mice; *P,0.05, indicates a difference between control and experimental groups. (n = 4, in 4 mice; mean ± standard deviation (s.d.)). **b** Images of hair growth on the back skin of mice on day 3, day 7, day 10, and day 14 after KRT31/KRT34 siRNA transfection. **c** Histological images of the back skin of mice on day 7 after KRT31/KRT34 siRNA transfection and intradermal injection of keratin. Graphical representation and quantification of hair follicle formation in skin sections of mice (n = 36 sections, in 8 mice; mean ± standard deviation (s.d.)). *P,0.01, indicates a difference between control group and experimental groups. Scale bars, 200 μm. **d** Immunohistochemical images of the back skin of mice on day 7 after KRT31/KRT34 siRNA transfection and intradermal injection of keratin; β-catenin, red; P-cadherin, green; DAPI, blue. Scale bars, 20 μm. **e** Histological images of the back skin of mice on day 14 after KRT31/KRT34 siRNA transfection and intradermal injection of keratin. Graphical representation and quantification of hair follicle formation in skin sections of mice (n = 23 sections, in five mice; mean ± standard deviation (s.d.)). *P,0.01, indicates a difference between control group and experimental groups. Scale bars, 20 μm.

injected KRT31/KRT34 knockdown mice on day 7 in comparison with KRT31/KRT34 knockdown mice (Supplementary Fig. 30), and an additional injection of hair-derived keratin after KRT31/KRT34 siRNA transfection allowed the hair follicles to enter the anagen phase and regrow hair, similar to the controls. No obvious histological differences were found in the formation of hair follicles and hair growth between the control skin and keratin-injected skin of KRT31/KRT34 knockdown mice after 2 weeks (Fig. 7e). Furthermore, the formation of P-cadherin-positive germs and strong expression of β-catenin were observed in the region of anagen hair follicles in sections of control skin and keratin-injected skin of KRT31/KRT34 knockdown mice (Fig. 7d, Supplementary Fig. 32), and strong staining for KRT34 was found in the ORS region surrounding the DP, which corresponds to the caspase-6-positive region (Supplementary Fig. 32). Interestingly, it was found that the region stained positive for caspase-6, KRT34, and P-cadherin moved upward into the hair shaft region of the expanded hair follicles (Supplementary Fig. 32).

## Discussion
Our study shows that intradermal injection of human hair-derived keratin promotes hair growth with enhanced formation of anagen hair follicles and an increase in the size of hair follicles. Hair growth is controlled by the interactions between two distinct cell types: mesenchyme and epithelial cells, while DP and stem cells from the bulge region of the ORS participate in hair follicle formation[4–6,15]. The in vivo injection of keratin-induced follicle formation and hair growth. This effect could be associated with keratin-mediated DP cell condensation and germ formation via its interaction with cells, as evidenced by the strong expression of various signature molecules, such as β-catenin and P-cadherin, which are highly expressed during hair follicle formation[4–6,15], in keratin-treated DP ORS cell cultures. Secondary HG expressing P-cadherin emerges at the telogen stage and leads to the first stage of hair regeneration[32]. The interaction of P-cadherin with β-catenin plays an important role in maintaining the anagen phase of the hair cycle[35], and β-catenin-expressing cells migrated from

the bulge region of the ORS undergo further differentiation in the follicle region[6]. Keratin-treated ORS cells showed morphological changes, such as spindle shape, strong expression of β-catenin, and reduced expression of CD34, also a population of P-cadherin-expressing cells emerged. With the loss of CD34 expression, an Lgr5-expressing cell population was generated in keratin-treated ORS cells. CD34-positive stem cells have been reported to convert directly into P-cadherin-expressing HG cells[33], and CD34-positive stem cells migrate downward to the lower bulge region and convert to Lgr5-positive stem cells, which participate in HG formation during the telogen stage[24]. In addition to keratin-mediated ORS cell differentiation, treatment with exogenous keratin induces DP cell condensation. Epithelial fibroblast growth factor 20 (Fgf20) is known to control dermal condensate morphogenesis[36]; hence, we evaluated the presence of potent growth factors, such as FGF20, in hair-derived keratin to influence cellular behavior. Western blot analysis showed no presence of FGF20 in the keratin extract used in this study, and application of MALDI-TOF mass spectroscopy indicated that the extracted protein was purely keratin (Supplementary Fig. 33). Along with their differentiation, growth of ORS cells was stopped in the presence of keratin, but CCK-8 analysis, which measures mitochondrial dehydrogenase activity, showed a temporal increase in growth on day 1 of keratin treatment (Supplementary Fig. 14). This change in cellular metabolic activity by keratin treatment needs to be studied further. Hair regeneration is processed by hair follicles undergoing repeated cycles of anagen (hair growth stage), catagen (regression stage), and telogen (relative rest stage)[37]. During the telogen phase, secondary HG progressively appears at the base of the follicular epithelium. At that point, HG cells form a cell cluster and are activated to begin hair regeneration[10], and DP cells undergo condensation to form a follicular papilla beneath the secondary HG. The interaction between the DP condensate and secondary HG leads to the formation of new hair follicles by enveloping the DP with downwardly extended epithelial cells[10,32,38]. However, despite considerable progress in understanding the cellular interactions that control hair growth, it is not clear how secondary HG formation and DP condensation, the key biological events causing hair regeneration, are initiated at the beginning of a new hair cycle.

Hair growth in keratin-injected mice, keratin-mediated condensation of DP cells, and the formation of P-cadherin expressing germs, assessed by in vitro cell study, led us to explore the biological function of keratin in hair regeneration as a pilot study. At the anagen-catagen transition stages of the hair cycle, the local deposition of TGFβ2 in the lower region of the follicle is restricted due to the spatiotemporal secretion of TGFβ2 produced by DP cells[11], which is consistent with the spatial gradient of apoptosis of epithelial cells[9]. Spatiotemporally localized TGFβ2 induces apoptosis of ORS cells in the lower part of the hair bulb at the catagen stage, resulting in the expression of caspase and its-mediated fragmentation of insoluble keratin into soluble keratin fragments[8–11,29]. Although the relationship between TGFβ2 expression and the intrinsic property of DP cells related to condensation is not well known, our data showed that TGFβ2 expression was downregulated during DP cell condensation and rapidly upregulated during the dispersion of condensed DP cells (Supplementary Fig. 34a, b). Mesenchyme condensation such as dermal condensates is promoted by BMP signaling and transient downregulation of TGF-β signaling, showing an antagonistic relationship[18,39,40]. The condensed DP cells might maintain the property of DP cells[38] to drive hair growth by releasing paracrine factors, which can induce stem cell activation and differentiation during the anagen phase, and may induce spatiotemporal apoptosis of adjacent ORS cells via increased expression of TGFβ2

during the anagen to catagen transition. In our study, it was shown that local DP cell condensation and germ formation in TGFβ2-induced apoptotic ORS cells depend on the exposure of keratin via caspase-6 expression and consequently its-mediated keratin exposure from apoptotic cell death, as evidenced by the suppressed DP cell condensation and germ formation in caspase-6-knockdown or KRT31/KRT34-knockdown ORS cell culture, even if there was no distinct difference in TGFβ2-mediated ORS cell apoptosis. From these in vitro findings, it could be inferred that spatially increased keratin exposure, following gradual apoptosis during the regression stage, might provide a cue to derive germ formation and new hair cycle initiation from telogen to anagen, which spatial keratin deposit could be identified at upper void space unoccupied cells in the newly formed hair follicle (Supplementary Fig. 35) In addition, apoptosis-related keratin exposure also could be identified by similar spatial expressions of apoptosis-markers such as Annexin V and active caspase 3 and KRT 34 in the developing hair follicles containing proliferating cells (Supplementary Fig. 36).

Finally, to determine the biological function of keratin in vivo, the effect of downregulating KRT31/KRT34 mRNA expression on hair growth in mice was evaluated. Exogenous keratin injection in KRT31/34 knockdown mice resulted in a relatively reduced formation of catagen follicles on day 7. This might be due to the temporal inhibition of keratin expression accompanied by stem cell differentiation into the matrix and shaft during the anagen phase, which might influence catagen formation. In addition, the formation of anagen hair follicles and hair growth were suppressed in mice with temporal downregulation of KRT31/KRT34, which could be recovered by intradermal injection of additional exogenous keratin. With the poor formation of anagen follicles in KRT31/KRT34 knockdown mice, P-cadherin and Lgr5-expressing cell population was scarcely observed in telogen follicles in these mice (Supplementary Figs. 29 and 30). In vitro KRT31/34 knockdown ORS cells did not show any change of ability for cell growth and differentiation into P-cadherin and Lgr5-expressing cells (Supplementary Figs. 27 and 28), and such P-cadherin and Lgr5-expressing cell population could be formed by injecting exogenous keratin into KRT31/34 knockdown mice even though less than control mice (Supplementary Figs. 29 and 30). These findings indicate that alteration of hair keratin gene expression might influence hair growth following stem cell differentiation into the matrix and shaft, and hair cycle transitions might be controlled by keratin-mediated microenvironmental change. However, the reduced anagen follicle formation could not be explained in KRT31/34 knockdown mice model starting at synchronized telogen phase, which hair cycle-dependent keratin expression and apoptosis-related keratin exposure even during telogen to anagen transition need to be studied further. The expression of hair keratins is not restricted to the anagen phase, showing cell growth and differentiation-mediated hair keratin production, and are found at all stages of the hair cycle[41–43]. A distinct epithelial cell population expressing Bcl-2 in secondary HG and DP was found during the telogen-anagen transition, which shows differential susceptibility to apoptosis[44]. Such programmed cell death-related cellular processes were also detected during telogen, and the stimulation of autophagy following programmed cell death initiated the telogen-anagen transition[45]. In addition, A study using transgenic mice overexpressing an anti-apoptotic gene reported that the inhibition of the apoptotic death of ORS cells even during anagen resulted in the early termination of hair follicle stem cell activation and proliferation, whereas the initiation of a new hair cycle was postponed by inhibiting the apoptotic death of ORS cells during telogen[46]. These studies suggested that telogen might not be the only resting phase in hair growth, but also an activating phase, including DP condensation

and secondary HG formation[47,48]. These studies indicate that spatiotemporal apoptosis during the hair cycle can be an essential process in controlling hair regeneration, and our findings show that keratin can be an important factor influencing hair growth. However, these pilot observations require follow-up studies of hair keratin expression and its apoptosis-related exposure during the hair cycle using genetically modified ORS cells equipped with an on-off expression system for keratin expression and a xenograft mouse model to determine the in vivo mechanisms.

Despite in vitro and in vivo studies of keratin-mediated hair growth, the mechanism by which keratin induces DP condensation and hair germ formation remain unclear. In this study, DP cell condensation was also induced on Matrigel (Supplementary Fig. 9), and we found a decrease in the hardness of Matrigel in the presence of keratin. Hence, we tested the keratin-mediated change in the hardness of Matrigel, and keratin treatment resulted in a partial disintegration of Matrigel (Supplementary Fig. 37), which might influence cell and matrix interactions. In addition, loss of vinculin, which participates in local cell adhesion, was found in keratin-mediated DP cell condensation (Supplementary Fig. 38), and highly decreased expression of vinculin was observed in keratin-treated ORS cells (Supplementary Fig. 39). A recent report showed that mechanical instability of cells to ECM contact was a factor in controlling activation of hair follicle stem cells in the bulge region, which was proved by the finding that loss of vinculin allowed hair follicle stem cells to escape quiescence and forced the initiation of a new hair cycle[49]. Keratin exposure from TGFβ2-induced spatiotemporal apoptotic ORS cells might influence the mechanical properties of the microenvironment and cell-to-ECM interactions, which might be a cue to drive DP cell condensation and activation of hair follicle stem cells participating in hair germ formation. However, further studies on keratin-mediated mechanotransduction are required.

Taken together, the results presented in this study reveal that hair regeneration is regulated by keratin-mediated germ formation and DP condensation through biological events including TGFβ2-induced ORS cell apoptosis and the spatial exposure of keratin derived from apoptotic ORS cells (Supplementary Fig. 40). Our pilot study indicated that keratin is not only a major structural component of hair, but can also play a functional role in the induction of HG formation and DP condensation, facilitating entry into a new hair cycle. In conclusion, considering the biological function of keratin in hair growth, our study suggests that keratin can be a potent biomaterial for developing therapeutic agents for hair loss treatment. Understanding how cellular behavior is regulated by spatiotemporal keratin exposure from apoptotic epithelial cells can provide additional insight into deciphering cellular interactions between epithelial cells and mesenchyme in the morphogenesis of other tissues.

## Methods

**Experimental design**. This study aimed to unravel the biological functions of keratin in hair growth. First, to determine the activity of hair growth, mouse back skin hair was removed, and hair follicle formation and hair regrowth were evaluated after intradermal injection of hair-derived keratin (Fig. 1b–d). Next, to define the interaction of keratin with major cells participating in hair growth, cellular behaviors such as DP cell condensation and germ formation were studied by treating in vitro DP cells and ORS cell cultures with keratin (Fig. 1e, f, Figs. 2 and 3). The results (Figs. 1–3) led to the hypothesis that keratin-induced hair growth could be closely related to a biological cascade that occurs during the hair cycle with regard to the release of keratin from TGF-β2-induced apoptotic ORS cells at the stages of the anagen-catagen transition. To characterize keratin release from apoptotic ORS cells or deposition, apoptosis of ORS cells and the subsequent keratin release from TGF-β2-treated ORS cells and deposition were evaluated. Direct co-culture of DP cells with TGF-β2-treated ORS cells and DP cell culture in conditioned medium collected from TGF-β2-treated ORS cell culture was done to evaluate the effect of released or deposited keratin on DP cell condensation (Fig. 4). Following DP cell condensation induced by released and deposited keratin, apoptosis-related caspase 3 and caspase 6 expression, and caspase-mediated keratin

degradation was characterized. The release and deposition of keratin through caspase-mediated keratin degradation in TGF-β2-treated apoptotic ORS cells and its effect on DP cell condensation were evaluated by in vitro siRNA-mediated downregulation of caspase 6 mRNA expression in TGF-β2-treated apoptotic ORS cells (Fig. 5). To confirm the effect of keratin released from TGF-β2-treated apoptotic ORS cells on DP cell condensation, keratin was eliminated from the conditioned medium collected from TGF-β2-treated cell culture by immunodepletion, and DP cell condensation was evaluated by culturing DP cells in keratin-depleted conditioned medium (Fig. 6a–c). To prove the effect of keratin released from TGF-β2-treated apoptotic ORS cells on DP cell condensation, keratin 31 (KRT31) and keratin 34 (KRT34) expression were downregulated in ORS cells by in vitro KRT31/KRT34 siRNA transfection. Then P-cadherin-expressing germ formation by ORS cells was observed in KRT31/KRT34-knockdown ORS cell culture in the presence of TGF-β2 to evaluate the effect of keratin spatially deposited on germ formation (Fig. 6d). Finally, to study the role of keratin in hair follicle formation and hair growth in vivo, KRT31/KRT34 were downregulated using Invivofectamine KRT31/KRT34 siRNA transfection in mice (Fig. 7). The whole experimental procedure is illustrated in Supplementary Fig. 41.

**Cell culture**. Human ORS cells (ORS; CEFO, CB-ORS-001: 36 years old female) and human DP cells (DP; CEFO, CB-HDP-001: 54 years old male) were purchased and expanded in each human ORS cell growth medium (CEFO, CB-ORS-GM) and human DP growth medium (CEFO, CB-HDP-GM) at 37 °C in a humidified atmosphere containing 5% CO₂ according to the manufacturer's instructions. Cultures were fed every two days and passaged by treatment with 0.25% trypsin/EDTA (Gibco, 25200056), and the expanded DP cells within 5 passages and ORS cells within 3 passages were used in this study.

**Human hair keratin extraction**. Human hair keratin was extracted by slightly modified Sindai method[12], and kindly provided by Gapi Bio. Briefly explaining, human hair was washed using general detergent and delipidized with chloroform (JUNSEI CHEMICAL, 28560-0350):methanol (Merck Millipore, 106009) (2:1, v/v) for 24 hr at room temperature. The delipidized hair was oxidized with 2(w/v)% peracetic acid (Sigma-Aldrich, 269336) for 12 hr at 37 °C. The hair was reacted with Shindai solution (5 M urea (Sigma-Aldrich,U5378), 2.4 M thiourea (Sigma-Aldrich, T7875), 5% 2-mercaptoethanol (Sigma-Aldrich, M6250), 24 mM trizma base (Sigma-Aldrich, T1503), pH 8.5) for 72 hr at 50 °C. After reaction, The mixture was centrifuged at 3500 rpm for 20 min and supernatant was dialyzed (12–14 kDa cut-off, Spectra/Por 4 dialysis membrane, 132706) against deionized water for 5 days with three changes in water a day. Solution of hair keratin was centrifuged at 3500 rpm for 20 min and supernatant was lyophilized by freeze-dryer.

**Sodium dodecyl sulfate-polyacrylamide gel electrophoresis (SDS-PAGE)**. Characterization of human hair keratin was performed by SDS-PAGE analysis.100 μg of human hair keratin was dissolved in 13 μl of DBPS and mixed with 5 μl of LDS (Invitrogen, B0007). and 2 μl of sample reducing agent (Invitrogen, B0009). Each sample was denatured by heating at 75 °C for 10 min before loading into precast Bolt 4 to 12% Bis-Tris Mini Protein Gels (Invitrogen, NW04122BOX). Electrophoresis was carried out at a constant voltage of 200 V for 22 min in 1× MES Running Buffer (Invitrogen, B0002). Subsequently, separated proteins were stained with SimplyBlue SafeStain (Invitrogen, LC6060).

**Matrix-assisted laser desorption ionization-time of flight mass spectrometer**. Protein spots were enzymatically digested in-gel using modified porcine trypsin (Promega modified). Gel pieces were washed with 50% acetonitrile to remove SDS, salt, and stain. Washed and dehydrated spots were then vacuum dried to remove solvent and rehydrated with trypsin(8–10 ng/μl) solution in 50 mM ammonium bicarbonate pH 8.7 and incubated 8–10 h at 37°C. Samples were analyzed using the BRUKER autoflex maX with LIFTTM ion optics. Both MS and MS/MS data were acquired with a SMARTBEAM LASER with 2 kHz repetition rate, and up to 4000 shots were accumulated for each spectrum. MS/MS mode was operated with 2 keV collision energy; air was used as the collision gas such that nominally single collision conditions were achieved. Although the precursor selection has a possible resolution of 200, in these studies of known single-component analytes a resolution of 100 was utilized. Both MS and MS/MS data were acquired using the instrument default calibration, without applying internal or external calibration. MS/MS ions searches were performed with the license Mascot for in-house use.

**Human hair keratin-mediated hair growth test in mice**. In vivo mouse studies for keratin-mediated hair growth were done separately at Chemon Inc and Konkuk University.

For in vivo studies, six-week-old C57BL/6 male mice were purchased from Orientbio Inc. (Gyeonggi-do, Korea). This study was performed within the animal facility area of Chemon Inc in Gyeonggi bio center, and the animals were housed in a room that was maintained at a temperature of 23 ± 3 °C and a relative humidity of 55 ± 15%, with artificial lighting from 08:00 to 20:00, 150–300 Lux of luminous intensity. Throughout the experimental period, the temperature and humidity of animal room were measured every hour with a computer-based automatic sensor,

and as a result of measurements, there were no deviations to have adverse effect to the result of study. Animals were offered irradiation-sterilized pellet food for lab animal (Teklad certified irradiated global 18% protein rodent diet; 2918 C, Envigo, UK). Underground water disinfected by ultraviolet sterilizer and ultrafiltration was given via water bottle, *ad libitum*. Examination of water was performed by an authorized Gyeonggido Institute of Health & Environment (Suwon, Gyeonggi-do, Republic of Korea), and there were no factors that could affect results. All animal experiments were performed according to Chemon Inc's standard operating procedures. The study protocol was approved by the Institutional Animal Care and Use Committee of Chemon Inc, which is accredited by the Association for Assessment and Accreditation of Laboratory Animal Care International (Approval No.: 2018-08-011). To examine the hair growth promoting effect of keratin according to keratin concentration 0.5(w/v)% and 1.0 (w/v)% keratin in phosphate-buffered saline (PBS; Gibco, 10010023) was used. Animals were anesthetized by intraperitoneally administration of zoletyl and rumpoon mixture (4:1, v/v) at the dose of 1 mL/kg. The dorsal area was shaved with an animal clipper. Upon shaving the mice all of the hair follicles were synchronized in the telogen stage, showing pink color. The wound-free animals were selected and weighed. The selected animals were distributed in a randomized manner so that the average weight of each group was distributed as uniformly as possible according to the weighted weight. All animals were randomized into 5 groups based on different topical applications: normal control, DPBS administration, 0.5% keratin administration, 1% keratin administration, and 3% minoxidil as a positive control. The test substance was administered intradermally, the clinically planned route, and the positive control substance was applied tropically. The keratin groups were administered once at day 1, and the positive control substance was administered once/day, 5 times/week, and 4 weeks. The intradermal administration was divided into 2 sites in the dorsal part of animals inhaled with isoflurane using a 0.3 mL insulin syringe (31 G), and divided into 75 µl at 1 site. Topical administration was applied to the back of the animal, which was corrected for 0.15 mL using a 1 mL syringe, rubbed 10 times with a glass rod, and applied evenly. The mice were sacrificed after 24 weeks.

For another in vivo studies, male C57BL/6 mice were used, which were purchased from YoungBio (Samtako, 1404957265). The mice were housed under controlled condition at a temperature of $23 \pm 2\,°C$, humidity of $50 \pm 5\%$, and light-dark cycle of 12 h. Mice were provided with a laboratory diet and water ad libitum. All animal experiments were approved by the Institutional Animal Care and Use Committee of Konkuk University (KU18159, KU19066), and procedures on animals were performed in accordance with the relevant guidelines and regulations. The hair on the dorsal skin of mice was shaved repeatedly using an electric clipper to synchronize the hair follicle cycle. Before treatment, the dorsal hair was completely removed using the commercial hair removal cream Veet® (Reckitt Benckiser, 62200809951). To examine the hair growth-promoting effect of keratin, 1.0 (w/v)% keratin in phosphate-buffered saline (PBS; Gibco, 10010023) was used. 10-week old mice were shaved repeatedly to synchronize the hair cycle and randomly assigned to three groups: Neg. Con group; 3% Minoxidil group with daily topical application of 100 µl of 3% Minoxidil (Minoxyl® 3%; Hyundai Pharm, Co., Seoul, Korea); 1.0 (w/v)% Keratin group with intradermal injection of 100 µl of 1.0 (w/v)% keratin once. The mice were sacrificed after 2 weeks.

**In vivo dispersion test of fluorescent dye-conjugated keratin.** Human hair keratin was labeled with the fluorescent probe Alexa 488 (Alexa Fluor™ 488 Protein Labeling Kit, Invitrogen, A10235), using a procedure provided by the manufacturer. Briefly, human hair keratin was dissolved in DPBS and sodium bicarbonate (Component B) was added. Subsequently, the keratin solution was reacted at reactive dye (Component A) and room temperature for 1 h. Labeled human hair keratin was purified using purification columns, and injected intradermally into SCID nude mice. In vivo fluorescence signal was observed by the in vivo optical imaging system (Perkin Elmer, IVIS Spectrum), and the fluorescence signal was observed and imaged up to 14 days.

**Interaction assay of DP cells with keratin.** DP cells were seeded at a density of $2 \times 10^4$ cell/cm² on 12 well and six-well non-treated tissue culture plate (SPL LIFE SCIENCES, 32012, 32006). The DP cells were adjusted to be stable for 1 day in human DP growth medium (CEFO, CB-HDP-GM) at 37 °C in a humidified atmosphere containing 5% $CO_2$ prior to keratin treatment. After 1 day of adjustment, DP cells were cultured in human DP growth medium containing 1.0(w/v)% keratin or not. The morphological change of DP cells in the presence of keratin was observed under inverted fluorescent microscopy (Olympus IX71), and the number of condensed DP cell aggregates was counted. Cell proliferation upon keratin treatment was measured using Cell Counting Kit-8 (Dojindo Molecular Technologies, CK04-20). DP cells were seeded on 12 well non-treated tissue culture plate (SPL LIFE SCIENCES, 32012) at a seeding density of $1 \times 10^4$ cells/cm², and cultured in human DP growth medium (CEFO, CB-HDP-GM) containing 1.0(w/v)% keratin or not in a humidified atmosphere of 5% $CO_2$ at 37 °C, and the medium was refreshed every two days. At specific time points (1, 3, and 5 days), each well had 10 µl of the Cell Counting Kit-8 solution added and then was incubated at 37 °C for 2 h. Cell proliferation assays were performed in a 96-well plate reader by measuring the absorbance at a wavelength of 450 nm. In a parallel study, BrdU incorporation assay was done by BrdU Cell Proliferation ELISA Kit (Abcam,

ab126556). The analysis was performed in accordance with the manufacturer's instructions. In addition, LIVE/DEAD™ visibility/cytotoxicity kit (Invitrogen, L3224) was used for the LIVE/DEAD assay. Before the LIVE/DEAD assay, 5 µl of calcein AM (Component A) and 20 µl of EthD-1 (Component B) were added to 10 ml of DPBS to prepare a LIVE/DEAD assay working solution. Cells were washed by DPBS, and LIVE/DEAD assay working solution was added. After incubation at room temperature for 10 min, the stained cells were observed, and images were taken using inverted fluorescent microscopy (Olympus IX71).

For DP cell condensation assay according to different cell seeding density, DP cells were seeded at a seeding density of $5 \times 10^3$ cell/cm², $1 \times 10^4$ cell/cm² and $2 \times 10^4$ cell/cm² on 6 well non-treated tissue culture plate (SPL LIFE SCIENCES, 32006). The DP cells were adjusted to be stable for 1 day in human DP growth medium (CEFO, CB-HDP-GM) at 37 °C in a humidified atmosphere containing 5% $CO_2$ prior to keratin treatment. After 1 day of adjustment, DP cells were cultured in human DP growth medium containing 1.0 (w/v)% keratin or not. The number of condensed DP cell aggregates was counted using inverted fluorescent microscopy (Olympus IX71).

**Interaction assay of DP cells with keratin on matrigel.** For DP cell condensation assay on matrigel, the matrigel was diluted at 1:2 volume ratio in ice-cold serum-free DMEM media (CORNING, 10-013-CV), and gelation was done by incubating at 37 °C for 30 min. DP cells were seeded at a seeding density of $2 \times 10^4$ cell/cm² on top of matrigel, and the DP cells on matrigel were adjusted to be stable for 1 day in human DP growth medium (CEFO, CB-HDP-GM) at 37 °C in a humidified atmosphere containing 5% $CO_2$ prior to keratin treatment. After 1 day of adjustment, DP cells were cultured in human DP growth medium containing 1.0(w/v)% keratin or not. Cell viability of DP cells on matrigel was visualized using two-color fluorescence cell viability assay, EthD-1/ Calcein AM live/dead assay (Invitrogen, L3224), which was done according to the manufacturer's instruction, and the morphological change and live/dead stage of DP cells on matrigel in the presence of keratin was observed under inverted fluorescent microscopy (Olympus IX71).

**Indirect analysis of physical property of matrigel.** 1 ml of Matrigel (Growth factor reduced basement membrane matrix, Corning, CLS356231) was mixed to contain 100,000,000 FluoSpheres (Invitrogen, F13080). 500 µl of Matrigel mixed with FluoSpheres was added to the insert of the transwell (Corning, CLS3460) and incubated at 37 °C for >1 h. Matrigel reacted to various conditions in the presence or absence of cells or human hair keratin. FluoSpheres emitted by changes in the physical properties of Matrigel were measured by fluorescence density by spectrofluorometer (Biotec Ex 430/ Em 465).

**Interaction assay of ORS cells with keratin.** ORS cells were seeded at $2 \times 10^4$ cell/cm² on 12 well and six-well tissue culture plate (SPL LIFE SCIENCES, 30012, 30006). The ORS cells were adjusted to be stable for 1 day in human ORS cell growth medium (CEFO, CB-ORS-GM) at 37 °C in a humidified atmosphere containing 5% $CO_2$ prior to keratin treatment. After 1 day of adjustment, ORS cells were cultured in human DP growth medium containing 1.0(w/v)% keratin or not. The morphological change of ORS cells in the presence of keratin was observed under inverted fluorescent microscopy (Olympus IX71) and time-lapse images were captured. Cell proliferation upon keratin treatment was measured using Cell Counting Kit-8 (Dojindo Molecular Technologies, CK04-20). ORS cells were seeded on 12 well tissue culture plate (SPL LIFE SCIENCES, 30012) at a seeding density of $1 \times 10^4$ cells/cm², and cultured in human ORS cell growth medium (CEFO, CB-ORS-GM) containing 1.0(w/v)% keratin or not in a humidified atmosphere of 5% $CO_2$ at 37 °C, and the medium was refreshed every two days. At specific time points (1, 3, and 5 days), each well had 10 µl of the Cell Counting Kit-8 solution added and then was incubated at 37 °C for 2 h. Cell proliferation assays were performed in a 96-well plate reader by measuring the absorbance at a wavelength of 450 nm. In a parallel study, BrdU staining was done by immunocytochemical staining with mouse anti-BrdU antibody (Santa Cruz Biotechnology, sc-32323, diluted 1:100). In addition, BrdU incorporation assay was done by BrdU Cell Proliferation ELISA Kit (Abcam, ab126556) according to the manufacturer's instructions. The final concentration of BrdU in the cell culture medium was 10 µM, and BrdU was added before 24 h of assay. DNA hydrolysis was additionally performed by 1 M HCl.

**RNA extraction and sequencing.** To perform transcriptome sequencing (RNA-Seq) analysis of DP cells and ORS cells, total RNA was extracted from the ORS and DP cells in the absences of keratin and in the presence of keratin. Quality and integrity of the extracted total RNA was assessed by BioAnalyzer and the standard illumina sequencing system protocol (TruSeq Stranded mRNA LT Sample Prep Kit) have been used to make libraries for RNA-Seq. Around 300 bp fragments were isolated using gel electrophoresis, amplified by PCR and sequenced on the Illumina HiSeq 2500 in the paired-end sequencing mode ($2 \times 10^1$ bp reads).

**RNA-seq read processing and differential gene expression analysis.** Quality of the raw sequencing reads were assessed, and qualified raw sequencing reads were aligned to the human genome reference hg19 using TopHat alignment tool (v2.1.0) [PMID: 23618408]. Uniquely and properly mapped read pairs have been used for

further analysis. Gene annotation information was downloaded from Ensembl (release 75) biomart (http://www.ensembl.org/). To evaluate expression levels of genes, the RPKM (reads per kilobase of exon per million mapped reads) measurement unit was used [PMID: 18516045] and the fold change between two samples (untreated and treated with keratin) was calculated based on the calculated RPKM. DESeq2 R package [PMID: 20979621] was used to identify differentially expressed genes between undifferentiated neural stem cell and the differentiated dopaminergic neuron. Differentially expressed genes were defined as those with changes of at least twofold between samples at a false discovery rate (FDR) of 5%.

**DP cell spheroid formation and maintenance assay of the replated DP cell spheroids.** For DP cell spheroid formation, cell spheroids as a micro tissue unit were generated by docking DP cells into polyethylene glycol (PEG) microwell array with 450 μm in diameter. PEG microwells were fabricated by microfabrication procedures, reported previously[50]. Micropatterns with 450 μm in diameter were generated on a silicon wafer using SU-8 photoresist (MicroChem Corp, USA). Poly(dimethylsiloxane) (PDMS) molds were generated by mixing silicone elastomer base solution and curing agent (Sylgard 184, Essex Chemical, USA) in a 10:1 ratio and pouring on the patterned silicon master. The generated PDMS stamps were used to mold PEG microwells. Hydrogel microwells were fabricated using micromolding of poly(ethylene) glycol (PEG)-diacrylate (1000 Da) mixed with 1% (w/w) of the photoinitiator Irgacure 2959 (Ciba Specialty Chemicals Corp., Tarrytown, NY). Glass substrates were treated with 3-(Trimethoxysilyl) propylmethacrylate (TMSPMA) (Sigma, USA) for 30 min and baked at 70 °C overnight. A microfabricated PDMS stamp was placed on the PEG monomer solution on a treated glass slide. The monomers were crosslinked by exposing to UV light (350–500 nm wavelength, 100 mW/cm$^2$) for 27 sec. After peeling the PDMS stamp from the substrate, the microwell structures were washed with ethanol and 1× DPBS overnight before using them to culture cells. In all, 200 μL of cell suspension (1 × 10$^6$ cells per mL) was spread on PEG microwells mounted on a glass slide, and undocked cells were removed by gentle washing with PBS after 30 mins of incubation. DP cell spheroid was formed within PEG microwell by incubating the cell-docked microwell in human DP growth medium (CEFO, CB-HDP-GM) at 37 °C in a humidified atmosphere containing 5% CO$_2$ for 1 day. The formed DP cell spheroids were retrieved from PEG microwell by gentle agitation, and then replated on 12 well and six-well tissue culture plate (SPL LIFE SCIENCES, 30012, 30006). The replated DP cell spheroids were adjusted to adhere on tissue culture plate for 1 day in human DP growth medium (CEFO, CB-HDP-GM) at 37 °C in a humidified atmosphere containing 5% CO$_2$ prior to keratin treatment. After 1 day of adjustment, DP cells were cultured in human DP growth medium containing 1.0(w/v)% keratin or not. The morphological change of DP cell spheroids in the presence of keratin was observed under inverted fluorescent microscopy (Olympus IX71) and CLSM (Confocal Laser Scanning Microscopy, Nikon, D-ECLIPSE C1).

**TGFβ2-mediated ORS cell apoptosis and co-culture with DP cells.** ORS cells were seeded at 2 × 10$^5$ cell/cm$^2$ on 12 well tissue culture plate (SPL LIFE SCIENCES, 30012) to make confluent ORS cell layer. The ORS cells were adjusted to be stable for 1 day in human ORS cell growth medium (CEFO, CB-ORS-GM) at 37 °C in a humidified atmosphere containing 5% CO$_2$. After 1 day of adjustment, ORS cells were cultured in human DP growth medium containing 100 ng/ml TGFβ2 (PeproTech, 100-35B) for 5 days, and the media was refreshed every day.

To evaluate DP cell condensation in direct co-culture of DP cells and TGFβ2-treated ORS cells, DP cells were stained with a cell tracker (Red CMTPX, Invitrogen, C34552) according to manufacturer's instruction. The stained DP cells were seeded at a density of 2 × 10$^4$ cell/cm$^2$ on confluent TGFβ2-treated ORS cell layer which cultured for 5 days prior to co-culture, and cultured in human DP growth medium (CEFO, CB-HDP-GM) at 37 °C in a humidified atmosphere containing 5% CO$_2$. After 1 and 2 days of co-culture, DP cell condensation was observed under inverted fluorescent microscopy (Olympus IX71).

To evaluate DP cell condensation under conditioned media from TGFβ2-treated ORS cell layer, the conditioned media were collected from TGFβ2-treated ORS cell layer after 5 days of culture. DP cells were seeded at a density of 2 × 10$^4$ cell/cm$^2$ on 12 well and 6 well non-treated tissue culture plate (SPL LIFE SCIENCES, 32012, 32006). The DP cells were adjusted to be stable for 1 day in human DP growth medium (CEFO, CB-HDP-GM) at 37 °C in a humidified atmosphere containing 5% CO$_2$. After 1 day of adjustment, DP cells were cultured under various culture media; human DP growth medium, human ORS cell growth medium containing 100 ng/ml TGFβ2, conditioned medium collected from TGFβ2-treated ORS cell layer and human DP growth medium containing 1(w/v)% keratin. After 1 and 2 days of culture, DP cell condensation was observed under inverted fluorescent microscopy (Olympus IX71).

**Immunodepletion study.** To study the role of keratin released from TGFβ2-induced apoptotic ORS cells in DP condensation and germ formation of ORS cells, the released keratin in conditioned media from TGFβ2-treated ORS cell layer culture was removed by immunodepletion method. First, antibodies-conjugated beads were prepared as follows; 150μl of nProtein A Sepharose (GE Healthcare, 17528001) was incubated with 400μl of guinea pig anti-Type I + II Hair Keratins

antibody (PROGEN, GP-panHK) or guinea pig normal IgG (Sigma-Aldrich, I4756), as another negative control, for 18 h at 4 °C. Non-specific binding was prevented with blocking buffer containing 1% bovine serum albumin (BSA; Sigma-Aldrich, A9418) in TBS (Tris-Buffered Saline; Biosesang, TR2005-000-74) with 0.1% Tween 20 (Duchefa Biochemie, P1362.1000) for 3 h at 4 °C. The conditioned media were collected from TGFβ2-treated ORS cell layer cultured for 5 days, and 30 ml of the conditioned media were mixed with 75μl antibodies-conjugated beads, and then incubated with gentle shaking overnight at 4 °C. After incubation, antibodies-conjugated beads were removed by passing the mixture through a Centrifuge Columns (Thermo Scientific, 89898).

To evaluate DP cell condensation under keratin-removed conditioned media, DP cells were seeded at a density of 2 × 10$^4$ cell/cm$^2$ on 12 well and six-well non-treated tissue culture plate (SPL LIFE SCIENCES, 32012, 32006). The DP cells were adjusted to be stable for 1 day in human DP growth medium (CEFO, CB-HDP-GM) at 37 °C in a humidified atmosphere containing 5% CO$_2$. After 1 day of adjustment, DP cells were cultured under various culture media; human DP growth medium, conditioned medium collected from TGFβ2-treated ORS cell layer, conditioned medium collected from TGFβ2-treated ORS cell layer and incubated with guinea pig normal IgG-conjugated beads, conditioned medium collected from TGFβ2-treated ORS cell layer and incubated with guinea pig anti-Type I + II Hair Keratins antibody-conjugated beads, and human DP growth medium containing 1(w/v)% keratin. After 1 and 2 days of culture, DP cell condensation was observed, and the number of condensed DP cell aggregates was counted using inverted fluorescent microscopy (Olympus IX71).

To evaluate P-cadherin expressing germ formation of ORS cells under keratin-removed conditioned media, ORS cells were seeded at a density of 2 × 10$^5$ cell/cm$^2$ on 12 well tissue culture plate (SPL LIFE SCIENCES, 30012) to make confluent ORS cell layer. The ORS cells were adjusted to be stable for 1 day in human ORS cell growth medium (CEFO, CB-ORS-GM) at 37 °C in a humidified atmosphere containing 5% CO$_2$. After 1 day of adjustment, ORS cells were cultured under various culture media; human ORS cell growth medium, conditioned medium collected from ORS cell layer, conditioned medium collected from TGFβ2-treated ORS cell layer, conditioned medium collected from TGFβ2-treated ORS cell layer and incubated with guinea pig normal IgG-conjugated beads, conditioned medium collected from TGFβ2-treated ORS cell layer and incubated with guinea pig anti-Type I + II Hair Keratins antibody-conjugated beads, and human ORS cell growth medium containing 1(w/v)% keratin. After 3 days of culture, P-cadherin expressing germ formation of ORS cells was characterized by immunocytochemical staining.

**Caspase-3 and caspase-6-mediated hair keratin digestion assay.** 1(w/v)% hair keratin was dissolved in the reaction solution composed of 50 mM HEPES (Gibco, 15630-080), 50 mM NaCl (JUNSEI CHEMICAL, 19015-1250), 0.1% CHAPS (Sigma-Aldrich, C3023), 10 mM EDTA (Sigma-Aldrich, 03609), 5% glycerol (SAMCHUN CHEMICALS, G0274) and 10 mM DTT (Sigma-Aldrich, 43815) at pH 7.2. 5U/ml Casase-3 (Enzo, ALX-201-059) or 5U/ml Casase-6 (Enzo, ALX-201-060) was added to the reaction solution containing hair keratin and incubated at 37 °C for 0, 1, 3, and 24 h. After the reaction, samples were denatured at 70 °C for 10 min in LDS sample buffer (Invitrogen, B0007). Equal amounts of denatured samples were loaded in pre-casted 4–12% Bis-Tris Plus Gels (Invitrogen, NW04120BOX), and the electrophoresis was done by running at 200 V for 22 min. The gel was rinsed three times with distilled water for 5 min each and stained by SimplyBlue SafeStain (Invitrogen, LC6060). After 1 hr of staining, the gel was rinsed using distilled water until the background was removed thoroughly, and then images of the gel was obtained using a commercialized scanner (Canon, TS8090).

**In vitro caspase-6 gene knockdown study.** To evaluate the effect of caspase-6-mediated ORS cell degradation during TGFβ2-induced ORS cell apoptosis on keratin release or deposition and DP condensation, caspase-6 gene expression in ORS cells was silenced by caspase-6 siRNA transfection. Capase-6 siRNA duplex (Bioneer, 839-1) or negative control siRNA duplex (Bioneer, SN-1012) was diluted in 250 μl Opti-MEM (Gibco, 31985062) to make a final concentration of 100 nM. 3.5 μl Lipofectamine 2000 (Invitrogen, 11668-030) was mixed in 250 μl Opti-MEM, and the mixture was incubated for 5 min at room temperature. The diluted caspase-6 siRNA duplex and diluted Lipofectamine 2000 were mixed and incubated for 20 min at room temperature. Before transfection, ORS cells were seeded at a density of 2 × 10$^5$ cell/cm$^2$ on 12 well tissue culture plate (SPL LIFE SCIENCES, 30012) to make confluent ORS cell layer. The ORS cells were adjusted to be stable for 1 day in human ORS cell growth medium (CEFO, CB-ORS-GM) at 37 °C in a humidified atmosphere containing 5% CO$_2$. After 1 day of adjustment, human ORS cell growth medium (CEFO, CB-ORS-GM) was changed with fresh same medium without serum. The capase-6 siRNA/Lipofectaminr 2000 mixture or negative control siRNA/Lipofectamine 2000 mixture was added to ORS cell culture, and incubated for 5 hr at 37 °C. After transfection, the medium was changed with a fresh medium containing serum, and the transfected ORS cells were cultured in the presence of 100 ng/ml TGFβ2 or in the absence of TGFβ2 for 5 days.

To evaluate DP cell condensation in direct co-culture of DP cells and TGFβ2-treated/caspase-6-silenced ORS cells, DP cells were stained with a cell tracker (Invitrogen, C34552) according to manufacturer's instruction. The stained DP cells were seeded at a density of 2 × 10$^4$ cell/cm$^2$ on confluent TGFβ2-treated/caspase-6-

silenced ORS cell layer which was cultured for 5 days prior to co-culture, and cultured in human DP growth medium (CEFO, CB-HDP-GM) at 37 °C in a humidified atmosphere containing 5% $CO_2$. After 2 days of co-culture, DP cell condensation was observed under inverted fluorescent microscopy (Olympus IX71).

To evaluate DP cell condensation under conditioned media from TGFβ2-treated/caspase-6-silenced ORS cells, the conditioned media were collected from TGFβ2-treated/caspase-6-silenced ORS cell culture after 5 days. DP cells were seeded at a density of $2 \times 10^4$ cell/$cm^2$ on 12 well and 6 well non-treated tissue culture plate (SPL LIFE SCIENCES, 32012, 32006). The DP cells were adjusted to be stable for 1 day in a human DP growth medium (CEFO, CB-HDP-GM) at 37 °C in a humidified atmosphere containing 5% $CO_2$. After 1 day of adjustment, DP cells were cultured under various culture media; human DP growth medium, human ORS cell growth medium containing 100 ng/ml TGFβ2, conditioned medium collected from TGFβ2-treated/negative control siRNA-transfected ORS cell layer, conditioned medium collected from TGFβ2-treated/negative control siRNA-transfected ORS cell layer containing additional 100 ng/ml of TGFβ2, and conditioned medium collected from TGFβ2-treated/caspase-6-silenced ORS cell layer. After 1 and 2 days of culture, DP cell condensation was observed under inverted fluorescent microscopy (Olympus IX71).

**In vitro KRT31/KRT34 gene knockdown study**. To evaluate the effect of KRT31/KRT34 gene knockdown during TGFβ2-induced ORS cell apoptosis on keratin release or deposition and germ formation of ORS cells, KRT31 and KRT34 gene expressions in ORS cells were silenced by KRT31/KRT34 siRNA transfection. KRT31 siRNA duplex (Bioneer, 3881-1) and KRT34 siRNA duplex (Bioneer, 3885-1) or negative control siRNA duplex (Bioneer, SN-1002) were diluted in 250 μl Opti-MEM (Gibco, 31985062) to make a final concentration of 100 nM. 3.5-μl Lipofectamine 2000 (Invitrogen, 11668-030) was mixed in 250 μl Opti-MEM, and the mixture was incubated for 5 min at room temperature. The diluted KRT31/KRT34 siRNA duplex and diluted Lipofectamine 2000 were mixed and incubated for 20 min at room temperature. Before transfection, ORS cells were seeded at a density of $2 \times 10^5$ cell/$cm^2$ on 12 well tissue culture plate (SPL LIFE SCIENCES, 30012) to make confluent ORS cell layer. The ORS cells were adjusted to be stable for 1 day in human ORS cell growth medium (CEFO, CB-ORS-GM) at 37 °C in a humidified atmosphere containing 5% $CO_2$. After 1 day of adjustment, human ORS cell growth medium (CEFO, CB-ORS-GM) was changed with fresh same medium without serum. The KRT31/KRT34 siRNA/Lipofectamine 2000 mixture or negative control siRNA/Lipopectamin 2000 mixture was added to ORS cell culture, and incubated for 5 hr at 37 °C. After incubation, the medium was changed with a fresh medium containing serum, and the transfected ORS cells were cultured in the presence of 100 ng/ml TGFβ2 for 5 days, and morphological change was observed under inverted fluorescent microscopy (Olympus IX71).

In order to evaluate the activity of KRT31/KRT34 knockdown ORS cells for cell growth and differentiation, cell proliferation was tested by Cell Counting Kit-8 up to 5 days after gene knockdown. Each well-containing cells was replaced with 100 μl of DMEM and 10 μl of Cell Counting Kit-8 solution, and incubated at 37 °C for 1 h. Cell proliferation assays were performed in a 96-well plate reader by measuring the absorbance at a wavelength of 450 nm. In addition, BrdU was added to each well-containing cells to observe BrdU incorporation activity by immunocytochemical staining. BrdU was added as final concentration of 10 μM, and the treated cells were cultured for 24 h. After cell fixation and permeabilization for immunofluorescent staining, DNA hydrolysis was performed using 1 M Hcl. Subsequent procedures were performed in accordance with the procedures of Immunocytochemical staining.

**Apoptosis and growth factor antibody array**. TGFβ2-induced ORS cell apoptosis was evaluated by comparative analysis using human apoptosis antibody array (Abcam, ab134001), and the comparative analysis of growth factors present in the conditioned medium collected from TGFβ2-treated ORS cell culture in immuno-depletion study were done using human growth factor antibody array (Abcam, ab134002) according to manufacturer's instructions.

For apoptosis array analysis, cell lysates were prepared as follows; cells on culture plate were washed with PBS three times. After washing, cells were incubated with the 1× Lysis buffer at 4 °C for 30 min, and lysates were centrifuged at 12,000 rpm for 10 min to remove the debris. After centrifuge, the supernatant was collected and used for the apoptosis array analysis. Following process of apoptosis antibody array was done according to manufacturer's instructions. Dot blots on membrane were analyzed using the plugin Protein Array Analyzer (http://image.bio.methods.free.fr) on ImageJ (https://imagej.nih.gov/ij/).

For growth factor antibody array analysis, in the immunodepletion study, conditioned media were collected from TGFβ2-treated ORS cell culture and prepared as follows; the collected conditioned medium were centrifuged to remove debris, and the supernatant was collected. The supernatant was concentrated ~20 times by Amicon Ultra centrifugal filter units (Millipore, Z717185). The concentrated conditioned media were used for growth factor antibody array analysis and following process of growth factor antibody array was done according to manufacturer's instructions. Dot blots on membrane were analyzed using the plugin Protein Array Analyzer (http://image.bio.methods.free.fr) on ImageJ (https://imagej.nih.gov/ij/).

**Western blot analysis**. Molecular expressions of KRT34 and β-catenin in hair keratin-treated ORS cells, keratin content at protein level in conditioned medium collected from TGFβ2-treated ORS cell culture, the keratin content in keratin-removed condition medium collected from TGFβ2-treated ORS cell culture in immunodepletion study, keratin content at protein level in conditioned medium collected from KRT31/KRT34-silenced ORS cell culture or molecular keratin expression in KRT31/KRT34-silenced ORS cell, molecular expressions of caspase 6 in TGFβ2-treated ORS cells and keratin content at the protein level in conditioned medium collected from caspase 6-silenced ORS cell culture were evaluated by western blot analysis.

Proteins from various conditioned medium were concentrated about twentyfold by Amicon Ultra centrifugal filter units (Millipore, Z717185), and cells were lysed on ice for 30 min in 100 μl ice-cold RIPA lysis buffer (Millipore, 20–188) containing a protease inhibitor cocktail (Roche, 4693116001), and then lysates were centrifuged at 12,000 rpm to remove debris. Samples were denatured at 70 °C for 10 min in LDS sample buffer (Invitrogen, B0007), and equal amounts of denatured samples were loaded in pre-casted 4–12% Bis-Tris Plus Gels (Invitrogen, NW04120BOX), and the electrophoresis was done by running at 200 V for 22 min. After electrophoresis, the proteins in the gel were transferred to PVDF membranes (Bio-Rad, 1620174) using electrophoretic transfer cell (Bio-Rad, 1703930). Immunoblotting for the membranes was carried out as follows; the membranes were incubated in TBS containing 5% skim milk (bioWORLD, 30620074-1) at room temperature for 60 min. The membranes were further incubated in TBS containing 1% skim milk and primary antibodies for overnight at 4 °C. The primary antibodies used in western blot analysis were as follows; rabbit Anti-KRT34 (LifeSpan BioSciences, LS-B15620, diluted 1:1000), guinea pig Anti-Type I + II Hair Keratins (PROGEN, GP-panHK, diluted 1:1000), mouse anti-FGF-20 (Santa Cruz, sc-398722, diluted 1:1000), rabbit anti-Annexin V (Abcam, ab14196, diluted 1:2000), rabbit anti-Caspase-3 (Cell Signaling Technology,8G10, diluted 1:2000) and Mouse Anti-GAPDH (Abcam, ab8245, diluted 1:5000). After incubation with primary antibodies, the membranes were incubated with following secondary antibodies in TBS containing 1% skim milk at room temperature for 2 hr; HRP conjugated goat anti-guinea pig IgG (Abcam, ab97155, diluted 1:10000), HRP conjugated mouse anti-rabbit IgG (Santa Cruz Biotechnology, sc-2357, diluted 1:5000) and HRP conjugated goat anti-mouse IgG (Santa Cruz Biotechnology, sc-2005, diluted 1:5000). At each process, the membranes were washed three times with TBS containing 0.1% Tween 20 (Duchefa Biochemie, P1362.1000) for 10 min. The membranes were treated with ECL substrate (Bio-Rad, 1705061) to visualize signal, and the signals on the membranes were transferred to X-ray film (AGFA, CP-BU New).

**Real-time quantitative polymerase chain reaction (RT-qPCR)**. The gene expressions indicative of DP cell's intrinsic property and TGFβ2 gene expressions of DP cell spheroids and the replated DP cell spheroids were evaluated by RT-qPCR. The gene expressions indicative of DP cell's intrinsic properties were evaluated by RT-qPCR. Total RNA was extracted from DP cells and DP Cell spheroids using Hybrid-R (GeneAll, 305-101). cDNAs were synthesized by reverse transcription reaction using a CycleScript RT PreMix (Bioneer, K-2044). After cDNA synthesis, cDNAs were mixed with SYBR kit (PhileKorea, QS105-10) and primers (Table S1). The mixture was reacted by thermal cycler (QIAGEN, Rotor-Gene Q) and analyzed by the program provided (Rotor-Gene Q Series Software, V1.7). The whole process was carried out according to the manufacturer's instructions.

TGFβ2 gene expressions of DP cell spheroids and the replated DP cell spheroids were evaluated by RT-qPCR. DP cell spheroids were generated by PEG microwell-mediated condensation as previously described. The formed DP cell spheroids were retrieved from PEG microwell, and then replated on 6 well tissue culture plate (SPL LIFE SCIENCES, 30006). The replated DP cell spheroids were cultured in human DP growth medium (CEFO, CB-HDP-GM) at 37 °C in a humidified atmosphere containing 5% $CO_2$. After 0, 1, 3, 5, and 7 days of culture, total RNA was extracted from DP cell spheroid, and RT-qPCR was done by the previously described same method. The primer used in this experiment was noted in Table S1.

**Indirect enzyme-linked immuno-sorbent assay (ELISA)**. The molecular expressions indicative of DP cell's intrinsic property from the replated DP spheroids cultured in the presence of keratin were evaluated by ELISA. Replated DP spheroids cultured in the presence of keratin were harvested and lysed with RIPA lysis buffer (Millipore, 20–188) containing a protease inhibitor cocktail (Roche, 4693116001). Lysates were centrifuged for 15 min at 12,000 rpm to remove the debris, and the supernatant was collected. The lysates were diluted to a final concentration of 20 μg/ml in DPBS (Dulbecco's Phosphate-Buffered Saline; Gibco, 14190-144). Antigens were coated on wells in a 96-well ELISA plate (Invitrogen, 44-2404-21) by loading 50 μl of the diluted lysate, and the wells were incubated at 4 °C for 18 hr. The plate was washed three times with 200 μl of DPBS, and non-specific binding of antibodies was blocked by treating with DPBS containing 5(w/v)% bovine serum albumin (BSA; Sigma-Aldrich, A9418) for 2 hr at room temperature. Primary antibodies such as rabbit anti-β-actin (Abcam, ab8227, diluted 1:1000), rabbit anti-β-catenin (Abcam, ab16051, diluted 1:1000), rabbit anti-FGF7 (Santa Cruz Biotechnology, sc-7882, diluted 1:200), Goat anti-FGF10 (Santa Cruz Biotechnology, sc-7375, diluted 1:200) and goat anti-BMP6 (Santa Cruz Biotechnology, sc-7406, diluted 1:200) were diluted in DPBS containing 1(w/

v)% BSA, and added 100 µl to each well. The plate was incubated at room temperature for 2 hr for the reaction of antibodies with antigens. After incubation with primary antibodies, the wells were washed with DPBS, and 100 µl of HRP conjugated secondary antibodies (anti-rabbit IgG-HRP (Santa Cruz Biotechnology, sc-2004) or anti-goat IgG-HRP (Santa Cruz Biotechnology, sc-2033)) was added to each well, and then was incubated for 2 hr at room temperature. After incubation with secondary antibodies and following washing with DPBS. 100 µl of TMB substrate solution (Thermo Scientific, N301) was added to each well and incubated for 30 min at room temperature. The reaction was stopped by 100 µl of the stop solution (2 M sulfuric acid; Sigma-Aldrich, 258105), and the absorbance of samples was measured at 450 nm on a microplate spectrophotometer reader (Bio-Rad, Benchmark Plus, BR170-6930).

**In vivo KRT31/KRT34 gene knockdown study.** To confirm the effect of keratin on hair growth, KRT31/KRT34 was silenced by lipofectamine-mediated delivery of KRT31/KRT34 siRNAs. First, Invivofectamine complex for KRT31/KRT34 siRNA delivery was prepared as follows; siRNAs of KRT31 (Bioneer, 16660-1), KRT34 (Bioneer, 16672-1) and negative control (Bioneer, SN-1003) were purchased, and siRNAs of KRT31 and KRT34 were dissolved in RNase-free water as each 24 mg/ml concentration respectively. The two solutions, KRT31 siRNA and KRT34 siRNA, were combined as 1:1 volume ratio to be 12 mg/ml of final concentration. 12 mg of negative control siRNA was also dissolved in 1 ml of RNase-free water. siRNAs-Invivofectamine (Thermo Fisher Scientific, IVF3005) complex was prepared by the manufacturer's instruction, and 0.5 mg/ml of the complex was prepared finally prior to injection to mice. For in vivo study, 6-week-old mice were shaved repeatedly to synchronize the hair cycle and randomly assigned to three groups: Con group with IV injection of 200 µl of negative control siRNA injection; siRNA group with IV injection of 200 µl of KRT31/KRT34 siRNA injection; siRNA+Keratin group with KRT31/KRT34 siRNA injection (IV, 200 µl) and intradermal injection of total 100 µl of keratin a day after first siRNA injection. For each group, mice were sacrificed at either day 7 or day 14. Pictures of the back skin were taken on day 3, 7, 10, and 14 to examine the hair growth. The silencing of KRT31/KRT34 gene expressions was confirmed by RT-qPCR.

Total RNA was extracted from the cells using Hybrid-R (GeneAll, 305-101). cDNAs were synthesized by reverse transcription reaction using a CycleScript RT PreMix (Bioneer, K-2044). After cDNA synthesis, cDNAs were mixed with SYBR kit (PhileKorea, QS105-10) and primers (Table S2). The mixture was reacted by a thermal cycler (QIAGEN, Rotor-Gene Q) and analyzed by the program provided (Rotor-Gene Q Series Software, V1.7). The whole process was carried out according to the manufacturer's instructions.

**Histological analysis.** The skin tissues were fixed with 10% neutral-buffered formalin (BBC Biochemical, 0141). The tissues were embedded in paraffin and sectioned at 4-µm thickness, followed by staining with hematoxylin and eosin for histological analysis. The number of hair follicles in each cycle and the diameter of anagen hair follicles was quantified in multiple fields on perpendicular sections at ×100 magnification.

**Immunocytochemical staining.** DP cells and ORS cells were fixed for 10 min in 3.7% paraformaldehyde (Sigma-Aldrich, F8775), and were permeabilized in 0.2% Triton X-100 (Sigma-Aldrich, T9284). Non-specific binding was blocked by treating with 4% BSA (Sigma-Aldrich, A9418). Cells were incubated in primary antibody diluents (GBI Labs, E09-500) containing the following primary antibodies for overnight at 4 °C; rabbit anti-β-catenin (Abcam, ab16051, diluted 1:100), rabbit anti-SOX2 (Cell Signaling Technology, 3579 S, diluted 1:200), rabbit anti-CD133 (Abcam, ab16518, diluted 1:50), mouse anti-integrin β1 (Santa Cruz Biotechnology, sc-59829, diluted 1:50), rabbit anti-P-cadherin (Cell Signaling Technology, 2189 S, diluted 1:50), mouse anti-E-cadherin (Abcam, ab1416, diluted 1:100), mouse anti-alkaline phosphatase (Abcam, ab126820, diluted 1:100), mouse anti-RUNX1 (Santa Cruz Biotechnology, sc-365644, diluted 1:50), rabbit anti-KRT34 (LifeSpan BioSciences, LS-B15620, diluted 1:100), rabbit anti-FGF7 (Santa Cruz Biotechnology, sc-7882, diluted 1:50), goat anti-FGF10 (Santa Cruz Biotechnology, sc-7375, diluted 1:50), goat anti-BMP6 (Santa Cruz Biotechnology, sc-7406, diluted 1:50), rabbit anti-CD34 (Abcam, ab81289, diluted 1:100), rabbit anti-SOX9 (Abcam, ab185966, diluted 1:100), rabbit anti-Annexin V (Abcam, ab14196, diluted 1:100), rabbit anti-caspase-3 (Abcam, ab13847, diluted 1:100), rabbit anti-caspase-6 (Abcam, ab52951, diluted 1:100), mouse anti-BrdU (Invitrogen, MA3-071, diluted 1:100), rabbit anti-Ki67 (Cell Signaling Technology, 9027 S, diluted 1:100), rabbit anti-Lgr5 (Abcam, ab219107, diluted 1:100) and rabbit anti-Vinculin (Abcam, ab129002, diluted 1:100). After incubation with primary antibodies, cells were washed three times with DPBS and were incubated with the following secondary antibodies for 1 hr at room temperature; Alexa Fluor 488-conjugated goat anti-rabbit IgG (Invitrogen, A-11034, diluted 1:200), Alexa Fluor 594-conjugated goat anti-rabbit IgG (Invitrogen, A-11012, diluted 1:200), Alexa Fluor 594-conjugated goat anti-mouse IgG (Invitrogen, A-11032, diluted 1:200) and Alexa Fluor 488-conjugated goat anti-mouse IgG (Invitrogen, A-11001, diluted 1:200). With secondary antibodies, actin was stained using rhodamine phalloidin (Invitrogen, R415, diluted 1:400), and Tunel staining was performed using the Turner Enzyme (Roche, 11767305001) and Tunel Label (Roche, 11767291910). Briefly, cells were washed twice with DPBS (Gibco, 14190-144) and incubated at 37 °C for 1 hr with

200 µl of TUNEL reaction mixture (Turner Enzyme: Tuner Label, 1:9, v/v). After incubation with secondary antibodies, phalloidin, and Tunel reaction mixture, cells were washed with DPBS three times and counterstained with DAPI (Sigma-Aldrich, D9542). Finally, stained cells were observed under an inverted fluorescence microscope (OLYMPUS, IX71) and captured.

**Immunohistochemical staining.** Paraffin-embedded tissue sections were deparaffinized by treating xylene three times for 5 min and rehydrated in graded ethanol (100%, 95%, 90%, 80%, and 50%). Antigen retrieval step was performed by incubating the sections in sodium citrate buffer (10 mM Sodium Citrate; Biosesang, C2004, 0.05% Tween 20; Duchefa Biochemie, P1362.1000, pH 6.0) at sub-boiling temperature for 20 min. Non-specific binding of antibody was blocked by treating with blocking buffer containing 1% normal horse serum (Abcam, ab7484) and 5% BSA (Sigma-Aldrich, A9418) for 30 min at room temperature. After blocking, tissue sections were incubated in primary antibody diluents (GBI Labs, E09-500) containing the following primary antibodies for overnight at 4 °C; goat anti-P-cadherin (R&D Systems, AF761, diluted 1:50), rabbit anti-β-catenin (Abcam, ab16051, diluted 1:50), rabbit anti-KRT34 (Biorbyt, orb628339, diluted 1:100), guinea pig anti-type I + II hair keratins (PROGEN, GP-panHK, diluted 1:50), mouse anti-Ki67 (Abcam, ab279653, diluted 1:50), rabbit anti-Annexin V (Abcam, ab14196, diluted 1:100), rabbit anti-Caspase-3 (Abcam, ab13847, diluted 1:50), and rabbit anti-active Caspase-3 (Abcam, ab32042, diluted 1:50). When the host species of primary antibodies were a mouse, endogenous mouse Ig of the tissue was blocked with Mouse on Mouse blocking agent (Abcam, BMK-2202) according to the manufacturer's instruction. After washing tissue sections with DPBS three times, were incubated with the following secondary antibodies overnight at 4 °C; donkey Alexa Fluor 488-conjugated anti-goat IgG (Invitrogen, A-11055, diluted 1:200), goat Alexa Fluor 594-conjugated anti-rabbit IgG (Invitrogen, A-11012, diluted 1:200), goat Alexa Fluor 488-conjugated anti-rabbit IgG (Invitrogen, A-11034, diluted 1:200) and goat Alexa Fluor 647-conjugated anti-guinea pig IgG (Abcam, ab150187, diluted 1:200). After incubation with secondary antibodies, tissue sections were washed with DPBS three times, counterstained with DAPI (Sigma-Aldrich, D9542) and mounted with mounting medium (Sigma-Aldrich, F4680). Finally, images were acquired using an inverted fluorescence microscope (OLYMPUS, IX71).

**In situ RNA hybridization.** In situ hybridization (ISH) was performed with the RNAscope (Advanced Cell Diagnostics, ACD), including the RNAscope 2.5 Duplex reagent kit and RNAscope probes. ISH was performed manually with an RNAscope 2.5 HD Duplex assay (Chromogenic) in accordance with the manufacturer's instructions for Formalin-Fixed Paraffin-Embedded (FFPE) tissue. RNA probes hybridizing to mouse LGR5 RNA (ACD, 312178 -C2) and mouse CD34 RNA (ACD, 319168 -C2) were hybridized to the tissues. Steps of amplification and detection were conducted as recommended by the RNAscope 2.5 HD duplex detection kit user manual (ACD, 322500-USM). Tissue sections were observed under a standard light microscope (OLYMPUS, IX71) and captured.

**Flow cytometric analysis.** Molecular expressions of P-Cadherin and RUNX1 for were analyzed by flow cytometry. ORS cells were cultured in the presence of TGFβ2 and harvested by incubating with 0.25% Trypsin/EDTA for 10 min at 37 °C. Cells were recovered by centrifugation and fixed with 0.01% formaldehyde in DPBS for 15 min at room temperature. Cells were incubated for 30 min at room temperature with primary antibodies such as Mouse anti-RUNX1 (Santa Cruz Biotechnology, sc-365644, diluted 1:50) or Mouse anti-P-Cadherin (R&D Systems, FAB861G, diluted 1:100). Cells were resuspended in 100 µl of diluted fluorochrome-binding secondary antibody (Alexa Fluor 488-conjugated anti-mouse IgG, A-11001). Washing with DPBS was performed in each step. Flow cytometric analysis was done using a FACSCanto (BD Biosciences, USA), and analysis of flow cytometry data was done with FlowJo V10 program.

**Statistics and reproducibility.** All values obtained from in vitro and in vivo analysis are presented as the mean ± standard deviation (SD). Statistically, differences were identified by two-sided Student's t test or one-way ANOVA parametric test. A P value of <0.05 was considered significant. The number of samples per independent experiment is described in the legends.

**Reporting summary.** Further information on research design is available in the Nature Portfolio Reporting Summary linked to this article.

## Data availability

All data that support the findings of this study are available from the corresponding author upon reasonable request. Full-length uncropped original western blots used in the manuscript are shown in Supplementary Fig. 42. The numerical data that make up the all graphs in the paper are shown in Supplementary Data 1–6. The transcriptome sequencing data (RNA-Seq) have been deposited at NCBI GenBank under BioProject ID PRJNA576064 (BioSample SAMN12924151–SAMN12924158). The mass spectrometry proteomics data have been deposited to the ProteomeXchange Consortium via the jPOST partner repository with accession numbers PXD037830 (JPST001909).

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

## Acknowledgements

This work was supported by the National Research Foundation of Korea (NRF) grant funded by the Korean government (MSIT) (NRF-2016R1D1A1B03931933) and by the Bio & Medical Technology Development Program of NRF funded by the Korean government (MSIT) (no. 2017M3A9E4048170), and by a fund (2018ER610300) by Research of Korea Centers for Disease Control and Prevention.

## Author contributions

S.Y.A. performed most of the experiments and wrote the paper. S.Y.K. extract and purified human hair-derived keratin. S.Y.V. extract and purified human hair-derived keratin and analyzed gene expressions using real-time-qPCR. H.S.K. carried out in vivo silencing experiment and histological analysis. H.J.K. carried out in vivo mouse experiment and histological analysis. J.H.L. carried out RNA sequencing and data analysis. S.W.H. and I.K.K. discussed the results of the experiments and commented on the manuscript. C.K.L. carried out in vivo mouse experiment and histological analysis. Y.S.H. and S.H.D. directed the project, and Y.S.H. drafted the manuscript with input from all authors.

## Competing interests

The authors declare no competing interests.

**Additional information**

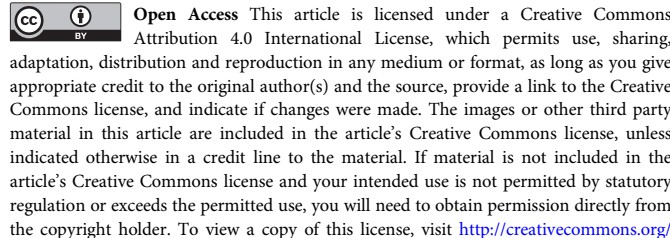

