## [Peer Review File · Communications Biology]

Reviewers' comments:

Reviewer #1 (Remarks to the Author):

In this manuscript, the authors sought to uncover the biological function of keratin in hair growth. They demonstrated that intradermal injection of Keratin could stimulate the hair growth, while silencing keratin in mice resulted in a marked suppression of anagen follicle formation. Through a group of experiments in vitro, they proposed that the keratin exposure, caused by TGF β -induced ORS apoptosis, is essential and critical for dermal papilla cell condensation and hair germ formation. For the most part, their techniques and approach were quite thorough and their controls were appropriate for the bulk of their experiments. Overall, the findings in the manuscript are interesting. However, in its present form, there are a few issues that I believe the authors should address before final publication.

Major comments:

1. A plenty of experiments in vitro showed the the significant effects of keratin in DP condensation and hair germ formation , however is the hair growth promotion by keratin in the mouse model really via its impact on DP/ORs, considering the hair follicle formation, including the DP and bulge, has been finished in adult mice. How could keratin promote a new telogen-anagen transition seems unclear since the in vitro experiments clarified the function of keratin in anagen-catagen transition? Could keratin promote the activation of hair follicle stem cell?

2. In figure 1, the data showed an increased total number of hair follicle with the treatment of keratin, is there any influence of keratin on epidermal stem cell differentiation to form new hair follicle?

3. In figure S15, the author showed a high expression level of caspase 6, P-cadherin and KRT 34 on P8 of mice, what is the intact expression pattern of KRT34 in hair follicle development and hair cycle?

Minor comments:

1. In Figure 1B-F, it will be more clear to identify the stage of hair follicles and more precise to quantify the number and size of HFs with a longitudinal section;

2. Some scale bar seems to be inconsistent with the real presentation, such as Fig 4B;

3. In Figure 4B, the authors showed that the coculture of TGF β 2 treated ORS with DP cells can promote the the DP condensation, without a negative control of naive ORS.

4. Fig 6B showed an increase of TGF β in conditioned medium after immunodepletion, is there any impact of TGF β on DP cell?

5. Figs 15A and Fig 7D, upper panel and lower panel of group siRNA, why the hair follicle size and stage showed different, while one in anagen another in telogen;

6. In Figure S11, the tunel staining seems to be inconsistent with the description;

7. In the method part, could the repeated hair shave with an electric clipper synchronize the hair follicle cycle? Is there any changes in hair cycle with the application of commercial hair removal cream?

Reviewer #2 (Remarks to the Author):

In this paper, Seong Yeong An et al. showed that keratin has the ability to promote hair growth when it was injected intradermally to mouse skin. The author's findings are somewhat surprising, but they tried to provide different lines of evidences to make their points. Although their finding is interesting, I

have several concerns to be addressed.

1. In Figure 1A, they showed clear differences between control and Minoxidil, or Keratin groups in their hair growth at 2 weeks. In supplemental Figure 1A, such changes were not seen in day 14, and started to appear at day 28 when control group also started hair growth. What is the difference between Figure 1A and supplemental Figure 1A? Is this single injection vs multiple treatment? (Nice to show experimental schedule in Figure)
2. In Figure 7B, control mice grow hair after 2 weeks, whereas control mice in Figure 1A didn't. Is this same experimental condition? How can authors explain this discrepancy?
3. In Figure 1A, why HF "size" is increased? Is it simply a hair cycle differences (anagen follicle vs telogen follicle) or due to proliferation of bulge/hair germ (increase of cell number)? Authors show cross-section of skin in Figure 1 but longitudinal sections would be better to define their histology.
4. Did authors detect any changes in morphology of DP (DP condensation) and their gene expression in tissue upon keratin injection (related to Figure 1)? I see staining data in Figure S15, siRNA + Keratin injection. How about Keratin injection only?

Overall, the authors need more careful description and experimental data of histological and marker changes of skin after keratin treatment in vivo to link with their findings in vitro. My main concern is what they found in vitro experiments with human cells are really happening in vivo mouse tissue.

Reviewer #3 (Remarks to the Author):

In their manuscript, entitled: "Keratin is not only a Structural Protein in Hair: Keratin-mediated Hair Growth", Seong Yeong An et al., aim to identify the molecular role of keratin filaments in hair follicle growth and maintenance. By intradermally injecting human hair-derived keratin to mice skin, authors demonstrated accelerated hair growth and a precocious hair formation. Mechanistically, secretion of TGF β 2 by the supportive dermal cells during the hair destruction phase stimulates epithelial cell death and release of keratin to the extracellular space, leading to dermal cell condensation and initiation of an additional cycle of hair growth. The authors then show that in vivo silencing of keratins inhibits anagen entrance and hair follicle growth.

Keratin intermediate filaments are major protein constituents in epithelial cells and the hair follicle. Although highly abundant, the exact nature of communication and mechanistic properties of keratins in hair follicle biology is still scant. Equally unclear is which keratin filament participate in basic processes such as cell differentiation and tissue homeostasis. Understanding the function of keratin produced by the epithelial hair cells may be of high medical interest for hair regeneration. Authors present an intriguing idea by which catagen-mediated apoptotic cell release of keratin can promote hair growth via the control of the supportive dermal compartment, which may extend our knowledge on the molecular basis of why cell death during hair destruction is essential for homeostatic maintenance of hair growth over time.

With that in mind, the current study lacks clarity, fails to provide essential controls and many of the listed claims have not been examined experimentally. The manuscript is not clearly organized into a logical storyline. Together, while the concepts put forth in the manuscript are of interest, the current report would be much improved by providing a clear narrative and more conclusive evidence to support the suggestions made by the authors.

Major points:

1. One major concern is whether intradermal injection of isolated human keratin was thoroughly

controlled in their experiments. It is unclear how far distance the injected keratin is within the skin following injection. If Keratin promotes DP cell condensation, as the authors suggested, it is unclear and completely unexplained why the effect is not locally restricted to the injection site and instead, spread throughout the back skin of the animals, as presented by a profound hair growth effect? Authors should evaluate how far distance keratin is in the back skin following injection, and what is the kinetics of hair growth at the site of injection as well as at distant areas in the back skin.

2. Authors show in Fig 1A a strong hair growth phenotype 2 weeks after a single intradermal injection of Keratin. It is unclear why results presented in Sup. Fig 1, are profoundly different and demonstrating hair growth phenotype only after 4 weeks of treatment (while comparing same keratin concentration as well as minoxidil treatment)?

3. Dermal cell condensation:

(a) Authors should confirm that other factors, known to regulate DP cell condensation are not present in the keratin-extracted solution (i.e Fgf20, etc).

(b) Similarly, authors should confirm that an in-vitro stimulation of recombinant Keratins and injection yields the same biological effect of systemic hair growth, DP gene expression, and cell condensation.

(c) While authors suggest reduced DP proliferation in response to keratin stimulation, authors should validate proliferative DP cells throughout the aggregates using Ki67 and apoptosis using caspase 3, to assess viability and toxicity.

(d) Importantly, I'm concerned that the condensed DP phenotype is not consistent throughout the paper, and in times, looks like an apoptotic aggregate of cells rather than cell condensation. It is unclear why the data presented in Fig. 1H looks like one big cell rather than a condensation of cells? The paper will benefit from demonstrating a small inset to convey nuclear vs cytoplasmic expression of the selected markers.

4. Authors intriguingly proposed that keratin released by apoptotic cells during the destructive hair cycle stage have a role in dermal cell condensation and hair growth, though the temporal consequences and the underlying mechanism is still silent. It seems that 2 weeks post keratin delivery, hair growth has reached full anagen (fig 1). In that case, it is unclear how a prolonged or short telogen phase is regulated by catagen initiation, concerning keratin delivery and DP cell condensation.

5. Authors suggest that as opposed to DP cells, ORS cells stimulated with Keratin are more proliferating (Fig. 3). Authors should confirm ORS vs DP cell proliferation with the appropriate markers (ki67/Edu), and clarify the cell-specific proliferative phenotype they observed.

6. ORS cell apoptosis following TGFb2 stimulation should be quantified for TUNEL, annexin, and caspase 3. Generally, authors have the tendency to present immunofluorescent images without proper quantifications and statistics, which should be revised throughout the paper.

7. Importantly, authors should clearly state and show how they quantified the different stages of the hair follicle. In general, hair follicles are synchronized across the hair coat in mice. It is therefore unclear how in control mice, the authors identify ~50% of the hair follicles in Telogen while ~30% are at catagen. Did the authors analyze hair stages at different locations within the back skin? In that case, how authors controlled for consistency within the different experimental groups? Additional concern stems from inconsistency between the phenotype presented image in Fig.1A-B (all follicles are at full anagen), and the quantification in Fig. 1C-D (only 30% of HF are at anagen?). A similar concern for data presented in Fig. 7.

8. A major concerning issue that is missing in this paper is how fragmented keratin stimulates DP cell condensation and HG formation. I find the data of HG formation to be extremely weak, not supported by data, and was not appropriately controlled. If indeed fragmented keratin filament induces DP cell condensation in-vivo, it is unclear how this fits into normal hair cycle progression and HG formation.

Authors should reconcile this discrepancy experimentally using in-vivo mice hair cycle progression. Equally unclear throughout the study and in light of their initial experiments, authors should confirm DP gene expression and cell condensation in an in vivo mice model of keratin injection.

9. The notion that Keratin released by apoptotic cells during hair regression has a role in dermal cell condensation and priming for the new hair cycle is intriguing. It will be of interest to show free Keratin in the extracellular space during catagen, and whether indeed inhibition of TGFb2 signaling can attenuate keratin secretion. Importantly, authors should also reconcile the consequence of events. It is unclear how keratin delivery during catagen will stimulate hair growth, particularly when the following stage might be a prolonged telogen, where hair does not grow.

10. The expression of KRT31/KRT34 should be evaluated in situ using immunofluorescence following intravascular lipofectamine-mediated delivery of KRT31/KRT34 siRNAs. Off targets should also be evaluated and the reduction of fragmented KRT31/KRT34 during catagen should be demonstrated. The authors should also consider other methods of assessing the role played by KRT31/KRT34 on hair growth, rather than just staging the hair cycle. For example, do the cells survive better upon loss of KRT31/KRT34? Do they proliferate more, or with a shorter cell cycle? Do follicles experience an accelerated catagen?

11. Curiously, authors show that KRT31/KRT34 silencing during telogen is sufficient to inhibit hair growth. It is unclear, however, whether keratin release happens independently of catagen and cell death. If the suggested working model is correct, hair growth delay will be noticeable only on the following hair cycle and not immediately, as cell death rarely happens during telogen to anagen transition. In that case, KRT31/KRT34 silencing might delay hair progression, irrespective of keratin release to the extracellular space. Without having rigorous experimental validations of their findings, the reader cannot evaluate the extent to which the conclusions drawn are valid.

12. Finally, the paper is not clearly written and it is hard to follow the narrative of how TGFb induces keratin secretion & fragmentation and the subsequent effect on HG formation and DP cell condensation. Few methodological approaches are not adequately listed and explained and many of the presented data are not quantified. While the topic of this study is of interest to a broad readership, a number of experiments to document the author's claims are lacking appropriate controls and hence many of the conclusions made are not justified (i.e. hair germ formation, keratin release, and KRT31/KRT34 silencing experiments, etc).

Minor comments:

1. Fig. 1B- it is hard to evaluate hair morphology and staging based on these sections. Authors should align follicles for better horizontal visualization and assessment of their data.
2. Fig. 1H. A few panels represent the same cells as a separate panel /experiment which makes it confusing and inconsistent with the data. If staining was done on the same set of fixed cells, the represented data should truthfully illustrate it. Additionally, the presented data makes it very hard to determine which marker is nuclear and which cytoplasmic. The authors should combine the staining with a membrane marker (same for Sup. Fig2C, etc and provide insets, when appropriate).
3. Immunofluorescence images in Fig5 are not presented with good quality and it is hard to assess the merit of the presented results. Generally, image quality should be improved throughout the paper.
4. Quantification and classification of hair follicle stages in the back skin is unclear and not supported by data. During homeostatic conditions, the mice hair coat is largely synchronized. The images presented do not support the conclusion of a mixed telogen, anagen, and catagen stages in the hair coat all at the same time (Fig. 1 and Fig. 7).
5. The authors are using human DP and ORS cell lines. it should be noted how these cells were isolated and to include markers that distinguish these cells.
6. The current paper could benefit tremendously from professional language editing.

Manuscript #: COMMSBIO-20-3162

Title: Keratin is not only a Structural Protein in Hair: Keratin-mediated Hair Growth

A List of Changes

As reviewer's comments, all comments are answered in a list of changes, and all changes in our manuscripts are made. Some of our answers are repeated for reviewer's similar comments, several supplementary figures are newly added, and some figures are provided only to answer the comments in this letter.

Reviewer 1's Comments: In this manuscript, the authors sought to uncover the biological function of keratin in hair growth. They demonstrated that intradermal injection of Keratin could stimulate the hair growth, while silencing keratin in mice resulted in a marked suppression of anagen follicle formation. Through a group of experiments *in vitro*, they proposed that the keratin exposure, caused by TGF β -induced ORS apoptosis, is essential and critical for dermal papilla cell condensation and hair germ formation. For the most part, their techniques and approach were quite thorough and their controls were appropriate for the bulk of their experiments. Overall, the findings in the manuscript are interesting. However, in its present form, there are a few issues that I believe the authors should address before final publication.

1. Reviewer's Comment:

- A plenty of experiments *in vitro* showed the significant effects of keratin in DP condensation and hair germ formation, however is the hair growth promotion by keratin in the mouse model really via its impact on DP/ORS, considering the hair follicle formation, including the DP and bulge, has been finished in adult mice. How could keratin promote a new telogen-anagen transition seems unclear since the *in vitro* experiments clarified the function of keratin in anagen-catagen transition? Could keratin promote the activation of hair follicle stem cell?
- In figure 1, the data showed an increased total number of hair follicle with the treatment of keratin, is there any influence of keratin on epidermal stem cell differentiation to form new hair follicle?

1. Answer:

- Your comment is appreciated. ORS cells in the bulge region contain hair follicle stem cells expressing different markers according to their differentiation stages [Development (2015)]. As your kind comments, molecular and gene expressions indicating stem cell activation and differentiation were evaluated additionally in ORS culture in the presence of keratin by immunocytochemical staining and real time-qPCR. With the loss of CD34 expression, P-cadherin expressing cell population was found in keratin-treated ORS cell culture, which was supported by the emergence of Lgr5-positive cell population during ORS cell culture in the presence of keratin [Supplementary Fig. 17]. Such upregulated Lgr5 expression was also observed in the skin section of keratin-injected mice, which proved by *in situ* RNA hybridization [Supplementary Fig.2]. In addition, several gene expressions indicating further differentiation, such as matrix and shaft differentiation, of stem cells were found to be upregulated in keratin-treated ORS cells [Supplementary Fig.18]. As your kind

comments, the manuscript was re-written more scientifically for better understanding and several additional supplemental figures and references were added more to support.

The added and corrected sentences in the section of result and discussion are as follows;

The corrected paragraphs from line 99 to line 116 and from line 146 to line 163 in the section of result are as follows;

“First, we performed *in vivo* experiments in mice to evaluate keratin-mediated hair growth by injecting hair-derived keratin into the hair-removed dorsal skin area. The dorsal area of C57BL/6 mice was shaved with an animal clipper, and 0.5% (w/v) and 1.0% (w/v) keratin were injected. Twenty-eight days after keratin injection, hair growth and hair follicle formation in keratin-injected C57BL/6 mice were analyzed for the number and stage of hair follicles. We found that hair growth and the formation of anagen follicles were promoted (Fig. 1A), compared to non-treated mice (Fig. 1B-D). Only a single injection of keratin resulted in much higher hair growth compared to the control, and almost equivalent hair growth compared to minoxidil, applied every day for 28 days. In addition, *in situ* RNA hybridization showed a significant increase in the Lgr5-positive cell populations in the lower bulge region and hair follicles after keratin injection (Supplementary Fig. 2). Such keratin-mediated hair growth was confirmed in a separate mouse study, and the promoting effect of keratin injection on hair growth was verified. There were no significant differences in the number of hair follicles formed between mice injected with keratin and those treated with minoxidil (Supplementary Fig. 3A-C). However, the number of anagen follicles and the size of hair follicles were found to be increased in keratin-injected mice compared to minoxidil-treated mice (Supplementary Fig. 3D-F). Keratin injection-mediated hair growth was observed throughout the surface of the back skin of mice, which was due to the dispersion of keratin solution after injection, and the injected keratin remained up to 2 weeks after injection (Supplementary Fig 4).”

“ORS cells formed colonies within a few hours of exposure to keratin, and subsequently formed strand-like extended structures by day 3 (Fig. 3A, B). The proliferation of ORS cells was suppressed upon exposure to keratin, as indicated by decreased BrdU incorporation. High β -catenin expression, known to occur during migration, further differentiation of stem cells in the ORS region⁶, and a local cell population expressing P-cadherin, known to be a marker of secondary HG formation^{20,21}, were observed along with extended structures in keratin-treated ORS cells (Fig. 3C, Supplementary Fig. 16). In addition, keratin-treated ORS cells showed lower expression levels of CD34 than untreated ORS cells but maintained high levels of SOX9 expression (Fig. 3C). With reduced expression of CD34, the Lgr5+ cell population, which is known to participate in hair germ formation²²⁻²⁴, emerged in keratin-treated ORS cells (Supplementary Fig. 17). In addition, real-time qPCR analysis showed upregulation of the expression of various genes, such as SOX9, EDAR, FOXN1²⁵, MSX2²⁶, SHH²⁷, EFNB1, ITGA6²⁸ and β -catenin, indicating matrix and shaft differentiation of stem cells from the ORS bulge region (Supplementary Fig. 18). Furthermore, RNA sequencing analysis of keratin-treated ORS cells revealed upregulation of mRNA expression levels of acidic hair keratins, mainly KRT31, KRT33B, KRT34, and KRT37 (Supplementary Fig. 19A). We also observed increased expression of KRT34 and β -catenin proteins (Supplementary Fig. 19B). These findings imply that hair keratin-mediated alterations in the protein and gene expression profiles indicate germ formation and further differentiation of ORS cells.”

The corrected paragraphs from line 303 to line 307 in the section of discussion are as follows;
“With the loss of CD34 expression, a Lgr5-expressing cell population was generated in keratin-treated ORS cells. CD34-positive stem cells have been reported to convert directly into P-cadherin-expressing HG cells³³, and CD34-positive stem cells migrate downward to the lower bulge region and convert to Lgr5-positive stem cells, which participate in HG formation during the telogen stage²⁴.”

[References]

22. Krieger, T. & Simons, B.D. Dynamic stem cell heterogeneity. *Development* **142**, 1396-1406 (2015)
23. Rompolas, P. & Greco, V. Stem cell dynamics in the hair follicle niche. *Sem. Cell. Dev. Biol.* **25-26**, 34-42 (2014)
24. Jaks, V. et al. Lgr5 marks cycling, yet long-lived, hair follicle stem cells. *Nat. Gene.* **40**, 1291-1299 (2008)
25. Rezza, A. et al. Signaling networks among stem cell precursors, transit-amplifying progenitors, and their niche in developing hair follicles. *Cell. Rep.* **14**, 3001-3018 (2016)
26. Wang, A.B., Zhang, Y.V. & Tumber, T. Gata6 promotes hair follicle progenitor cell renewal by genome maintenance during proliferation. *EMBO* **36(1)**, 61-78 (2017)
27. Lim, C.H. et al. Hedgehog stimulates hair follicle neogenesis by creating inductive dermis during murine skin wound healing. *Nat. Com.* **9**, 4903 (2018)
28. Tumber, T. et al. Defining the epithelial stem cell niche in skin. *Science.* **303(5656)**, 359-363 (2004)

2. Reviewer's Comment:

- In figure S15, the author showed a high expression level of caspase 6, P-cadherin and KRT 34 on P8 of mice, what is the intact expression pattern of KRT34 in hair follicle development and hair cycle?

2. Answer:

- Your comment is appreciated. As your kind comments, molecular expressions of KRT34 was evaluated in hair follicles at different stages, which was proved by immunohistological staining of the longitudinal and cross sections of normal mouse skin. At anagen stage, major KRT34 expressions were found in outer root sheath region of growing hairs and in cortex of hair at catagen stage. This figure is only provided to answer the reviewer's comment.

3. Reviewer's Comment:

- In Figure 1B-F, it will be more clear to identify the stage of hair follicles and more precise to quantify the number and size of HFs with a longitudinal section;

3. Answer:

- Your comment is appreciated. As your kind comments, the histological images and data to analyze the numbers and stages of hair follicles formed in keratin-injected mouse were rearranged by exchanging histological data in figure 1 with those in supplementary figure 1. The rearranged figure 1 shows images of both longitudinal sections and cross sections to clarify the stages of hair follicle more.

4. Reviewer's Comment:

- Some scale bar seems to be inconsistent with the real presentation, such as Fig4B;

4. Answer:

- Your comment is appreciated. As your kind comments, scale bars were adjusted. The images were taken at different magnifications, and hence all scale bars were adjusted to indicate 200 μ m.

5. Reviewer's Comment:

- In Figure 4C, the authors showed that the coculture of TGF β 2 treated ORS with DP cells can promote the DP condensation, without a negative control of naive ORS.

5. Answer:

- Your comment is appreciated. As your kind comments, the microscopic and immunocytochemical images of DP cell coculture on control (non-treated) ORS cell layer were added more in supplementary figure 22A to compare with DP cells cocultured on TGF β 2 treated ORS cell layer.

6. Reviewer's Comment:

- Fig 6B showed an increase of TGF β in conditioned medium after immunodepletion, is there any impact of TGF β on DP cell?

6. Answer:

- Your comment is appreciated. The increase of TGF β 2 in conditioned medium could be due to the exogenous TGF β 2 added to induce ORS cell apoptosis. In addition, as your kind comments, the impact of TGF β 2 on DP cell condensation was tested, and the treatment of TGF β 2 only did not show any influence on DP cell condensation, which result was added in supplementary figure 25. As your kind comments, some paragraphs were added to explain the result.

The added paragraphs from line 219 to line 221 in the section of result are as follows;

“The increase in TGF β 2 in the conditioned medium was derived from the exogenous TGF β 2 added to induce apoptosis of ORS cells, and TGF β 2 did not influence DP cell condensation by itself (Supplementary Fig. 25).

7. Reviewer’s Comment:

- Figs15A and Fig7D, upper panel and lower panel of group siRNA, why the hair follicle size and stage showed different, while one in anagen another in telogen

7. Answer:

- Your comment is appreciated. Upper panel and lower panel of siRNA-treated group (Supplementary Fig. 15A) are different magnifications, lower panel was added to show overall images for KRT34 expressions and hardly to observe KRT expressing region in siRNA treated group. In addition, the stages of hair follicles were counted with several tissue sections, and the graphs to show percentage of tissue sections containing anagen stage was provided in Fig 7 C. In this study, KRT31/34-knockdown mouse model study showed keratin could derive anagen follicle formation. Distinctly reduced anagen follicle formation in KRT31/34-knockdown mouse and recovery of anagen follicle formation in KRT31/34-knockdown and exogenous keratin-injected mouse suggested that keratin might participate in anagen follicle formation. As shown in Fig 7C, anagen hair follicle could not be easily observed in siRNA treated group, and hence representative images to show P-cadherin and β -catenin expressions were selected even at different hair cycle stages. In addition, supplementary figure 30-32 were added newly to support the results.

8. Reviewer’s Comment:

- In FigureS11, the tunnel staining seems to be inconsistent with the description;

8. Answer:

- Your comment is appreciated. As your kind comments, the contrast of picture in Supplementary Figure 23 was adjusted to show the TUNEL staining more clearly. The order of supplementary figures was rearranged due to several newly added supplementary figures.
-

9. Reviewer’s Comment:

- In the method part, could the repeated hair shave with an electric clipper synchronize the hair follicle cycle? Is there any changes in hair cycle with the application of commercial hair removal cream

9. Answer:

- Your comment is appreciated. The difference of two mouse experiments (Fig 1 and Supplementary Fig 1.) is the use of hair removal cream, and the accelerated hair growth in a mouse experiment might be due to the use of hair removal cream which promotion effect on hair follicle formation is reported

[Korean J. Physiol. Pharmacol. (2021), Physiol. Zool. (1952)].

[References]

Tsai, P. *et al.* Depilatory creams increase the number of hair follicles, and dermal fibroblasts expressing interleukin-6, tumor necrosis factor- α , and tumor necrosis factor- β in mouse skin. *Korean J Physiol. Pharmacol.* **25(6)**. 497-506. (2021)

Rauch, H. Effects of topical applications of chemical agents on hair development. *Physiol. Zool.* **25**. 268-272. (1952).

Reviewer 2's Comments: In this paper, Seong Yeong An et al. showed that keratin has the ability to promote hair growth when it was injected intradermally to mouse skin. The author's findings are somewhat surprising, but they tried to provide different lines of evidences to make their points. Although their finding is interesting, I have several concerns to be addressed.

1. Reviewer's Comment:

- In Figure 1A, they showed clear differences between control and Minoxidil, or Keratin groups in their hair growth at 2 weeks. In supplemental Figure 1A, such changes were not seen in day 14, and started to appear at day 28 when control group also started hair growth. What is the difference between Figure 1A and supplemental Figure 1A? Is this single injection vs multiple treatment? (Nice to show experimental schedule in Figure)

1. Answer:

- Your comment is appreciated. The different hair growth activity might be related to individual variation of mice and experimental environment. Hence, in this study, to confirm keratin-mediated hair growth, separated individual mouse experiments were done separately, and in spite of variation in hair growth period, both mouse experiment showed the same tendency in keratin or minoxidil-mediated hair growth. In addition, the difference of two mouse experiments (Fig 1 and Supplementary Fig 3.) is the use of hair removal cream, and the accelerated hair growth in a mouse experiment might be due to the use of hair removal cream which promotion effect on hair follicle formation is reported [Korean J. Physiol. Pharmacol. (2021), Physiol. Zool. (1952)]. The order of supplementary figures was rearranged due to several newly added supplementary figures.

[Reference]

Tsai, P. *et al.* Depilatory creams increase the number of hair follicles, and dermal fibroblasts expressing interleukin-6, tumor necrosis factor- α , and tumor necrosis factor- β in mouse skin. *Korean J Physiol. Pharmacol.* **25(6)**. 497-506. (2021)

Rauch, H. Effects of topical applications of chemical agents on hair development. *Physiol. Zool.* **25**. 268-272. (1952).

- In this study, intradermal injection of keratin was done once, and minoxidil treatment was done repeatedly five days in a week during the whole mouse experiment. As your kind comments, experimental schedule for mouse experiment was illustrated in Fig. 1A and Supplementary Fig. 3A.
-

2. Reviewer's Comment:

- In Figure 7B, control mice grow hair after 2 weeks, whereas control mice in Figure 1A didn't. Is this same experimental condition? How can authors explain this discrepancy?

2. Answer:

- Your comment is appreciated. The different hair growth activity might be related to individual variation of mice and housing condition. It has been known that hair growth could be influenced by housing condition such as the number of mouse in a cage and that stressful condition induced barbering to influence hair growth activity [*J. Am. Ass. Lab. Ani. Sci.* (2011), *Nature.* (2021)] Hence, in this study, to confirm keratin-mediated hair growth, separated individual mouse experiments were done separately, and in spite of variation in hair growth period, both mouse experiment showed the same tendency in keratin or minoxidil-mediated hair growth.

[References]

Bechard, A., Meagher, R., & Mason G. Environmental enrichment reduces the likelihood of alopecia in adult C57BL/6J Mice. *J. Am. Ass. Lab. Ani. Sci.* **50(2)**, 171-174 (2011)

Choi, S. et al. Corticosterone inhibits GAS6 to govern hair follicle stem-cell quiescence. *Nature.* **592**, 428-432 (2021)

3. Reviewer's Comment:

- In Figure 1A, why HF "size" is increased? Is it simply a hair cycle differences (anagen follicle vs telogen follicle) or due to proliferation of bulge/hair germ (increase of cell number)? Authors show cross-section of skin in Figure1 but longitudinal sections would be better to define their histology.

3. Answer:

- Your comment is appreciated. As your kind comments, the histological images and data to analyze the numbers and stages of hair follicles formed in keratin-injected mouse were rearranged by exchanging histological data in figure 1 with those in supplementary figure 1. The rearranged figure 1 shows images of both longitudinal sections and cross sections to clarify the stages of hair follicle more.
- As your kind comment, we have more evaluated the keratin-mediated stem cell activation for hair germ formation. Lgr5+ cell population is known to be generated from CD34+ stem cells located in bulge region of ORS during telogen-anagen transition [*Development* (2015), *Sem. Cell. Dev. Biol.*], and Lgr5 expressing cell progeny moves downward to form hair germ and new hair follicle finally, which controls the new cycling of anagen hair follicle [*Nat. Gene.*2008]. In this study, Lgr5+ cell population could be generated from ORS cells when exposed to keratin [Supplementary Fig. 17], and also increased Lgr5+ cells in the lower bulge region were found in keratin injected mice [Supplementary Fig.2]. In addition, gene expression profile of ORS cells was also evaluated when exposed to keratin in vitro, and highly upregulated mRNA expressions of several markers, such as

SOX9, EDAR, FOXN1, MSX2, SHH, EFNB1, ITGA6 and β -catenin, indicating stem cell activation and differentiation was observed [Supplementary Fig. 18]. In contrast to ORS cell differentiation, ORS cell proliferation was reduced upon keratin treatment [Supplementary Fig. 15A]. Several supplementary figures were newly added to support keratin-mediated stem cell activation, and the manuscript was re-written more scientifically for better understanding and references were added more to support.

- The corrected paragraphs from line 106 to line 108, from line 146 to line 148 and from line 153 to line 158 in the section of result are as follows;

“In addition, *in situ* RNA hybridization showed a significant increase in the Lgr5-positive cell populations in the lower bulge region and hair follicles after keratin injection (Supplementary Fig. 2).”

“ORS cells formed colonies within a few hours of exposure to keratin, and subsequently formed strand-like extended structures by day 3 (Fig. 3A, B). The proliferation of ORS cells was suppressed upon exposure to keratin, as indicated by decreased BrdU incorporation.”

“With reduced expression of CD34, the Lgr5+ cell population, which is known to participate in hair germ formation²²⁻²⁴, emerged in keratin-treated ORS cells (Supplementary Fig. 17). In addition, real-time qPCR analysis showed upregulation of the expression of various genes, such as SOX9, EDAR, FOXN1²⁵, MSX2²⁶, SHH²⁷, EFNB1, ITGA6²⁸ and β -catenin, indicating matrix and shaft differentiation of stem cells from the ORS bulge region (Supplementary Fig. 18).”

[References]

22. Krieger, T. & Simons, B.D. Dynamic stem cell heterogeneity. *Development* **142**, 1396-1406 (2015)
 23. Rompolas, P. & Greco, V. Stem cell dynamics in the hair follicle niche. *Sem. Cell. Dev. Biol.* **25-26**, 34-42 (2014)
 24. Jaks, V. et al. Lgr5 marks cycling, yet long-lived, hair follicle stem cells. *Nat. Gene.* **40**, 1291-1299 (2008)
 25. Rezza, A. et al. Signaling networks among stem cell precursors, transit-amplifying progenitors, and their niche in developing hair follicles. *Cell. Rep.* **14**, 3001-3018 (2016)
 26. Wang, A.B., Zhang, Y.V. & Tumber, T. Gata6 promotes hair follicle progenitor cell renewal by genome maintenance during proliferation. *EMBO* **36(1)**, 61-78 (2017)
 27. Lim, C.H. et al. Hedgehog stimulates hair follicle neogenesis by creating inductive dermis during murine skin wound healing. *Nat. Com.* **9**, 4903 (2018)
 28. Tumber, T. et al. Defining the epithelial stem cell niche in skin. *Science.* **303(5656)**, 359-363 (2004)
- Hair follicle size is reported to be determined by the number of dermal papilla cells [*Development* (2013), *J. Invest. Dermatol.* (1999)], but, in this study, proliferation of DP cells or ORS cells was inhibited [Supplementary Fig. 6A and 15A]. Such hair follicle size is controlled not only by cell number but also by other microenvironment such as extracellular matrix volume and vessel formation. It was reported that VEGF induced DP cell proliferation and endothelial-mediated vascular tube formation around hair follicle could influence hair follicle size [*J. Clin. Invest.* (2001), *Exp. Cell. Res.* (2012)]. In this study, we have focused on the biological function of keratin for dermal papilla condensation and germ formation in hair growth, and other study on extracellular interaction of

keratin with other cell types such as endothelial cell, sebocyte and adipocyte which can influence hair growth has been on-going. We found enhanced keratin-mediated tubulogenesis of human microvascular endothelial cells. When human microvascular endothelial cells were treated with keratin in the culture on tissue culture plate, highly improved vascular tube formation was observed and characterized by immunocytochemical staining with anti VEGF-receptor 2 and CD31, which might derive microenvironmental change to be favorable for the increase of hair follicle size and hair growth, and we are preparing for next paper submission about extracellular interaction of keratin with other cell types.

[References]

Chi, W., Wu, E., Morgan, B.A. Dermal papilla cell number specifies hair size, shape and cycling and its reduction causes follicular decline. *Development* **140**, 1676-1683 (2013)

Elliot, K., Stephenson, T.J., Messenger, A.G. Differences in hair follicle dermal papilla volume are due to Extracellular matrix volume and cell number: Implications for the control of hair follicle size and androgen responses. *J. Invest. Dermatol.* **113**, 873-877 (1999)

Yano, K., Brown, L.E., Detmar, E. Control of hair growth and follicle size by VEGF-mediated angiogenesis *J. Clin. Invest.* **107**, 409-417 (2001)

Wei Li., et al. VEGF induces proliferation of human hair follicle dermal papilla cells through VEGFR-2-mediated activation of ERK. *Exp. Cell. Res.* **318**, 1633-1640 (2012)

This figure is only provided to answer the reviewer's comment.

[data redacted]

4. Reviewer's Comment:

- Did authors detect any changes in morphology of DP (DP condensation) and their gene expression in tissue upon keratin injection (related to Figure 1)? I see staining data in Figure S15, siRNA + Keratin injection. How about Keratin injection only?

4. Answer:

- Your comment is appreciated. As your kind comments, fluorescent keratin was developed by conjugating fluorescent dye (Cy5.5), and DP cell condensation was observed in the skin section of fluorescent-keratin injected mouse. As shown in the below figure, condensed DP morphology could be observed at day 7 of injection. The below figure is provided only to answer the reviewer's comment.

- As your kind comments, gene expression profile of keratin-treated DP cells was evaluated, and highly upregulated mRNA expressions of several markers such as CD133, Sox2, Corin, Shh, versican, β -catenin, BMP6, FGF7 and FGF10 were detected in keratin-treated DP cell culture, which result is newly added as Supplementary Figure 13, and sentences demonstrating the result of gene expression profile are added in the section of result.

The paragraphs from line 137 line 140 in the section of result are as follows;

“Real-time qPCR analysis showed upregulation of various genes, such as CD133, SOX2, corin, SHH, versican, β -catenin, BMP6, FGF7, and FGF10, known as a molecular identity signature reflecting the hair-inductive property of DP cells¹⁶⁻¹⁹ (Supplementary Fig. 13).”

[References]

16. Driskell, R.R., Giangreco, A., Jensen, K.B., Mulder, K.W. & Watt, F.M. SOX2-positive dermal papilla cells specify hair follicle type in mammalian epidermis. *Development* **136**, 2815-2823 (2009).
 17. Yamauchi, K. & Kurosaka, A. Inhibition of glycogen synthase kinase-3 enhances the expression of alkaline phosphatase and insulin-like growth factor-1 in human primary dermal papilla cell culture and maintains mouse hair bulbs in organ culture. *Arch. Dermatol. Res.* **301**, 357-365 (2009).
 18. Rendl, M., Polak, L. & Fuchs, E. BMP signaling in dermal papilla cells is required for their hair follicle-inductive properties. *Genes Dev.* **22**, 543-557 (2008).
- Your comment is appreciated. In this study, keratin-mediated hair growth was tested in separate mouse models by intradermal injection of keratin into mice in comparison with minoxidil (Fig. 1 and Supplementary Fig.3). Following *in vitro* cell assay and mouse study, *in vitro* and *in vivo* KRT31/KRT34-knockdown study were designed to evaluate the influence of endogenous keratin expression on hair growth activity by blocking endogenous keratin expression. KRT31/34-knockdown mouse model study showed keratin could derive anagen follicle formation. Distinctly reduced anagen follicle formation in KRT31/34- knockdown mouse and recovery of anagen follicle formation in KRT31/34-knockdown and exogenous keratin-injected mouse suggested that keratin might participate in anagen follicle formation.

5. Reviewer's Comment:

- Overall, the authors need more careful description and experimental data of histological and marker changes of skin after keratin treatment *in vivo* to link with their findings *in vitro*. My main concern is what they found *in vitro* experiments with human cells are really happening *in vivo* mouse tissue.

5. Answer:

- Your comment is appreciated. As your kind comments, the manuscript was re-written more scientifically for better understanding, and this study was done as a pilot study to suggest a potent biological role of keratin in hair growth, which was majorly based on *in vitro* cell assay. As your kind comments, the animal study to link our *in vitro* finding with *in vivo* mechanism in hair growth should be necessary. Hence, we have developed recombinant keratin protein and will show the biological mechanism using genetically modified ORS cells equipped with an on-off expression system for

keratin expression and a xenograft mouse model to determine the *in vivo* mechanisms in another continuous study, which was demonstrated in the section of discussion.

The paragraphs from line 379 line 384 in the section of discussion are as follows;

“These studies indicate that spatiotemporal apoptosis during the hair cycle can be an essential process in controlling hair regeneration, and our findings show that keratin can be an important factor influencing hair growth. However, these pilot observations require follow-up studies of hair keratin expression and its apoptosis-related release during the hair cycle using genetically modified ORS cells equipped with an on-off expression system for keratin expression and a xenograft mouse model to determine the *in vivo* mechanisms.”

Reviewer 3’s Comments: In their manuscript, entitled: “Keratin is not only a Structural Protein in Hair: Keratin-mediated Hair Growth”, Seong Yeong An et al., aim to identify the molecular role of keratin filaments in hair follicle growth and maintenance. By intradermally injecting human hair-derived keratin to mice skin, authors demonstrated accelerated hair growth and a precocious hair formation. Mechanistically, secretion of TGFb2 by the supportive dermal cells during the hair destruction phase stimulates epithelial cell death and release of keratin to the extracellular space, leading to dermal cell condensation and initiation of an additional cycle of hair growth. The authors then show that *in vivo* silencing of keratins inhibits anagen entrance and hair follicle growth.

Keratin intermediate filaments are major protein constituents in epithelial cells and the hair follicle. Although highly abundant, the exact nature of communication and mechanistic properties of keratins in hair follicle biology is still scant. Equally unclear is which keratin filament participate in basic processes such as cell differentiation and tissue homeostasis. Understanding the function of keratin produced by the epithelial hair cells may be of high medical interest for hair regeneration. Authors present an intriguing idea by which catagen-mediated apoptotic cell release of keratin can promote hair growth via the control of the supportive dermal compartment, which may extend our knowledge on the molecular basis of why cell death during hair destruction is essential for homeostatic maintenance of hair growth over time.

With that in mind, the current study lacks clarity, fails to provide essential controls and many of the listed claims have not been examined experimentally. The manuscript is not clearly organized into a logical storyline. Together, while the concepts put forth in the manuscript are of interest, the current report would be much improved by providing a clear narrative and more conclusive evidence to support the suggestions made by the authors.

1. Reviewer’s Comment:

- One major concern is whether intradermal injection of isolated human keratin was thoroughly controlled in their experiments. It is unclear how far distance the injected keratin is within the skin following injection. If Keratin promotes DP cell condensation, as the authors suggested, it is unclear and completely unexplained why the effect is not locally restricted to the injection site and instead, spread throughout the back skin of the animals, as presented by a profound hair growth effect? Authors should evaluate how far distance keratin is in the back skin following injection, and what is

the kinetics of hair growth at the site of injection as well as at distant areas in the back skin

1. Answer:

- Your comment is appreciated. As your kind comments, the dispersion of keratin after intradermal injection was tested by injecting fluorescent dye-conjugated keratin into the back skin of severe combined immunodeficiency (SCID) mouse. Keratin molecules were found to disperse soon after injection, and the dispersion of keratin could be detected throughout the whole back skin after 1 day of injection. As your kind comments, the result is added as a Supplementary Figure 4, and some paragraphs were added in the section of result.

The added paragraphs from line 113 line 116 in the section of result are as follows;

“Keratin injection-mediated hair growth was observed throughout the surface of the back skin of mice, which was due to the dispersion of keratin solution after injection, and the injected keratin remained up to 2 weeks after injection (Supplementary Fig 4).”

2. Reviewer’s Comment:

- Authors show in Fig 1A a strong hair growth phenotype 2 weeks after a single intradermal injection of Keratin. It is unclear why results presented in Sup. Fig 1, are profoundly different and demonstrating hair growth phenotype only after 4 weeks of treatment (while comparing same keratin concentration as well as minoxidil treatment)?

2. Answer:

- Your comment is appreciated. The different hair growth activity might be related to individual variation of mice and experimental environment. Hence, in this study, to confirm keratin-mediated hair growth, separated individual mouse experiments were done separately, and in spite of variation in hair growth period, both mouse experiment showed the same tendency in keratin or minoxidil-mediated hair growth. In addition, the difference of two mouse experiments (Fig 1 and Supplementary Fig 3.) is the use of hair removal cream, and the accelerated hair growth in a mouse experiment might be due to the use of hair removal cream which promotion effect on hair follicle formation is reported [Korean J. Physiol. Pharmacol. (2021), Physiol. Zool. (1952).]. The order of supplementary figures was rearranged due to several newly added supplementary figures.

[Reference]

Tsai, P. *et al.* Depilatory creams increase the number of hair follicles, and dermal fibroblasts expressing interleukin-6, tumor necrosis factor- α , and tumor necrosis factor- β in mouse skin. *Korean J Physiol. Pharmacol.* **25(6)**. 497-506. (2021)

Rauch, H. Effects of topical applications of chemical agents on hair development. *Physiol. Zool.* **25**. 268-272. (1952).

3. Reviewer’s Comment:

- Dermal cell condensation:
 - (a) Authors should confirm that other factors, known to regulate DP cell condensation are not present

in the keratin-extracted solution (i.e Fgf20, etc).

- (b) Similarly, authors should confirm that an in-vitro stimulation of recombinant Keratins and injection yields the same biological effect of systemic hair growth, DP gene expression, and cell condensation.
- (c) While authors suggest reduced DP proliferation in response to keratin stimulation, authors should validate proliferative DP cells throughout the aggregates using Ki67 and apoptosis using caspase 3, to assess viability and toxicity.
- (d) Importantly, I'm concerned that the condensed DP phenotype is not consistent throughout the paper, and in times, looks like an apoptotic aggregate of cells rather than cell condensation. It is unclear why the data presented in Fig. 1H looks like one big cell rather than a condensation of cells? The paper will benefit from demonstrating a small inset to convey nuclear vs cytoplasmic expression of the selected markers.

3. Answer:

- (a) Your comment is appreciated. As your kind comments, first we characterized the presence of FGF20, known as a potent growth factor to induce dermal papilla condensation [Genes. Dev. (2013)], in the extracted keratin by western blot analysis with anti-human FGF20 antibody. The western blot analysis showed no presence of FGF20 in the extracted keratin used in this research. We also evaluated the possibility of masking the epitope of FGF20 by interacting with keratin molecule upon reacting with anti-human FGF20 antibody. In the mixture of FGF20 and keratin, FGF20 was detected clearly in western blot analysis, which indicates no hindrance of keratin molecule in the reactivity of anti-human FGF20 antibody against rhFGF20. In addition, MALDI-TOF spectroscopic analysis showed the pure identity of keratin used in this research. As your kind comments, the result is added as a supplementary figure 34, and some paragraphs and a reference were added in the section of discussion.

The corrected paragraphs from line 307 to line 313 in the section of Discussion are as follows;

“In addition to keratin-mediated ORS cell differentiation, treatment with exogenous keratin induces DP cell condensation. Epithelial fibroblast growth factor 20 (Fgf20) is known to control dermal condensate morphogenesis³⁶; hence, we evaluated the presence of potent growth factors, such as FGF20, in hair-derived keratin to influence cellular behavior. Western blot analysis showed no presence of FGF20 in the keratin extract used in this study, and application of MALDI-TOF mass spectroscopy indicated that the extracted protein was purely keratin (Supplementary Fig. 34).”

[Reference]

36. Huh, S.H. et al. Fgf20 governs formation of primary and secondary dermal condensations in developing hair follicles. *Gene. Dev.* **27**, 450-458 (2013)

- (b). Your comment is appreciated. As your kind comments, the biological function of keratin in interaction with dermal papilla cell was also evaluated with recombinant KRT34. rhKRT34 (recombinant human KRT34) was made from E.coli-based protein production process, and the extracted rhKRT34 was characterized by SDS-PAGE, MALDI-TOF and glycosylation staining assay. As is shown in below figure, rhKRT34 also induced effectively the condensation of dermal papilla cells, and showed inductive activity of DP cell condensation even at much lower concentration such as 0.001(w/v)% in comparison with the extracted hair keratin, which could be expected by considering that hair-derived keratin is composed of heterogeneous keratin subclasses and various

fragments with different molecular weights and some of them could easily aggregate finally to influence solubility and biological function. The research on the hair growth effect of rhKRT is still on going, and we have been collecting data for another paper submission and a patent application. Hence, please excuse we can provide this figure only to answer the reviewer's comment.

MALDI-TOF

Accession	Description	Score	Coverage [%]	# Peptides	# PSMs	# Unique Peptides	# AAs	MW [kDa]	calc. pI
O76011	Keratin, type I cuticular Ha4	209.29	55	29	72	29	436	49.4	5.06

A

Accession	Description	Score	Coverage [%]	# Peptides	# PSMs	# Unique Peptides	# AAs	MW [kDa]	calc. pI
O76011	Keratin, type I cuticular Ha4	827.07	73	46	250	3	436	49.4	5.06
AOA140TA69	Keratin, type I cuticular Ha4	753.71	67	44	227	1	436	49.4	5.06

B

Accession	Description	Score	Coverage [%]	# Peptides	# PSMs	# Unique Peptides	# AAs	MW [kDa]	calc. pI
O76011	Keratin, type I cuticular Ha4	135.84	41	20	50	20	436	49.4	5.06

C

Accession	Description	Score	Coverage [%]	# Peptides	# PSMs	# Unique Peptides	# AAs	MW [kDa]	calc. pI
O76011	Keratin, type I cuticular Ha4	8.15	5	3	4	3	436	49.4	5.06

D

Accession	Description	Score	Coverage [%]	# Peptides	# PSMs	# Unique Peptides	# AAs	MW [kDa]	calc. pI
O76009	Keratin, type I cuticular Ha3-I	4.22	4	2	2	2	404	45.9	4.82

E

[data redacted]

[data redacted]

(c) and (d). Your comment is appreciated. As your kind comments, DP proliferation in the presence of keratin was evaluated by immunocytochemical staining with anti-Ki67 antibody and anti-BrdU antibody. The decrease in proliferation of keratin treated DP cells tested by CCK-8 assay was consistent with the reduced expression of Ki67 and BrdU incorporation. Some DP cells with cell aggregates in the presence of keratin was found to express Ki67, and live/dead staining showed most DP cells in the aggregates were detected to be viable. In addition, to show clear images of DP cell aggregates, magnified images were added as a supplementary figure 7, which show distinct β -catenin expressions in the contact region between DP cells in aggregates. As your kind comments, the result is added as a supplementary figure 6, 7, and 8, and some paragraphs were corrected in the section of results.

The corrected paragraphs from line 127 to line 131 in the section of results are as follows;
“However, the growth of DP cells was suppressed upon exposure to keratin (Supplementary Fig. 5A), and relatively lower Ki67 expression and BrdU incorporation were observed in keratin-treated DP cells (Supplementary Fig. 6B, C). Condensed DP cell aggregates contained cells expressing Ki67, and β -catenin were expressed in contact region between cells within DP cell aggregates (Supplementary Fig. 7), in which DP cells remained viable during culture (Supplementary 8).”

4. Reviewer’s Comment:

- Authors intriguingly proposed that keratin released by apoptotic cells during the destructive hair cycle stage have a role in dermal cell condensation and hair growth, though the temporal consequences and the underlying mechanism is still scant. It seems that 2 weeks post keratin delivery, hair growth has

reached full anagen (fig 1). in that case, it is unclear how a prolonged or short telogen phase is regulated by catagen initiation, concerning keratin delivery and DP cell condensation.

4. Answer:

- Your comment is appreciated. In this study, 6 weeks-aged C57BL/6 mice were used to study keratin-mediated hair growth, and most hair follicles were synchronized in telogen during hair cycle at initial point by repeated shaving. As the reviewer's concern, the hair follicles enter anagen stage and proceed normal hair cycle progress. Our proposed action of keratin might induce DP condensation and 2nd hair germ formation during telogen-anagen transition. During the anagen-catagen transition stage, TGFβ2 is synthesized from DP cells stimulated by dihydrotestosterone, and is spatiotemporally localized in the lower part of the hair bulb at the catagen stage, thus suppressing the proliferation of epithelial cells, but inducing caspase-mediated apoptosis [*J. Dermatol. Sci.* (2004)]. A study using transgenic mice overexpressing an anti-apoptotic gene reported that the inhibition of the apoptotic death of ORS cells during anagen resulted in the early termination of hair follicle stem cell activation and proliferation, while the initiation of a new hair cycle was postponed by inhibiting the apoptotic death of ORS cells during telogen [*EMBO Journal* (1999)]. Lgr5⁺ cell population is known to be generated from CD34⁺ stem cells located in bulge region of ORS during telogen-anagen transition [*Development* (2015), *Sem. Cell. Dev Biol.* (2014)], and Lgr5 expressing cell progeny moves downward to form hair germ and new hair follicle finally, which controls the new cycling of anagen hair follicle [*Nat. Gene.* 2008]. Our study showed TGFβ2 induced apoptosis of ORS cells, and keratin exposed from TGFβ2-induced apoptotic ORS cells could control DP cell condensation and the emergency of P-cadherin expressing cell population, which proved by various *in vitro* cell assays using various methodologies including direct co-culture of DP cells with ORS cells, indirect coculture of DP cells under conditioned medium, KRT31/34-knockdown of ORS cells, and immunodepletion of keratin from the conditioned medium. Furthermore, as your kind comments, to support our finding, several assays were done newly, and Lgr5⁺ cell population could be generated from ORS cells when exposed to keratin [Supplementary Fig.17]. In addition, in KRT31/34 -silenced mouse model study, distinctly reduced anagen follicle formation in KRT31/34-knockdown mouse and recovery of anagen follicle formation in KRT31/34-knockdown and exogenous keratin-injected mouse suggested that keratin might participate in anagen follicle formation. In addition, the emergency of Lgr5⁺ cell population in ORS region was further characterized by *in situ* RNA hybridization with the skin sections of keratin-injected mouse, which is added in Supplementary Figure 2. From these results, the injection of exogenous keratin might play a role of inducing new anagen follicle formation like endogenous keratin exposed from spatiotemporally TGFβ2-induced apoptotic ORS cells, and keratin might trigger new hair follicle and entry of new hair cycle from telogen to anagen. However, as your concern, the animal study to link our *in vitro* finding with *in vivo* mechanism in hair growth should be necessary. In addition, we also showed the keratin-mediated maintenance in aggregation activity of DP cell spheroid in figure 2, and it could be expected that the injection of keratin might influence prolonged anagen hair follicle maintenance, but the effect of keratin on follicles on other stages of anagen and catagen need to be studied more. This study was done as a pilot study to suggest a potent biological role of keratin in hair growth, which was majorly based on *in vitro* cell assay. Hence, these pilot observations require follow-up studies of hair keratin expression and its apoptosis-related release during the hair cycle using genetically modified ORS cells equipped with an on-off expression system for keratin expression and a xenograft mouse model to determine the *in vivo* mechanisms. As your kind comments, our proposed biological mechanism of keratin in hair growth is illustrated in Supplementary Figure 39 for better understanding, and the manuscript was re-written more

scientifically with newly added several references and supplementary figures.

The corrected and added paragraphs from line 349 to line 384 in the section of Discussion are as follows;

“Finally, to determine the biological function of keratin *in vivo*, the effect of downregulating KRT31/KRT34 mRNA expression on hair growth in mice was evaluated. Exogenous keratin injection in KRT31/34 knockdown mice resulted in a relatively reduced formation of catagen follicles on day 7. This might be due to the temporal inhibition of keratin expression accompanied by stem cell differentiation into the matrix and shaft during the anagen phase, which might influence catagen formation. In addition, the formation of anagen hair follicles and hair growth were suppressed in mice with temporal downregulation of KRT31/KRT34, which could be recovered by intradermal injection of additional exogenous keratin. With the poor formation of anagen follicles in KRT31/KRT34 knockdown mice, P-cadherin and Lgr5 expressing cell population was not scarcely observed in telogen follicles in these mice (Supplementary Fig. 30, 31). *In vitro* KRT31/34 knockdown ORS cells did not show any change of ability for cell growth and differentiation into P-cadherin and Lgr5 expressing cells (Supplementary Fig. 28, 29), and such P-cadherin and Lgr5 expressing cell population could be formed by injecting exogenous keratin into KRT31/34 knockdown mice even though less than control mice (Supplementary Fig. 31). These findings indicate that alteration of hair keratin gene expression might influence hair growth following stem cell differentiation into the matrix and shaft, and hair cycle transitions might be controlled by keratin-mediated microenvironmental change. The expression of hair keratins is not restricted to the anagen phase, showing cell growth and differentiation-mediated hair keratin production, and are found at all stages of the hair cycle⁴¹⁻⁴³. However, keratin expression in hair and its related biological roles during the hair cycle remain unknown. In addition, although apoptosis is known to be a main event indicating the entry of catagen, it is not clear whether programmed cell death, such as apoptosis, occurs during other stages of the hair cycle. A distinct epithelial cell population expressing Bcl-2 in secondary HG and DP was found during the telogen-anagen transition, which shows differential susceptibility to apoptosis⁴⁴. Such programmed cell death-related cellular processes were also detected during telogen, and the stimulation of autophagy following programmed cell death initiated the telogen-anagen transition⁴⁵. A study using transgenic mice overexpressing an anti-apoptotic gene reported that the inhibition of the apoptotic death of ORS cells even during anagen resulted in the early termination of hair follicle stem cell activation and proliferation, whereas the initiation of a new hair cycle was postponed by inhibiting the apoptotic death of ORS cells during telogen⁴⁶. In addition, a study suggested that telogen might not be the only resting phase in hair growth, but also an activating phase, including DP condensation and secondary HG formation^{47,48}. These studies indicate that spatiotemporal apoptosis during the hair cycle can be an essential process in controlling hair regeneration, and our findings show that keratin can be an important factor influencing hair growth. However, these pilot observations require follow-up studies of hair keratin expression and its apoptosis-related release during the hair cycle using genetically modified ORS cells equipped with an on-off expression system for keratin expression and a xenograft mouse model to determine the *in vivo* mechanisms.”

[References]

11. Hibino, T. & Nishiyama, T. Role of TGF- β 2 in the human hair cycle. *J. Dermatol. Sci.* **35**, 9-18 (2004).
22. Krieger, T. & Simons, B.D. Dynamic stem cell heterogeneity. *Development* **142**, 1396-1406

- (2015)
23. Rompolas, P. & Greco, V. Stem cell dynamics in the hair follicle niche. *Sem. Cell. Dev. Biol.* **25-26**, 34-42 (2014)
 24. Jaks, V. et al. Lgr5 marks cycling, yet long-lived, hair follicle stem cells. *Nat. Gene.* **40**, 1291-1299 (2008)
 41. Gu, L. & Coulombe, P.A. Keratin expression provides novel insight into the morphogenesis and function of the companion layer in hair follicles. *J. Invest. Dermatol.* **127**, 1061-1073 (2007)
 42. Veniaminova, N.A. et al. Keratin 79 identifies a novel population of migratory epithelial cells that initiates hair canal morphogenesis and regeneration. *Development* **140**, 4870-4880 (2013)
 43. Wiener, D.J. et al. Transcriptome profiling and differential gene expression in canine microdissected anagen and telogen hair follicles and interfollicular epidermis. *Genes* **11**, 884 (2020)
 44. Tsutomu, S. & Toshihiko, H. Dominant Bcl-2 expression during telogen-anagen transition phase in human hair. *J. Derm. Sci* **36**, 183-185 (2004)
 45. Min, C. et al. Stimulation of Hair growth by small molecules that activate autophagy. *Cell. Rep.* **27**, 3413-3421 (2019)
 46. Pena, J.C., Kelekar, A., Fuchs, E.V. & Thompson, C.B. Manipulation of outer root sheath cell survival perturbs the hair-growth cycle. *EMBO J* **18 (13)** 3596-3603 (1999)
 47. Geyfman, M., Plikus, M.V., Treffeisen, E., Andersen, B. & Paus, R. Resting no more: re-defining telogen, the maintenance stage of the hair growth cycle. *Biol. Rev. Camb. Philos. Soc.* **90(4)**, 1179-1196 (2015).
 48. Geyfman, M., Gordon, W., Paus, R. & Andersen, B. Identification of novel telogen markers underscores that telogen is far from a quiescent hair cycle phase. *J Invest Dermatol.* **132**, 721-724 (2012)
-

5. Reviewer's Comment:

- Authors suggest that as opposed to DP cells, ORS cells stimulated with Keratin are more proliferating (Fig. 3). Authors should confirm ORS vs DP cell proliferation with the appropriate markers (ki67/Edu), and clarify the cell-specific proliferative phenotype they observed.

5. Answer:

- Your comment is appreciated. As your kind comments, the proliferation of DP cells and ORS cells in the presence of keratin was evaluated by immunocytochemical staining with anti-Ki67 antibody and anti-BrdU antibody. The decrease in proliferation of keratin treated DP cells tested by CCK-8 assay was consistent with the reduced expression of Ki67 and BrdU incorporation. As your kind comments, the result is added in Supplementary Figure 6, and 15, and some paragraphs were corrected in the section of results and discussion.

The corrected paragraphs from line 127 to line 131, from line 146 to line 148, and from line 313 to line 317 in the section of results and discussion are as follows;

“However, the growth of DP cells was suppressed upon exposure to keratin (Supplementary Fig. 6A), and relatively lower Ki67 expression and BrdU incorporation were observed in keratin-treated DP cells (Supplementary Fig. 6B, C). Condensed DP cell aggregates contained cells expressing Ki67, and β -catenin were expressed in contact region between cells within DP cell aggregates (Supplementary

Fig. 7), in which DP cells remained viable during culture (Supplementary 8).”

“ORS cells formed colonies within a few hours of exposure to keratin, and subsequently formed strand-like extended structures by day 3 (Fig. 3A, B). The proliferation of ORS cells was suppressed upon exposure to keratin, as indicated by decreased BrdU incorporation (Supplementary Fig. 15).”

“Along with their differentiation, growth of ORS cells was stopped in the presence of keratin, but CCK-8 analysis, which measures mitochondrial dehydrogenase activity, showed a temporal increase in growth on day 1 of keratin treatment (Supplementary Fig. 15). This change in cellular metabolic activity by keratin treatment needs to be studied further.”

6. Reviewer’s Comment:

- ORS cell apoptosis following TGFβ2 stimulation should be quantified for TUNEL, annexin, and caspase 3. Generally, authors have the tendency to present immunofluorescent images without proper quantifications and statistics, which should be revised throughout the paper.

6. Answer:

- Your comment is appreciated. As your kind comments, western blot analysis was done to quantify the molecular expressions of annexin and caspase 3, which result is added in a Supplementary Figure 21. In addition, real time-qPCR analysis to evaluate several gene expressions for keratin-mediated DP cell property and ORS cell differentiation are added, and apoptosis array analysis was done triplet to quantify apoptosis-related markers. Supplementary Figure 18 and Figure 4A were newly added and corrected.
-

7. Reviewer’s Comment:

- Importantly, authors should clearly state and show how they quantified the different stages of the hair follicle. In general, hair follicles are synchronized across the hair coat in mice. It is therefore unclear how in control mice, the authors identify ~50% of the hair follicles in Telogen while ~30% are at catagen. Did the authors analyze hair stages at different locations within the back skin? In that case, how authors controlled for consistency within the different experimental groups? Additional concern stem from inconsistency between the phenotype presented image in Fig. 1A-B (all follicles are at full anagen), and the quantification in Fig. 1C-D (only 30% of HF are at anagen?). A similar concern for data presented in Fig. 7.
- Quantification and classification of hair follicle stages in the back skin is unclear and not supported by data. During homeostatic conditions, the mice hair coat is largely synchronized. The images presented do not support the conclusion of a mixed telogen, anagen, and catagen stages in the hair coat all at the same time (Fig. 1 and Fig. 7).

7. Answer:

- Your comment is appreciated. As shown in below figure, hair follicle stage and other calculation was done by analyzing several tissue sections of the tested mice, and the images in figures were selected as a representative in each experimental group. The anagen hair follicles were hardly observed in control mice, and keratin or minoxidil treated mice showed mixed stages of hair follicles, which discrepancy in synchronization of hair follicle stage might be related to spontaneous normal hair cycle progression.

In addition to the effect of keratin on anagen hair follicle formation, we showed the keratin-mediated-DP cell spheroid in figure 2, and it could be expected that the injection of keratin might influence prolonged anagen hair follicle maintenance, but the effect of keratin on follicles on stage of anagen and catagen need to be studied more. Even though standard deviation was relatively high in the quantification of hair follicle stage, the difference between experimental groups and control was distinct. The below figure is provided only to answer reviewers comment and for better understanding.

8. Reviewer's Comment:

- A major concerning issue that is missing in this paper is how fragmented keratin stimulates DP cell condensation and HG formation. I find the data of HG formation to be extremely weak, not supported by data, and was not appropriately controlled. if indeed fragmented keratin filament induces DP cell condensation in-vivo, it is unclear how this fit into normal hair cycle progression and HG formation. Authors should reconcile this discrepancy experimentally using in-vivo mice hair cycle progression. Equally unclear throughout the study and in light of their initial experiments, authors should confirm DP gene expression and cell condensation in an in vivo mice model of keratin injection.

8. Answer:

- Your comment is appreciated. As your kind comment, we have more evaluated the keratin-mediated stem cell activation for hair germ formation. *Lgr5*⁺ cell population is known to be generated from *CD34*⁺ stem cells located in bulge region of ORS during telogen-anagen transition [*Development* (2015), *Sem. Cell. Dev. Biol.* (2014)], and *Lgr5* expressing cell progeny moves downward to form hair germ and new hair follicle finally, which controls the new cycling of anagen hair follicle [*Nat. Gene.* 2008]. In this study, *Lgr5*⁺ cell population could be generated from ORS cells when exposed to keratin [Supplementary Fig.17], and also increased *Lgr5*⁺ cells in the lower bulge region were found

in keratin injected mice [Supplementary Fig.2]. In addition, gene expression profile of ORS cells was also evaluated when exposed to keratin *in vitro*, and highly upregulated mRNA expressions of several markers, such as integrin $\alpha 6$, Sox9, FoxN1, MSX2 and Shh, indicating stem cell activation was observed [Supplementary Fig.18]. In contrast to ORS cell differentiation, ORS cell proliferation was reduced upon keratin treatment [Supplementary Fig.15]. Several supplementary figures were newly added to support keratin-mediated stem cell activation, and the manuscript was re-written more scientifically for better understanding and references were added more to support.

- The corrected paragraphs from line 106 to line 108, from line 146 to line 148 and from line 153 to line 158 in the section of result are as follows;

“In addition, *in situ* RNA hybridization showed a significant increase in the Lgr5-positive cell populations in the lower bulge region and hair follicles after keratin injection (Supplementary Fig. 2).”

“ORS cells formed colonies within a few hours of exposure to keratin, and subsequently formed strand-like extended structures by day 3 (Fig. 3A, B). The proliferation of ORS cells was suppressed upon exposure to keratin, as indicated by decreased BrdU incorporation.”

“With reduced expression of CD34, the Lgr5+ cell population, which is known to participate in hair germ formation²²⁻²⁴, emerged in keratin-treated ORS cells (Supplementary Fig. 17). In addition, real-time qPCR analysis showed upregulation of the expression of various genes, such as SOX9, EDAR, FOXN1²⁵, MSX2²⁶, SHH²⁷, EFNB1, ITGA6²⁸ and β -catenin, indicating matrix and shaft differentiation of stem cells from the ORS bulge region (Supplementary Fig. 18).”

[References]

22. Krieger, T. & Simons, B.D. Dynamic stem cell heterogeneity. *Development* **142**, 1396-1406 (2015)
23. Rompolas, P. & Greco, V. Stem cell dynamics in the hair follicle niche. *Sem. Cell. Dev. Biol.* **25-26**, 34-42 (2014)
24. Jaks, V. et al. Lgr5 marks cycling, yet long-lived, hair follicle stem cells. *Nat. Gene.* **40**, 1291-1299 (2008)

- As your kind comments, gene expression profile of keratin-treated DP cells was evaluated, and highly upregulated mRNA expressions of several markers such as CD133, Sox2, Corin, Shh, versican, β -catenin, BMP6, FGF7 and FGF10 were detected in keratin-treated DP cell culture, which result is newly added as Supplementary Figure 13, and sentences demonstrating the result of gene expression profile are added in the section of result.

The paragraphs from line 137 line 140 in the section of result are as follows;

“Real-time qPCR analysis showed upregulation of various genes, such as CD133, SOX2, corin, SHH, versican, β -catenin, BMP6, FGF7, and FGF10, known as a molecular identity signature reflecting the hair-inductive property of DP cells¹⁶⁻¹⁹ (Supplementary Fig. 13).”

[References]

16. Driskell, R.R., Giangreco, A., Jensen, K.B., Mulder, K.W. & Watt, F.M. SOX2-positive dermal papilla cells specify hair follicle type in mammalian epidermis. *Development* **136**, 2815-2823 (2009).
17. Yamauchi, K. & Kurosaka, A. Inhibition of glycogen synthase kinase-3 enhances the expression

of alkaline phosphatase and insulin-like growth factor-1 in human primary dermal papilla cell culture and maintains mouse hair bulbs in organ culture. *Arch. Dermatol. Res.* **301**, 357-365 (2009).

18. Rendl, M., Polak, L. & Fuchs, E. BMP signaling in dermal papilla cells is required for their hair follicle-inductive properties. *Genes Dev.* **22**, 543-557 (2008).
- This study was done as a pilot study to suggest a potent biological role of keratin in hair growth, which was majorly based on *in vitro* cell assay. As reviewer's comment, keratin treated mice showed mixed stages of hair follicles, which might be related to spontaneous normal hair cycle progression. Such mixed stages of hair follicles were also found in minoxidil-treated group. In addition to the effect of keratin on anagen hair follicle formation, we showed the keratin-mediated maintenance in aggregation activity of DP cell spheroid in figure 2, and it could be expected that the injection of keratin might influence prolonged anagen hair follicle maintenance, but the animal study to link our *in vitro* finding with *in vivo* mechanism in hair growth should be necessary. Hence, we have developed recombinant keratin protein and will show the biological mechanism using keratin-silenced ORS cells transplantation in a xenograft model in another continuous study, which was demonstrated in the section of discussion. As your kind comments, the manuscript was re-written more scientifically for better understanding, and some paragraphs were added in the section of discussion.

The added paragraphs from line 379 to line 384 in the section of discussion are as follows;

“These studies indicate that spatiotemporal apoptosis during the hair cycle can be an essential process in controlling hair regeneration, and our findings show that keratin can be an important factor influencing hair growth. However, these pilot observations require follow-up studies of hair keratin expression and its apoptosis-related release during the hair cycle using genetically modified ORS cells equipped with an on-off expression system for keratin expression and a xenograft mouse model to determine the *in vivo* mechanisms.”

9. Reviewer's Comment:

- The notion that Keratin released by apoptotic cells during hair regression has a role in dermal cell condensation and priming for the new hair cycle is intriguing. It will be of interest to show free Keratin in the extracellular space during catagen, and whether indeed inhibition of TGFβ2 signaling can attenuate keratin secretion. Importantly, authors should also reconcile the consequence of events. It is unclear how keratin delivery during catagen will stimulate hair growth, particularly when the following stage might be a prolonged telogen, where hair does not grow.

9. Answer:

- Your comment is appreciated. We demonstrated that spatiotemporal exposure of keratin from TGFβ2-mediated apoptotic ORS cells induced DP condensation and 2nd germ formation from ORS cells containing stem cells, allowing to enter anagen stage. In addition, extracellular interaction of keratin could generate Lgr5⁺ cell population which is known to participate in germ formation [*Development* (2015)], from CD34⁺ ORS cells, which result is added newly in Supplementary Figure 17, and we added supplemental figure 36, 37 and 38 to show how the keratin to influence microenvironmental change which demonstrated in a previous comment. In this study, DP cell condensation was also

induced on matrigel (Supplementary Fig.10), and we found the decreased hardness of matrigel in the presence of keratin. Hence, we tested the change of hardness of matrigel in the presence of keratin or not, keratin treatment resulted in partial disintegration of matrigel (Supplementary Fig.36), which might influence cell and matrix interaction. In addition, loss of vinculin participating at local cell adhesion was found in keratin-mediated DP cell condensation, and highly decreased molecular expression of vinculin was observed in keratin treated ORS cells (Supplementary Fig.37 and 38). Recent report showed that mechanical instability of cell to ECM contact was a factor to control activation of hair follicle stem cell in bulge region, which proved by the finding that loss of vinculin allowed hair follicle stem cell to escape from quiescence and forced the initiation of new hair cycle [Dev. Cell (2021)]. Keratin released or deposited from TGFβ2-induced spatiotemporal apoptotic ORS cells might influence mechanical property of microenvironment and cell to ECM interaction, which might be a cue to derive DP cell condensation and activation of hair follicle stem cell participating in hair germ formation. However, further study on such keratin-mediated mechanotransduction is needed and going on. These additional results are added newly as supplemental figures, and some paragraphs were added in the section of discussion.

The added paragraphs from line 385 to line 400 in the section of Discussion are as follows;

“Despite *in vitro* and *in vivo* studies of keratin-mediated hair growth, the mechanism by which keratin induces DP condensation and hair germ formation remains unclear. In this study, DP cell condensation was also induced on Matrigel (Supplementary Fig.10), and we found a decrease in the hardness of Matrigel in the presence of keratin. Hence, we tested the keratin-mediated change in the hardness of Matrigel, and keratin treatment resulted in partial disintegration of Matrigel (Supplementary Fig.36), which might influence cell and matrix interactions. In addition, loss of vinculin, which participates in local cell adhesion, was found in keratin-mediated DP cell condensation (Supplementary Fig.37), and highly decreased expression of vinculin was observed in keratin-treated ORS cells (Supplementary Fig.38). A recent report showed that mechanical instability of cells to ECM contact was a factor in controlling activation of hair follicle stem cells in the bulge region, which was proved by the finding that loss of vinculin allowed hair follicle stem cells to escape quiescence and forced the initiation of a new hair cycle⁴⁹. Keratin released or deposited from TGFβ2-induced spatiotemporal apoptotic ORS cells might influence the mechanical properties of the microenvironment and cell-to-ECM interactions, which might be a cue to drive DP cell condensation and activation of hair follicle stem cells participating in hair germ formation. However, further studies on keratin-mediated mechanotransduction are required.”

[References]

49. Biswas, R., et al. Mechanical instability of adherent junctions overrides intrinsic quiescence of hair follicle stem cells. *Dev. Cell.* **56**, 761-780 (2021)

- As shown in figure 4B, 5H and below figure, TUNEL staining-positive apoptotic ORS cells was majorly found in the stranded structure formed by TGFβ2-treated ORS cells, and strong molecular expressions of caspase-6 and KRT34 was found at such stranded structure, which might indicate keratin exposure from apoptotic ORS cell population. However, as your kind comment, we added several figures to show keratin release from apoptotic ORS cells. As shown in below figure, intermediate filament was found to be released from apoptotic ORS cells, and, furthermore, keratin release from apoptotic ORS cells was evaluated using KRT34 expressing ORS cells. KRT34 expressing vector was transfected into ORS cells, and KRT34 positive staining was only found at

apoptotic ORS cells. Below figure is provided only to answer the reviewer's comment.

- Interestingly, we found higher populations of apoptotic cells in KRT34 vector-transfected ORS cell culture in comparison with normal ORS cells. Such hair keratin expressions including KRT34 within cells might be related to the differentiation of ORS cells into cuticle cells or other cell types participating in hair growth. During differentiation, hair keratins might be expressed and accumulated within cells, and differentiated cells expressing hair keratins might be easily ready to undergo programmed cell death such as apoptosis. However, differentiation stage-dependent hair keratin expressions and its related programmed cell death need to be studied more and is on-going for another publication.

10. Reviewer's Comment:

- Curiously, authors show that KRT31/KRT34 silencing during telogen is sufficient to inhibit hair growth. It is unclear, however, whether keratin release happens independently of catagen and cell death. If the suggested working model is correct, hair growth delay will be noticeable only on the following hair cycle and not immediately, as cell death rarely happens during telogen to anagen transition. In that case, KRT31/KRT34 silencing might delay hair progression, irrespective of keratin release to the extracellular space. Without having rigorous experimental validations of their findings, the reader cannot evaluate the extent to which the conclusions drawn are valid.

10. Answer:

- Your comment is appreciated. As the reviewer's concern, the silencing mouse model did not exactly link with the *in vitro* finding that keratin exposed from TGFβ2-induced apoptotic ORS cells induce DP condensation and germ formation, which might be related to TGFβ2-mediated spatiotemporal apoptosis during catagen and telogen and DP condensation and 2nd hair germ formation during telogen finally to form new anagen hair follicles. Although cell differentiation, TGFβ2-mediated epithelial apoptosis and microenvironmental change involved in the anagen and anagen-catagen transition have been studied extensively, it has not been clear that programmed cell death such as apoptosis is processed further during other stages of hair cycle. A study suggested that telogen might not be only resting phase in hair growth but also activating phase including DP condensation and secondary hair germ formation [Biol. Rev. Camb. Philos. Soc. (2015)]. In addition, it was reported that distinct cell population expressing Bcl-2 in epithelial cells of secondary HG and DP was found during telogen-anagen transition, which show differential susceptibility to apoptosis [J. Derm. Sci. (2004)], programmed cell death related cellular process such as apoptosis and was also detected during telogen, and controlling such programmed cell death induced telogen-anagen transition [Cell. Rep. (2019)]. A study using transgenic mice overexpressing an anti-apoptotic gene reported that the inhibition of the apoptotic death of ORS cells even during anagen resulted in the early termination of hair follicle stem cell activation and proliferation, while the initiation of a new hair cycle was postponed by inhibiting the apoptotic death of ORS cells during telogen [EMBO. J. (1999)]. In this study, KRT31/34-knockdown mice showed the highly prohibited formation of anagen follicle and the recovery of anagen follicle formation in KRT31/34-knockdown mice by injecting exogenous keratin. From this finding, KRT31/34 knockdown might influence also gene expressions of ORS cells during telogen and further telogen-anagen transition. According to the reviewer's concern, several supplementary figures were newly added, and the manuscript was re-written more scientifically for better understanding, and references were added more to support.

The added paragraphs from line 349 to line 384 in the section of discussion are as follows;

“Finally, to determine the biological function of keratin *in vivo*, the effect of downregulating KRT31/KRT34 mRNA expression on hair growth in mice was evaluated. Exogenous keratin injection in KRT31/34 knockdown mice resulted in a relatively reduced formation of catagen follicles on day 7. This might be due to the temporal inhibition of keratin expression accompanied by stem cell differentiation into the matrix and shaft during the anagen phase, which might influence catagen formation. In addition, the formation of anagen hair follicles and hair growth were suppressed in mice with temporal downregulation of KRT31/KRT34, which could be recovered by intradermal injection

of additional exogenous keratin. With the poor formation of anagen follicles in KRT31/KRT34 knockdown mice, P-cadherin and Lgr5 expressing cell population was not scarcely observed in telogen follicles in these mice (Supplementary Fig. 30, 31). *In vitro* KRT31/34 knockdown ORS cells did not show any change of ability for cell growth and differentiation into P-cadherin and Lgr5 expressing cells (Supplementary Fig. 28, 29), and such P-cadherin and Lgr5 expressing cell population could be formed by injecting exogenous keratin into KRT31/34 knockdown mice even though less than control mice (Supplementary Fig. 31). These findings indicate that alteration of hair keratin gene expression might influence hair growth following stem cell differentiation into the matrix and shaft, and hair cycle transitions might be controlled by keratin-mediated microenvironmental change. The expression of hair keratins is not restricted to the anagen phase, showing cell growth and differentiation-mediated hair keratin production, and are found at all stages of the hair cycle⁴¹⁻⁴³. However, keratin expression in hair and its related biological roles during the hair cycle remain unknown. In addition, although apoptosis is known to be a main event indicating the entry of catagen, it is not clear whether programmed cell death, such as apoptosis, occurs during other stages of the hair cycle. A distinct epithelial cell population expressing Bcl-2 in secondary HG and DP was found during the telogen-anagen transition, which shows differential susceptibility to apoptosis⁴⁴. Such programmed cell death-related cellular processes were also detected during telogen, and the stimulation of autophagy following programmed cell death initiated the telogen-anagen transition⁴⁵. A study using transgenic mice overexpressing an anti-apoptotic gene reported that the inhibition of the apoptotic death of ORS cells even during anagen resulted in the early termination of hair follicle stem cell activation and proliferation, whereas the initiation of a new hair cycle was postponed by inhibiting the apoptotic death of ORS cells during telogen⁴⁶. In addition, a study suggested that telogen might not be the only resting phase in hair growth, but also an activating phase, including DP condensation and secondary HG formation^{47,48}. These studies indicate that spatiotemporal apoptosis during the hair cycle can be an essential process in controlling hair regeneration, and our findings show that keratin can be an important factor influencing hair growth. However, these pilot observations require follow-up studies of hair keratin expression and its apoptosis-related release during the hair cycle using genetically modified ORS cells equipped with an on-off expression system for keratin expression and a xenograft mouse model to determine the *in vivo* mechanisms.”

[References]

41. Gu, L. & Coulombe, P.A. Keratin expression provides novel insight into the morphogenesis and function of the companion layer in hair follicles. *J. Invest. Dermatol.* **127**, 1061-1073 (2007)
42. Veniaminova, N.A. et al. Keratin 79 identifies a novel population of migratory epithelial cells that initiates hair canal morphogenesis and regeneration. *Development* **140**, 4870-4880 (2013)
43. Wiener, D.J. et al. Transcriptome profiling and differential gene expression in canine microdissected anagen and telogen hair follicles and interfollicular epidermis. *Genes* **11**, 884 (2020)
44. Tsutomu, S. & Toshihiko, H. Dominant Bcl-2 expression during telogen-anagen transition phase in human hair. *J. Derm. Sci* **36**, 183-185 (2004)
45. Min, C. et al. Stimulation of Hair growth by small molecules that activate autophagy. *Cell. Rep.* **27**, 3413-3421 (2019)
46. Pena, J.C., Kelekar, A., Fuchs, E.V. & Thompson, C.B. Manipulation of outer root sheath cell survival perturbs the hair-growth cycle. *EMBO J* **18 (13)** 3596-3603 (1999)
47. Geyfman, M., Plikus, M.V., Treffeisen, E., Andersen, B. & Paus, R. Resting no more: re-defining

telogen, the maintenance stage of the hair growth cycle. *Biol. Rev. Camb. Philos. Soc.* **90(4)**, 1179-1196 (2015).

48. Geyfman, M., Gordon, W., Paus, R. & Andersen, B. Identification of novel telogen markers underscores that telogen is far from a quiescent hair cycle phase. *J Invest Dermatol.* **132**, 721-724 (2012)
-

11. Reviewer's Comment:

- The expression of KRT31/KRT34 should be evaluated in situ using immunofluorescence following intravascular lipofectamine-mediated delivery of KRT31/KRT34 siRNAs. Off targets should also be evaluated and the reduction of fragmented KRT31/KRT34 during catagen should be demonstrated. The authors should also consider other methods of assessing the role played by KRT31/KRT34 on hair growth, rather than just staging the hair cycle. For example, do the cells survive better upon loss of KRT31/KRT34? Do they proliferate more, or with a shorter cell cycle? Do follicles experience an accelerated catagen?

11. Answer:

- Your comment is appreciated. As shown in Figure 7A, the knockdown of KRT31 and KRT34 mRNA was confirmed by real time-qPCR after 7 days of transfection, and also immunohistological images showed molecular expression of KRT34 was found to be distinctly decreased in tissue section of KRT31/34 siRNA transfected mice [Supplementary Fig.32]. In addition, as your kind comments, ORS growth assay was done to evaluate the effect of KRT31/34 knockdown by CCK-8 assay and BrdU immunocytochemical staining. As shown in the newly added Supplementary Figure 28, *in vitro* KRT31/34 knockdown was confirmed by real time-qPCR, and there was no significant difference in cell growth between control ORS cells and KRT31/34-knockdown ORS cells. In addition to cell proliferation, upon exogenous keratin treatment in normal ORS and KRT31/34-knockdown ORS cell culture, ORS cell differentiation to Lgr5-positive cell with the decrease of CD34 expression was not influenced by KRT31/34-knockdown. With the emergence of Lgr5+ cell population, P-cadherin expressing cells were also found in both control ORS cells and KRT31/34-knockdown ORS cells upon keratin treatment [Supplementary Fig.29]. Interestingly, P-cadherin was expressed strongly in the peripheral membrane region of keratin treated ORS cells, but a few control ORS cells and siRNA-transfected ORS cells showed nuclear location of P-cadherin not in the presence of keratin. It was reported that WNT/ β -catenin signaling pathway to activate stem cells was repressed by the loss of cell surface cadherin and nuclear location of cadherin [Science (2004), Oncogenesis (2015)]. A population showing nuclear location of P-cadherin with a capacity to form secondary germ might reside in inactivated status, and such inactivation might be relieved by keratin-mediated microenvironmental change. In addition, P-cadherin expression was found to be relatively well developed in peripheral region of KRT31/34-knockdown ORS cells in the presence of keratin in comparison with keratin-treated ORS cells [Supplementary Fig.29]. Keratin intermediate filaments within epithelial cells to form intracellular structural scaffold was known to allow cells to be resistant against external mechanical forces and to provide different mechanosensing in response to altered matrix rigidity [PNAS (2008), Sci Adv (2021)]. The decreased expression of KRT31/34 within KRT31/34-knockdown ORS cells might facilitate ORS cells to response keratin-mediated microenvironmental change. However, nuclear location of P-cadherin and its related underlying signaling is not elucidated, and the role of extracellular and intracellular hair keratins in physical

change of cell and microenvironment need to be studied further. According to the reviewer's concern, several supplementary figures were newly added, and the manuscript was re-written more scientifically for better understanding, and references were added more to support.

The added paragraphs from line 252 to line 253, from line 268 to line 287, and from line 349 to line 384 in the section of results and discussion are as follows;

“KRT31/KRT34 downregulation did not influence cell growth or the generation of Lgr5⁺ and P-cadherin⁺ cell populations in ORS cells (Supplementary Fig. 28, 29).”

“In addition, the formation of catagen follicles was relatively reduced on day 7 upon downregulation of KRT31/KRT34 in mice with or without exogenous keratin injection (Fig 7C). An anagen bulb containing a population of cells expressing P-cadherin was hardly seen in immunohistological sections of KRT31/KRT34 knockdown mice (Fig 7D, Supplementary Fig. 30). Emergence of Lgr5 positive cell population and molecular expression of KRT34 were also distinctly decreased in KRT31/KRT34 knockdown mice on day 7 (Supplementary Fig. 31, 32). In contrast, relatively higher Lgr5-positive staining and the injected exogenous keratin reacted with KRT34 antibody were observed in exogenous keratin-injected KRT31/KRT34 knockdown mice on day 7 in comparison with KRT31/KRT34 knockdown mice (Supplementary Fig. 31), and an additional injection of hair-derived keratin after KRT31/KRT34 siRNA transfection allowed the hair follicles to enter the anagen phase and regrow hair, similar to the controls. No obvious histological differences were found in the formation of hair follicles and hair growth between the control skin and keratin-injected skin of KRT31/KRT34 knockdown mice after 2 weeks (Fig 7E). Furthermore, the formation of P-cadherin-positive germs and strong expression of β -catenin were observed in the region of anagen hair follicles in sections of control skin and keratin-injected skin of KRT31/KRT34 knockdown mice (Fig 7D, Supplementary Fig. 30), and strong staining for KRT34 was found in the ORS region surrounding the DP, which corresponds to the caspase-6-positive region (Supplementary Fig. 33). Interestingly, it was found that the region stained positive for caspase-6, KRT34, and P-cadherin moved upward into the hair shaft region of the expanded hair follicles (Supplementary Fig. 33).”

“Finally, to determine the biological function of keratin *in vivo*, the effect of downregulating KRT31/KRT34 mRNA expression on hair growth in mice was evaluated. Exogenous keratin injection in KRT31/34 knockdown mice resulted in a relatively reduced formation of catagen follicles on day 7. This might be due to the temporal inhibition of keratin expression accompanied by stem cell differentiation into the matrix and shaft during the anagen phase, which might influence catagen formation. In addition, the formation of anagen hair follicles and hair growth were suppressed in mice with temporal downregulation of KRT31/KRT34, which could be recovered by intradermal injection of additional exogenous keratin. With the poor formation of anagen follicles in KRT31/KRT34 knockdown mice, P-cadherin and Lgr5 expressing cell population was not scarcely observed in telogen follicles in these mice (Supplementary Fig. 30, 31). *In vitro* KRT31/34 knockdown ORS cells did not show any change of ability for cell growth and differentiation into P-cadherin and Lgr5 expressing cells (Supplementary Fig. 28, 29), and such P-cadherin and Lgr5 expressing cell population could be formed by injecting exogenous keratin into KRT31/34 knockdown mice even though less than control mice (Supplementary Fig. 31). These findings indicate that alteration of hair keratin gene expression might influence hair growth following stem cell differentiation into the matrix and shaft, and hair cycle transitions might be controlled by keratin-mediated

microenvironmental change. The expression of hair keratins is not restricted to the anagen phase, showing cell growth and differentiation-mediated hair keratin production, and are found at all stages of the hair cycle⁴¹⁻⁴³. However, keratin expression in hair and its related biological roles during the hair cycle remain unknown. In addition, although apoptosis is known to be a main event indicating the entry of catagen, it is not clear whether programmed cell death, such as apoptosis, occurs during other stages of the hair cycle. A distinct epithelial cell population expressing Bcl-2 in secondary HG and DP was found during the telogen-anagen transition, which shows differential susceptibility to apoptosis⁴⁴. Such programmed cell death-related cellular processes were also detected during telogen, and the stimulation of autophagy following programmed cell death initiated the telogen-anagen transition⁴⁵. A study using transgenic mice overexpressing an anti-apoptotic gene reported that the inhibition of the apoptotic death of ORS cells even during anagen resulted in the early termination of hair follicle stem cell activation and proliferation, whereas the initiation of a new hair cycle was postponed by inhibiting the apoptotic death of ORS cells during telogen⁴⁶. In addition, a study suggested that telogen might not be the only resting phase in hair growth, but also an activating phase, including DP condensation and secondary HG formation^{47,48}. These studies indicate that spatiotemporal apoptosis during the hair cycle can be an essential process in controlling hair regeneration, and our findings show that keratin can be an important factor influencing hair growth. However, these pilot observations require follow-up studies of hair keratin expression and its apoptosis-related release during the hair cycle using genetically modified ORS cells equipped with an on-off expression system for keratin expression and a xenograft mouse model to determine the *in vivo* mechanisms.”

[Reference]

Nelson W.J., Nusse, R. Convergence of wnt, β -catenin, and cadherin pathways. *Science* **303**,1483-1487 (2004)

Su, Y.J., Chang, Y.W., Lin, W.H., Liang, C.W., Lee, J.L. An aberrant nuclear localization of E-cadherin is a potent inhibitor of Wnt/ β -catenin-elicited promotion of the cancer stem cell phenotype. *Oncogenesis* **4**, e157 (2015)

Sivaramakrishnan, S., DeGiulio, J.V., Lorand, L., Goldman, R.D., Ridge, K.M. Micromechanical properties of keratin intermediate filament networks. *PNAS* **105** (3) 889-894 (2008)

Laly, A.C., et al. The keratin network of intermediate filaments regulates keratinocyte rigidity sensing and nuclear mechanotransduction. *Sci. Adv.* **7** eabd6187 (2021)

- Furthermore, as the reviewer's comment, hair cycling and growth is not only controlled by DP cells and ORS cells containing stem cells, but also influenced by complex interplay of various kinds of cells such as sebocyte, endothelial and keratinocyte in skin. With the biological function of keratin in DP condensation and germ formation, we have studied on another side of biological function related to dehydrosterone clearance and tubulogenesis using sebocyte, keratinocyte and dermal microvascular endothelial cells in terms of tissue homeostasis related to hair growth. This pilot study has focused on the finding of a molecular cue to derive the initiation of new hair cycle, but, as the reviewer's concern, the effect of programmed cell death-derived keratin on different kinds of cell types composing hairy skin need to be studied, and is going on for another publication. A figure illustrating research in progress with other cell types is provided below only to answer the reviewer's comment.

[data redacted]

12. Reviewer's Comment:

- Finally, the paper is not clearly written and it is hard to follow the narrative of how TGFb induces keratin secretion & fragmentation and the subsequent effect on HG formation and DP cell condensation. Few methodological approaches are not adequately listed and explained and many of the presented data are not quantified. While the topic of this study is of interest to a broad readership, a number of experiments to document the author's claims are lacking appropriate controls and hence many of the conclusions made are not justified (i.e. hair germ formation, keratin release, and KRT31/KRT34 silencing experiments, etc).

12. Answer:

- Your comment is appreciated. As your kind comments, an illustration to show experimental procedure is added as supplemental figure 1 for better understanding. In addition, some molecular expressions were quantified by western blot analysis, apoptosis array and growth factor array, which added in supplementary figures. Furthermore, in order to support our finding, gene expression profile of keratin-treated DP cells and ORS cells was evaluated, and the emergency of Lgr5+ cell population in ORS region was further characterized by immunocytochemical staining of keratin treated ORS cells and in situ RNA hybridization with the skin sections of keratin-injected mouse, which is newly

added in supplementary figure. The manuscript was re-written more scientifically for better understanding and edited with more references.

13. Reviewer's Comment:

- Fig. 1B- it is hard to evaluate hair morphology and staging based on these sections. Authors should align follicles for better horizontal visualization and assessment of their data.

13. Answer:

- Your comment is appreciated. As your kind comments, the histological images and data to analyze the numbers and stages of hair follicles formed in keratin-injected mouse were rearranged by exchanging histological data in figure 1 with those in supplementary figure 1. The rearranged figure 1 shows images of both longitudinal sections and cross sections to clarify the stages of hair follicle more.
-

14. Reviewer's Comment:

- Fig. 1H. A few panels represent the same cells as a separate panel /experiment which makes it confusing and inconsistent with the data. If staining was done on the same set of fixed cells, the represented date should truthfully illustrate it. Additionally. The presented data makes it very hard to determined which marker is nuclear and which cytoplasmic. The authors should combine the staining with a membrane marker (same for Sup. Fig2C, etc and provide insets, when appropriate).

14. Answer:

- Your comment is appreciated. As your kind comments, to show clear images of DP cell aggregates, some images in figure 1 was changed, and confocal microscopic observation was done, and magnified images were added as a supplementary figure 7 and 14, which show distinct β -catenin expressions in the contact region between DP cell in aggregates.
-

15. Reviewer's Comment:

- Immunoflourence images in Fig5 are not presented with good quality and it is hard to assess the merit of the presented results. Generally, image quality should be improved throughout the paper.

15. Answer:

- Your comment is appreciated. It is regret to add lots of small images in figures in order to collect all data in the limited number of figures. We are sorry for making the images hard to see. As your kind comments, replaced some images with much clear one and enlarged immunocytochemical stained images throughout the figures and supplementary figures.
-

16. Reviewer's Comment:

- The authors are using human DP and ORS cell lines. it should be noted how these cells were isolated and to include markers that distinguish these cells.

16. Answer:

- Your comment is appreciated. For our study, primary DP cells and ORS cells were all purchased from CellBio company (<https://cefobio.com>), which was noted in the section of Method as follows; “Human outer root sheath cells (ORS; CEFO, CB-ORS-001) and human dermal papilla cells (DP; CEFO, CB-HDP-001) were purchased and expanded in each human outer root sheath cell growth medium (CEFO, CB-ORS-GM) and human dermal papilla growth medium (CEFO, CB-HDP-GM) at 37°C in a humidified atmosphere containing 5 % CO₂ according to the manufacturer's instructions.” . In addition, the information of DP cell and ORS cell which was provided from the company is added as follows;

Human Hair Cell System

HDP : Human hair follicle Dermal Papilla

ORS : Outer Root Sheath cells

Human Dermal Papilla Cell (DP Cell)

Rapid Test (All Negative)

Human Outer Root Sheath Cell (ORS Cell)

Rapid Test (All Negative)

18. Reviewer’s Comment:

- The current paper could benefit tremendously from professional language editing.

18. Answer:

- Your comment is appreciated. As your kind comments, the manuscript was re-written more scientifically for better understanding and English of manuscript was edited.
-

Reviewers' comments:

Reviewer #2 (Remarks to the Author):

The authors have responded to my and other reviewers' concerns. The data quality has shown improvement, and discussion has been added to points not clarified in the current paper. I believe that the revisions are appropriate and of a quality that allows the paper to be published.

Reviewer #3 (Remarks to the Author):

The revised manuscript of the study by Seong Yeong An et al., is considerably improved. The authors have taken the criticisms seriously and have adequately addressed most of the points raised by the reviewers. The current study contains substantial and essential controls with a more balanced discussion of the work. I still find many of the concerns raised during the revision process confusing and inconsistent with some of the presented results. The drawn narrative by which Keratin is indeed secreted during catagen hasn't been addressed, and discrepancies with Keratin silencing models are adding layers of confusion. However, given the much-improved controls and flow of the story, the intriguing idea behind the hair phenotype seems promising for future investigations.

A few minor comments:

- It will still be helpful to include in the discussion the possible ways by which Keratin exposure during the catagen stage may participate in the next hair cycle initiation, despite the appearance of a prolonged telogen stage in the next cycle.
- Immunofluorescent images should be presented with proper quantifications and statistics. This is particularly important while presenting data for apoptosis (TUNEL, Caspase 3), proliferation (BrdU, Ki67) and RUNX1 expression.
- The authors should clarify in the text that Keratin-treated mice showed mixed stages of hair follicles. That may stem from the typical distribution of Keratin following injection, as their new data suggests.
- Proliferation and apoptosis should have been done in vivo and not only in vitro following keratin injection.
- The point of whether Keratin is indeed released during catagen still needs to be addressed. However, it seems reasonable to publish the revised story as the notion that Keratin is released by apoptotic cells during hair regression and participates in promoting the new hair cycle is intriguing.
- Please provide a better explanation for concern #10.
- Why is the control skin in the siRNA-loaded lipofectamine group at anagen?
- Authors should correct their suggested model, demonstrating that Keratin is released to the extracellular space when such data hasn't been provided.

Manuscript #: COMMSBIO-20-3162

Title: Keratin is not only a Structural Protein in Hair: Keratin-mediated Hair Growth

A List of Changes

As reviewer's comments, all comments are answered in a list of changes, and all changes in our manuscripts are made. Several supplementary figures are newly added, and the corrected or added words or sentences in manuscript are highlighted in blue.

Reviewer #3's Comments: The revised manuscript of the study by Seong Yeong An et al., is considerably improved. The authors have taken the criticisms seriously and have adequately addressed most of the points raised by the reviewers. The current study contains substantial and essential controls with a more balanced discussion of the work. I still find many of the concerns raised during the revision process confusing and inconsistent with some of the presented results. The drawn narrative by which Keratin is indeed secreted during catagen hasn't been addressed, and discrepancies with Keratin silencing models are adding layers of confusion. However, given the much-improved controls and flow of the story, the intriguing idea behind the hair phenotype seems promising for future investigations.

1. Reviewer's Comment:

- It will still be helpful to include in the discussion the possible ways by which Keratin exposure during the catagen stage may participate in the next hair cycle initiation, despite the appearance of a prolonged telogen stage in the next cycle.

1. Answer:

- Your comment is appreciated. As your kind comments, the manuscript was re-written more scientifically for better understanding and additional supplemental figure was added more to support.

The added sentences from line 343 to line 354 in the section of Discussion are as follows;

“In our study, it was shown that local DP cell condensation and germ formation in TGFβ2-induced apoptotic ORS cells depend on the exposure of keratin via caspase-6 expression and consequently its-mediated **keratin exposure from apoptotic cell death**, as evidenced by the suppressed DP cell condensation and germ formation in caspase-6-knockdown or KRT31/KRT34-knockdown ORS cell culture, even if there was no distinct difference in TGFβ2-mediated ORS cell apoptosis. **From these *in vitro* findings, it could be inferred that spatially increased keratin exposure, following gradual apoptosis during regression stage, might provide a cue to derive germ formation and new hair cycle initiation from telogen to anagen, which spatial keratin deposit could be identified at upper void space unoccupied cells in the newly formed hair follicle (Supplementary Fig. 36). In addition, apoptosis-related keratin exposure also could be identified by similar spatial expressions of apoptosis-markers**

such as Annexin V and active caspase 3 and KRT 34 in the developing hair follicles containing proliferating cells (Supplementary Fig. 37).”

Supplementary Figure 36 | Keratin exposure from apoptotic cell death in developing hair follicles. Images of KRT34 exposure in hair follicles by immunohistological staining; DAPI, blue; KRT34, red or green. Scale bars, 20 μ m. Yellow arrows indicate keratin exposure in void space unoccupied by cells

- In addition, the paragraph to demonstrate the keratin-mediated microenvironmental change which might be a cue to influence DP cell condensation and ORS stem cell differentiation are previously noted from line 391 to line 406 in the section of Discussion are as follows;
“Despite *in vitro* and *in vivo* studies of keratin-mediated hair growth, the mechanism by which keratin induces DP condensation and hair germ formation remains unclear. In this study, DP cell condensation was also induced on Matrigel (Supplementary Fig.10), and we found a decrease in the hardness of Matrigel in the presence of keratin. Hence, we tested the keratin-mediated change in the hardness of Matrigel, and keratin treatment resulted in partial disintegration of Matrigel (Supplementary Fig.36), which might influence cell and matrix interactions. In addition, loss of vinculin, which participates in

local cell adhesion, was found in keratin-mediated DP cell condensation (Supplementary Fig.37), and highly decreased expression of vinculin was observed in keratin-treated ORS cells (Supplementary Fig.38). A recent report showed that mechanical instability of cells to ECM contact was a factor in controlling activation of hair follicle stem cells in the bulge region, which was proved by the finding that loss of vinculin allowed hair follicle stem cells to escape quiescence and forced the initiation of a new hair cycle ⁴⁹. Keratin exposure from TGFβ2-induced spatiotemporal apoptotic ORS cells might influence the mechanical properties of the microenvironment and cell-to-ECM interactions, which might be a cue to drive DP cell condensation and activation of hair follicle stem cells participating in hair germ formation. However, further studies on keratin-mediated mechanotransduction are required.”

- We are also now studying further keratin-mediated microenvironmental changes, and as shown in below figure, physical changes of Matrigel could be found to form specialized structure upon keratin treatment. This figure is only provided to answer the reviewer’s comment.

- In addition, Matrigel components such as type IV collagen, laminin, perlecan and nidogen was degraded spontaneously during incubation at 37°C due to the presence of several proteases such as

plasminogen, urokinase, MMP 2, and MMP9 in Matrigel, but a component, especially nidogen-1, was found to be stable in keratin-treated Matrigel, as shown in below figure. Also, keratin treatment influenced the expressions of basement membrane components in DP cell culture. Keratin exposed from apoptotic cells during regression might interact with extracellular matrix around DP cells and stem cells, and influence cell to matrix interaction which derive stem cell activation, differentiation and other biological process engaged in hair growth. Hence, the interaction of keratin with basement membrane components and its-mediated microenvironmental change are now studying more for next publication. This figure is only provided to answer the reviewer's comment.

[data redacted]

2. Reviewer's Comment:

- Immunofluorescent images should be presented with proper quantifications and statistics. This is particularly important while presenting data for apoptosis (TUNEL, Caspase 3), proliferation (BrdU, Ki67) and RUNX1 expression

2. Answer:

- Your comment is appreciated. As your kind comments, ORS cell populations expressing P-cadherin and Runx1 upon TGFβ2 treatment were quantified by flow cytometric analysis, which is newly added in Supplementary Fig. 27, and the sentences to demonstrate the results are added in the section of Result. In addition, BrdU incorporation assay was done to quantify proliferated cells both in keratin treated DP cells and ORS cells, which are added in Supplementary Fig. 6 and 15. The molecular expressions of Annexin V and caspase 3 was previously provided to support TGFβ2-driven apoptosis in Supplementary Fig. 21.

Supplementary Figure 27 | P-cadherin expressing germ formation of ORS cells in the presence of TGFβ2. A: Images of time-course P-cadherin expressing germ formation of ORS cells in the presence of TGFβ2 by immunofluorescent staining; DAPI, blue; phalloidin, RUNX1, red; P-cadherin, caspase 6, KRT34, green. Scale bars, 100 μm. B: Flow cytometric analysis of P-cadherin and RUNX1 of TGFβ2-treated ORS cells.

Supplementary Figure 6 | DP cell growth in the presence of hair keratin. A: Quantification of DP cell growth in the presence of keratin. *P<0.01, indicates a significant difference between control and keratin treated. (n=6; mean ± standard deviation (s.d.)). B: Image of DP cell and DP cell condensation on day 3 after keratin treatment by immunofluorescent staining; DAPI, blue; Ki67, red; Scale bars, 100μm. C: BrdU incorporation assay of DP cells in the presence of keratin. *P<0.01, indicates a significant difference between control and keratin treated. (n=4; mean ± standard deviation (s.d.)). D: Image of DP cell and DP cell condensation on day 3 after keratin treatment by immunofluorescent

staining; DAPI, blue; BrdU, red; b-catenin, green; Scale bars, 50µm.

Supplementary Figure 15 | ORS cell growth in the presence of hair keratin. **A:** Quantification of ORS cell growth in the presence of keratin. *P,0.01, indicates a significant difference between control and keratin treated. (n=6; mean ± standard deviation (s.d.)). **B:** Image of ORS cells on day 1 and 3 after keratin treatment by immunofluorescent staining; DAPI, blue; BrdU, red; β-catenin, green. Scale bars, 50µm. **C:** BrdU incorporation assay of ORS cells in the presence of keratin. *P,0.01, indicates a significant difference between control and keratin treated. (n=4; mean ± standard deviation (s.d.)).

3. Reviewer's Comment:

- The authors should clarify in the text that Keratin-treated mice showed mixed stages of hair follicles. That may stem from the typical distribution of Keratin following injection, as their new data suggests

3. Answer:

- Your comment is appreciated. As your kind comments, some sentences are added to clarify mixed stage of hair follicles in keratin-injected mice in the section of Results. The added sentences from line 114 to line 117 in the section of Results are as follows;
“Keratin injection-mediated hair growth was observed throughout the surface of the back skin of mice and mixed stages of hair follicles were also found, which might be due to the dispersion of keratin solution after injection, and the injected keratin remained up to 2 weeks after injection (Supplementary Fig. 4).”

4. Reviewer's Comment:

- Why is the control skin in the siRNA-loaded lipofectamine group at anagen?

4. Answer:

- Your comment is appreciated. If your comment is to ask the reason why we use negative siRNA-loaded lipofectamine injected mice as control group, we considered the effect of lipofectamine transfection on hair growth, which is the reason why the negative-siRNA-loaded lipofectamine treated mice was selected as control group.

5. Reviewer's Comment:

- Proliferation and apoptosis should have been done in vivo and not only in vitro following keratin injection.

5. Answer:

- Your comment is appreciated. As your kind comments, immunohistological staining of Ki67 and caspase 3 was done to show proliferation and apoptosis in keratin injected mice, which is added newly as Supplementary Fig. 37.

The added sentences from line 343 to line 354 in the section of Discussion are as follows;

“In our study, it was shown that local DP cell condensation and germ formation in TGFβ2-induced apoptotic ORS cells depend on the exposure of keratin via caspase-6 expression and consequently its-mediated keratin exposure from apoptotic cell death, as evidenced by the suppressed DP cell condensation and germ formation in caspase-6-knockdown or KRT31/KRT34-knockdown ORS cell culture, even if there was no distinct difference in TGFβ2-mediated ORS cell apoptosis. From these *in vitro* findings, it could be inferred that spatially increased keratin exposure, following gradual apoptosis during regression stage, might provide a cue to derive germ formation and new hair cycle

initiation from telogen to anagen, which spatial keratin deposit could be identified at upper void space unoccupied cells in the newly formed hair follicle (Supplementary Fig. 36). In addition, apoptosis-related keratin exposure also could be identified by similar spatial expressions of apoptosis-markers such as Annexin V and active caspase 3 and KRT 34 in the developing hair follicles containing proliferating cells (Supplementary Fig. 37).”

Supplementary Figure 37 | Proliferation and apoptosis in developing hair follicles of keratin-injected mice. Images of Annexin V, KRT34, caspase 3, active caspase 3 and Ki67 expression in hair follicles by immunohistological staining; DAPI, blue; KRT34, caspase 3, red; Annexin V, Ki67, green. Scale bars, 20 μ m or 50 μ m.

6. Reviewer's Comment:

- The point of whether Keratin is indeed released during catagen still needs to be addressed. However, it seems reasonable to publish the revised story as the notion that Keratin is released by apoptotic cells during hair regression and participates in promoting the new hair cycle is intriguing.

6. Answer:

- Your comment is appreciated. Apoptosis-related keratin exposure in the developing hair follicles was visualized by immunohistochemical staining of KRT34 and apoptosis-marker, and as shown in the newly added Supplementary Fig. 36 and 37, keratin deposit could be found at void space unoccupied by cells in the upper part of hair follicles. In addition, as your kind comments, we replaced the word “release” with “exposure” in manuscript, and added a paragraph that apoptosis-related increased keratin exposure might be a cue to derive new hair cycling.

The added and corrected sentences from line 343 to line 354 in the section of Discussion are as follows;

“In our study, it was shown that local DP cell condensation and germ formation in TGFβ2-induced apoptotic ORS cells depend on the exposure of keratin via caspase-6 expression and consequently its-mediated keratin exposure from apoptotic cell death, as evidenced by the suppressed DP cell condensation and germ formation in caspase-6-knockdown or KRT31/KRT34-knockdown ORS cell culture, even if there was no distinct difference in TGFβ2-mediated ORS cell apoptosis. From these *in vitro* findings, it could be inferred that spatially increased keratin exposure, following gradual apoptosis during regression stage, might provide a cue to derive germ formation and new hair cycle initiation from telogen to anagen, which spatial keratin deposit could be identified at upper void space unoccupied cells in the newly formed hair follicle (Supplementary Fig. 36) In addition, apoptosis-related keratin exposure also could be identified by similar spatial expressions of apoptosis-markers such as Annexin V and active caspase 3 and KRT 34 in the developing hair follicles containing proliferating cells (Supplementary Fig. 37).”

7. Reviewer’s Comment:

- Please provide a better explanation for concern #10. “Curiously, authors show that KRT31/KRT34 silencing during telogen is sufficient to inhibit hair growth. It is unclear, however, whether keratin release happens independently of catagen and cell death. If the suggested working model is correct, hair growth delay will be noticeable only on the following hair cycle and not immediately, as cell death rarely happens during telogen to anagen transition. In that case, KRT31/KRT34 silencing might delay hair progression, irrespective of keratin release to the extracellular space. Without having rigorous experimental validations of their findings, the reader cannot evaluate the extent to which the conclusions drawn are valid.”

7. Answer:

- Your comment is appreciated. As the reviewer’s concern, some sentences are added and corrected to in the section of Discussion. Apoptosis is not restricted in catagen phase, and spatial apoptosis and its related keratin exposure during telogen might be related to derive new anagen hair follicle formation, which need to be further more precisely. The added sentences from line 355 to line 390 in the section of Discussion are as follows;

The added sentences from line 355 to line 390 in the section of Discussion are as follows;

“Finally, to determine the biological function of keratin *in vivo*, the effect of downregulating KRT31/KRT34 mRNA expression on hair growth in mice was evaluated. Exogenous keratin injection in KRT31/34 knockdown mice resulted in a relatively reduced formation of catagen follicles on day 7.

This might be due to the temporal inhibition of keratin expression accompanied by stem cell differentiation into the matrix and shaft during the anagen phase, which might influence catagen formation. In addition, the formation of anagen hair follicles and hair growth were suppressed in mice with temporal downregulation of KRT31/KRT34, which could be recovered by intradermal injection of additional exogenous keratin. With the poor formation of anagen follicles in KRT31/KRT34 knockdown mice, P-cadherin and Lgr5 expressing cell population was not scarcely observed in telogen follicles in these mice (Supplementary Fig. 30, 31). *In vitro* KRT31/34 knockdown ORS cells did not show any change of ability for cell growth and differentiation into P-cadherin and Lgr5 expressing cells (Supplementary Fig. 28, 29), and such P-cadherin and Lgr5 expressing cell population could be formed by injecting exogenous keratin into KRT31/34 knockdown mice even though less than control mice (Supplementary Fig. 31). These findings indicate that alteration of hair keratin gene expression might influence hair growth following stem cell differentiation into the matrix and shaft, and hair cycle transitions might be controlled by keratin-mediated microenvironmental change. However, the reduced anagen follicle formation could not be explained in KRT31/34 knockdown mice model starting at synchronized telogen phase, which hair cycle-dependent keratin expression and apoptosis-related keratin exposure even during telogen to anagen transition need to be studied further. The expression of hair keratins is not restricted to the anagen phase, showing cell growth and differentiation-mediated hair keratin production, and are found at all stages of the hair cycle⁴¹⁻⁴³. A distinct epithelial cell population expressing Bcl-2 in secondary HG and DP was found during the telogen-anagen transition, which shows differential susceptibility to apoptosis⁴⁴. Such programmed cell death-related cellular processes were also detected during telogen, and the stimulation of autophagy following programmed cell death initiated the telogen-anagen transition⁴⁵. In addition, A study using transgenic mice overexpressing an anti-apoptotic gene reported that the inhibition of the apoptotic death of ORS cells even during anagen resulted in the early termination of hair follicle stem cell activation and proliferation, whereas the initiation of a new hair cycle was postponed by inhibiting the apoptotic death of ORS cells during telogen⁴⁶. These studies suggested that telogen might not be the only resting phase in hair growth, but also an activating phase, including DP condensation and secondary HG formation^{47,48}. These studies indicate that spatiotemporal apoptosis during the hair cycle can be an essential process in controlling hair regeneration, and our findings show that keratin can be an important factor influencing hair growth. However, these pilot observations require follow-up studies of hair keratin expression and its apoptosis-related exposure during the hair cycle using genetically modified ORS cells equipped with an on-off expression system for keratin expression and a xenograft mouse model to determine the *in vivo* mechanisms.”

8. Reviewer’s Comment:

- Authors should correct their suggested model, demonstrating that Keratin is released to the extracellular space when such data hasn’t been provided.

8. Answer:

- Your comment is appreciated. As your kind comments, the Supplementary Fig. 41 is changed.

Supplementary Figure 41 | Schematic illustration of the mechanism of keratin-mediated hair growth

REVIEWERS' COMMENTS:

Reviewer #3 (Remarks to the Author):

The revised manuscript of the study by Seong Yeong An et al., is considerably improved. The authors have adequately addressed most of the points raised by this reviewer, and as a result, the manuscript flow and data quality are much improved. I think that there are bigger questions that are still not addressed. However, it seems good to publish the revised story since it presents an intriguing idea.